# Transgenic ferret models define pulmonary ionocyte diversity and function

Feng Yuan[1], Grace N. Gasser[1], Evan Lemire[2], Daniel T. Montoro[3], Karthik Jagadeesh[3], Yan Zhang[1], Yifan Duan[4], Vitaly Ievlev[1], Kristen L. Wells[5], Pavana G. Rotti[6], Weam Shahin[1], Michael Winter[1], Bradley H. Rosen[7], Idil Evans[1], Qian Cai[1], Miao Yu[1], Susan A. Walsh[8], Michael R. Acevedo[8], Darpan N. Pandya[8], Vamsidhar Akurathi[8], David W. Dick[8], Thaddeus J. Wadas[8], Nam Soo Joo[9,10], Jeffrey J. Wine[9], Susan Birket[11], Courtney M. Fernandez[11], Hui Min Leung[12], Guillermo J. Tearney[12], Alan S. Verkman[13,14], Peter M. Haggie[13,14], Kathleen Scott[15], Douglas Bartels[1], David K. Meyerholz[16], Steven M. Rowe[11], Xiaoming Liu[1], Ziying Yan[1], Adam L. Haber[2]✉, Xingshen Sun[1]✉ & John F. Engelhardt[1]✉

Speciation leads to adaptive changes in organ cellular physiology and creates challenges for studying rare cell-type functions that diverge between humans and mice. Rare cystic fibrosis transmembrane conductance regulator (CFTR)-rich pulmonary ionocytes exist throughout the cartilaginous airways of humans[1,2], but limited presence and divergent biology in the proximal trachea of mice has prevented the use of traditional transgenic models to elucidate ionocyte functions in the airway. Here we describe the creation and use of conditional genetic ferret models to dissect pulmonary ionocyte biology and function by enabling ionocyte lineage tracing (*FOXI1*-Cre[ERT2]::ROSA-TG), ionocyte ablation (*FOXI1*-KO) and ionocyte-specific deletion of *CFTR* (*FOXI1*-Cre[ERT2]::*CFTR*[L/L]). By comparing these models with cystic fibrosis ferrets[3,4], we demonstrate that ionocytes control airway surface liquid absorption, secretion, pH and mucus viscosity—leading to reduced airway surface liquid volume and impaired mucociliary clearance in cystic fibrosis, *FOXI1*-KO and *FOXI1*-Cre[ERT2]::*CFTR*[L/L] ferrets. These processes are regulated by CFTR-dependent ionocyte transport of Cl⁻ and HCO₃⁻. Single-cell transcriptomics and in vivo lineage tracing revealed three subtypes of pulmonary ionocytes and a *FOXI1*-lineage common rare cell progenitor for ionocytes, tuft cells and neuroendocrine cells during airway development. Thus, rare pulmonary ionocytes perform critical CFTR-dependent functions in the proximal airway that are hallmark features of cystic fibrosis airway disease. These studies provide a road map for using conditional genetics in the first non-rodent mammal to address gene function, cell biology and disease processes that have greater evolutionary conservation between humans and ferrets.

Half a century ago, the first transgenic mouse containing the SV40 sequence was generated by Jaenisch and Mintz[5]. Since that time, sophisticated methods for manipulating the mouse genome have been applied to study difficult questions related to gene function, stem cell biology, organ regeneration and disease pathogenesis. Key to these mouse models is the ability to conditionally manipulate gene function in a cell-specific fashion[6]. Despite the incredible utility of transgenic mice, evolutionary bifurcations in gene function and organ cellular biology can lead to divergent phenotypes when modelling human diseases. The lung is particularly notable in this regard, where mutations in the *CFTR* gene lead to spontaneous bacterial colonization of the lung in ferrets, pigs and humans, but not mice[7]. Such phenotypic differences are thought to be the result of speciation in lung cellular anatomy and function. For example, CFTR-rich pulmonary ionocytes are associated with submucosal glands (SMGs) in the extrapulmonary and intrapulmonary cartilaginous airways of humans, pigs and

[1]Department of Anatomy and Cell Biology, Carver College of Medicine, University of Iowa, Iowa City, IA, USA. [2]Department of Environmental Health, Harvard T. H. Chan School of Public Health, Boston, MA, USA. [3]Broad Institute of MIT and Harvard, Cambridge, MA, USA. [4]Department of Biostatistics, Harvard T. H. Chan School of Public Health, Boston, MA, USA. [5]Barbara Davis Center for Childhood Diabetes, University of Colorado Anschutz Medical Campus, Aurora, CO, USA. [6]Synthetic Biology Center, Department of Biological Engineering, Massachusetts Institute of Technology, Cambridge, MA, USA. [7]Division of Pulmonary, Critical Care, Occupational, and Sleep Medicine, Department of Medicine, Indiana University School of Medicine, Indianapolis, IN, USA. [8]Department of Radiology, Carver College of Medicine, University of Iowa, Iowa City, IA, USA. [9]Cystic Fibrosis Research Laboratory, Department of Psychology, Stanford University, Stanford, CA, USA. [10]Department of Pediatrics, Stanford University School of Medicine, Stanford, CA, USA. [11]Department of Medicine, University of Alabama at Birmingham, Birmingham, AL, USA. [12]Wellman Center for Photomedicine, Massachusetts General Hospital, Boston, MA, USA. [13]Department of Medicine, UCSF, San Francisco, CA, USA. [14]Department of Physiology, UCSF, San Francisco, CA, USA. [15]Office of Animal Resources, University of Iowa, Iowa City, IA, USA. [16]Department of Pathology, University of Iowa, Iowa City, IA, USA. ✉e-mail: ahaber@hsph.harvard.edu; xingshen-sun@uiowa.edu; john-engelhardt@uiowa.edu

## Table 1 | Genomic information on CRISPR–Cas9 zygote targeting

| Ferret line | sgRNA sequence (PAM) | sgRNA genomic cleavage site* | Length of homology arms | Genomic site of homology arms[a] |
|---|---|---|---|---|
| *FOXI1*-KO | gRNA 1 5'-TGGTAAAGGCTCATCTCGGGG-3'<br>gRNA 2 5'-GCGGCCCCCCTATTCCTACTCGG-3' | gRNA 1: 72–95<br>gRNA 2: 368–391 | NA | NA |
| *FOXI1*-Cre[ERT2] | gRNA 5'-CTAGACCTCGGTGCCCTCCCTGG-3' | sgRNA: 2,579–2,602 | Right: 952 bp<br>Left: 967 bp | Right: 2,603–3,554<br>Left: 1,635–2,602 |
| *CFTR*[L/L] | gRNA-right: 5'-TAGCTAAATCCTTTGGGAACTGG-3'<br>gRNA-left: 5'-TATTTCCTGTTGAATGATGGAGG-3' | gRNA-right: 108,071–108,093<br>gRNA-left: 107,612–107,631 | Right: 777 bp<br>Left: 703 bp | Right: 108,077–108,853<br>Left: 106,922–107,624 |

[a]ENSEMBL contig and bp length: *CFTR* ENSMPUG00000007138, 169,387 bp; *FOXI1* ENSMPUG00000012982, 2,602 bp.
bp, base pair; gRNA, guide RNA; NA, not applicable; PAM, protospacer adjacent motif; sgRNA, single-guide RNA.

ferrets[1,2,8], whereas mice lack ionocytes, glands and cartilage in their intrapulmonary airways.

Pulmonary ionocytes constitute approximately 0.5–1% of the cartilaginous airway epithelium in humans and the proximal trachea of mice[1,2]. Although studies on ionocytes (also called mitochondrial-rich cells) in fish gills and the frog skin epithelium have demonstrated specialized functions in ion transport and pH regulation[9,10], their role in the mammalian respiratory system remains largely unknown. The forkhead box I1 (FOXI1) transcription factor is required for progenitor specification of ionocytes in multiple species. Although *Foxi1*-knockout (*Foxi1*-KO) mouse airway epithelia have decreased *Cftr* expression, they paradoxically have 'Cftr-like' anion transport that exceeds wild-type airway epithelia[1], suggesting this species has other cell types and/or channels that can compensate for the lack of CFTR-expressing ionocytes. Given that cystic fibrosis mice do not spontaneously develop lung disease, we sought to study pulmonary ionocyte function in the ferret, a species that develops a cystic fibrosis lung disease phenotype similar to humans[3,4].

## Ionocytes regulate airway fluid properties

To investigate ionocyte functions in ferrets, we disrupted the *FOXI1* gene (*FOXI1*-KO) using CRISPR–Cas9-mediated gene editing in zygotes (Extended Data Fig. 1a–d and Table 1). *FOXI1*-KO ferret kidneys lacked expression of the FOXI1 protein and transcripts enriched in intercalated cells (Extended Data Fig. 1e,f and Supplementary Fig. 1), consistent with FOXI1 being required for specification of intercalated cells in the distal renal tubular epithelium[11]. *FOXI1*-KO ferrets developed cystic kidneys and had fragile health with most not surviving until weaning (8 weeks of age).

To assess pulmonary ionocyte function in more detail, we measured CFTR-mediated ion transport in differentiated tracheal airway cultures derived from age-matched wild-type and *FOXI1*-KO donors and grown at an air–liquid interface (ALI). Polarized *FOXI1*-KO airway cultures demonstrated cAMP-inducible CFTR-mediated $Cl^-$ and $HCO_3^-$ currents that were 69% ($P < 0.0001$) and 68% ($P < 0.0001$) decreased, respectively, compared with wild-type cultures (Fig. 1a,b). These *FOXI1*-KO cultures also demonstrated near absent messenger RNA expression of ionocyte markers (*FOXI1*, *BSND* and *ASCL3*), as well as a 50% decrease in *CFTR* mRNA expression (Fig. 1c). Furthermore, freshly excised *FOXI1*-KO ferret trachea produced cAMP-inducible CFTR-mediated $Cl^-$ currents that were 73% ($P = 0.0025$) lower than age-matched wild-type trachea (Extended Data Fig. 7a).

Active fluid absorption and secretion are fundamental processes that maintain airway clearance, and CFTR plays a major role in regulating both fluid absorption and secretion by directing the movement of $Na^+$ and $Cl^-$ ions across the airway epithelium[12]. Studies have shown that fish gill ionocytes pump out excess $Na^+$ and $Cl^-$ ions in salt water environments, and function to retain $Na^+$ and $Cl^-$ ions in freshwater environments against a major concentration gradient[10,13]. Thus, we proposed that pulmonary ionocytes might have similar roles in regulating airway surface liquid (ASL) homeostasis. To this end, we measured ASL height and the rate of fluid absorption in polarized *FOXI1*-KO, *CFTR*-KO and wild-type ALI cultures following a small volume challenge to the apical surface. Similar to human cystic fibrosis cultures[12], *FOXI1*-KO ALI cultures demonstrated a 2.3-fold ($P < 0.0092$) slower fluid absorption rate (Fig. 1d) and 40% reduction ($P < 0.0001$) in ASL height at 24 h of equilibration, compared with wild-type cultures (Fig. 1e and Extended Data Fig. 2e). Notably, the absorptive and secretory phases of ASL height equilibration observed in wild-type cultures were both lost in *FOXI1*-KO and *CFTR*-KO cultures (Fig. 1e and Extended Data Fig. 2e). Micro-optical coherence tomography (μOCT) imaging confirmed a decreased ASL depth in *FOXI1*-KO tracheae compared with wild-type tracheae ($P < 0.001$) (Fig. 1f and Extended Data Fig. 1h), whereas the periciliary liquid (PCL) layer height and ciliary beat frequency (CBF) were unchanged between genotypes (Extended Data Fig. 1h–j). Similar to human and pig cystic fibrosis cultures[14–16], *FOXI1*-KO and *CFTR*-KO ferret cultures failed to alkalinize the ASL following CFTR stimulation compared with wild-type cultures ($P < 0.0001$) (Fig. 1g). However, baseline pH was significantly lower in *CFTR*-KO cultures ($P < 0.0001$) compared with *FOXI1*-KO and wild-type cultures (Fig. 1g). Notably, *Foxi1*-KO and *Cftr*-KO mouse airway ALI cultures lack ASL pH and volume abnormalities[1,16], reinforcing evolutionary divergence of pulmonary ionocyte functions in mouse versus human and ferret airways.

CFTR-mediated $Cl^-$ and $HCO_3^-$ secretion are thought to control mucus viscosity in multiple cystic fibrosis-affected organs by respectively regulating fluid secretion and $Ca^{2+}$ chelation required for proper unfolding of mucus[17,18]. In cystic fibrosis airways, defects in these two CFTR-dependent processes are thought to increase mucus viscosity and impair mucociliary clearance (MCC)[17]. As predicted by the observed decreases in ASL volume and $HCO_3^-$ secretion of *FOXI1*-KO and *CFTR*-KO ferret ALI cultures, both genotypes demonstrated increased ASL viscosity relative to wild-type controls (Fig. 1h). In agreement with these findings, the absence of ionocytes in *FOXI1*-KO ferrets led to a significant ($P < 0.0001$) reduction in tracheal MCC compared with wild-type controls (Fig. 1i,j, Extended Data Fig. 1g and Supplementary Video 1). The level of MCC impairment in *FOXI1*-KO ferrets was also similar to *CFTR*[G551D/G551D] cystic fibrosis ferrets removed from treatment with a CFTR modulator (VX-770) that corrects the gating defect in the CFTR[G551D] channel (Fig. 1i,j and Supplementary Video 2). Furthermore, *CFTR*[G551D/G551D] ferrets maintained on VX-770 had clearance rates equivalent to wild-type ferrets, demonstrating the dependence of clearance on CFTR function (Fig. 1i,j). Collectively, these in vitro and in vivo data demonstrate that pulmonary ionocytes are essential regulators of ASL properties required for effective MCC in the proximal airways and their absence largely recapitulates the cystic fibrosis phenotype.

## Generation of transgenic ferret models

To study CFTR function in pulmonary ionocytes, we generated two transgenic ferret models capable of lineage tracing ionocytes

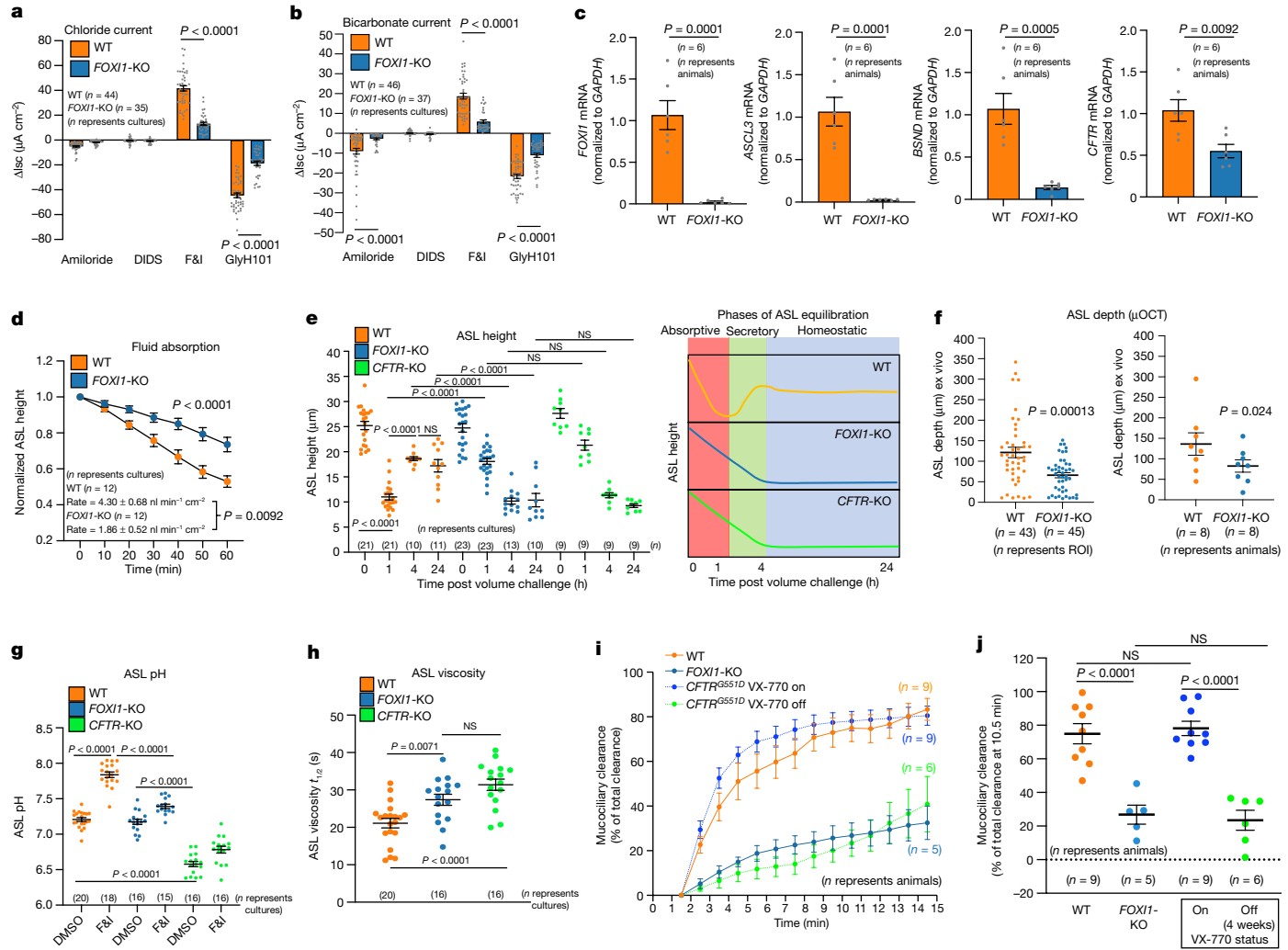

**Fig. 1 | Depletion of pulmonary ionocytes impairs CFTR-mediated regulation of ASL volume, pH, viscosity and MCC. a,b,** Change in short circuit current (ΔIsc) for Cl⁻ (**a**) and HCO₃⁻ (**b**) from ferret ALI cultures of the indicated genotypes. F&I, forskolin and IBMX; WT, wild type. **c,** RT–qPCR for ionocyte-enriched transcripts in ferret ALI cultures. **d,** Fluid absorption showing the ASL height normalized to time zero following small volume addition to the apical surface. Fluid absorption rates are marked on the graph. **e,** Changes in ASL height over time following small volume challenge (at 0 h) to ALI cultures. Right schematic depicts absorptive and secretory phases of ASL equilibration that are altered in *FOXI1*-KO and *CFTR*-KO cultures. **f,** μOCT imaging of ferret tracheal ASL depth. ASL depths were compared by region of interest (ROI) and animal averages. **g,** Alkalinization of ASL pH in ALI cultures following CFTR stimulation with forskolin/IBMX. **h,** ASL viscosity in ALI cultures. **i,** In vivo ferret tracheal MCC measured by PET/CT for the indicated

genotypes and CFTR modulator (VX-770) treatment status. **j,** Percentage tracheal clearance at 10.5 min following instillation of radioactive tracer for ferrets evaluated in **h**. Data are mean ± s.e.m. for the *n* indicated in each graph (ALI cultures or animals). Statistical significance was determined by: one-way analysis of variance (ANOVA) and Sidak's multiple comparisons test (**a**–**c**); two-way ANOVA for graphed genotypic differences and two-tailed Student's *t*-test for rates (**d**); one-way ANOVA and Tukey's multiple comparison test (**e**,**g**,**h**,**j**); ROI by *t*-tests with pooled s.d. by *R* and animal averages by paired one-tailed Student's *t*-test (**f**). The numbers of independent ferrets used for each experiment were: 12 WT, 9 *FOXI1*-KO (**a**); 10 WT, 8 *FOXI1*-KO (**b**); 6 in each group (**c**,**d**); 9 WT, 10 *FOXI1*-KO, 3 *CFTR*-KO (**e**); 8 in each group (**f**); 6 WT, 5 *FOXI1*-KO, 4 *CFTR*-KO (**g**); 6 WT, 4 *FOXI1*-KO, 3 *CFTR*-KO (**h**); 9 WT, 5 *FOXI1*-KO, 9 *CFTR* (**i**,**j**). DIDS, 4,4′-diisothiocyanato-stilbene-2,2′-disulfonic acid; NS, not significant; RT–qPCR, quantitative PCR with reverse transcription.

(*FOXI1*-IRES-Cre^ERT2; hereafter called *FOXI1*-Cre^ERT2) and conditional deletion of *CFTR* (herein called *CFTR^L/L*) using CRISPR homology-directed repair (HDR) (Extended Data Fig. 2a,b). To this end, we performed HDR in ferret zygotes to target IRES-Cre^ERT2 into the 3′ untranslated region (UTR) of the *FOXI1* gene and to flank *CFTR* exon-16 with *loxP* sites (Table 1).

To enable lineage tracing of ionocytes in vivo, *FOXI1*-Cre^ERT2 founders were bred to ROSA-TG Cre reporter ferrets[19], which harbour a transgene that converts from Tomato to EGFP expression upon tamoxifen activation of Cre^ERT2. Tamoxifen induction of *FOXI1*-Cre^ERT2::ROSA-TG ferrets labelled pulmonary ionocytes in both the surface airway epithelium and SMGs (Fig. 2a) and RNAscope confirmed these cells co-expressed *CFTR*,

*FOXI1* and *EGFP* mRNA (Fig. 2b). We also observed EGFP⁺ cells labelled in organs known to be enriched in *FOXI1*-expressing ionocyte-like cells in humans (kidney cortex and epididymis), in addition to previously unstudied oesophageal glands (Fig. 2a).

## Ionocytes require CFTR to transport anions

Given the small numbers of pulmonary ionocytes in the proximal airway, it has remained unclear whether ionocytes directly participate in anion transport or rather serve some sensory role for directing anion movement through other luminal cell types. To enable single-cell analysis of CFTR-dependent anion movement through

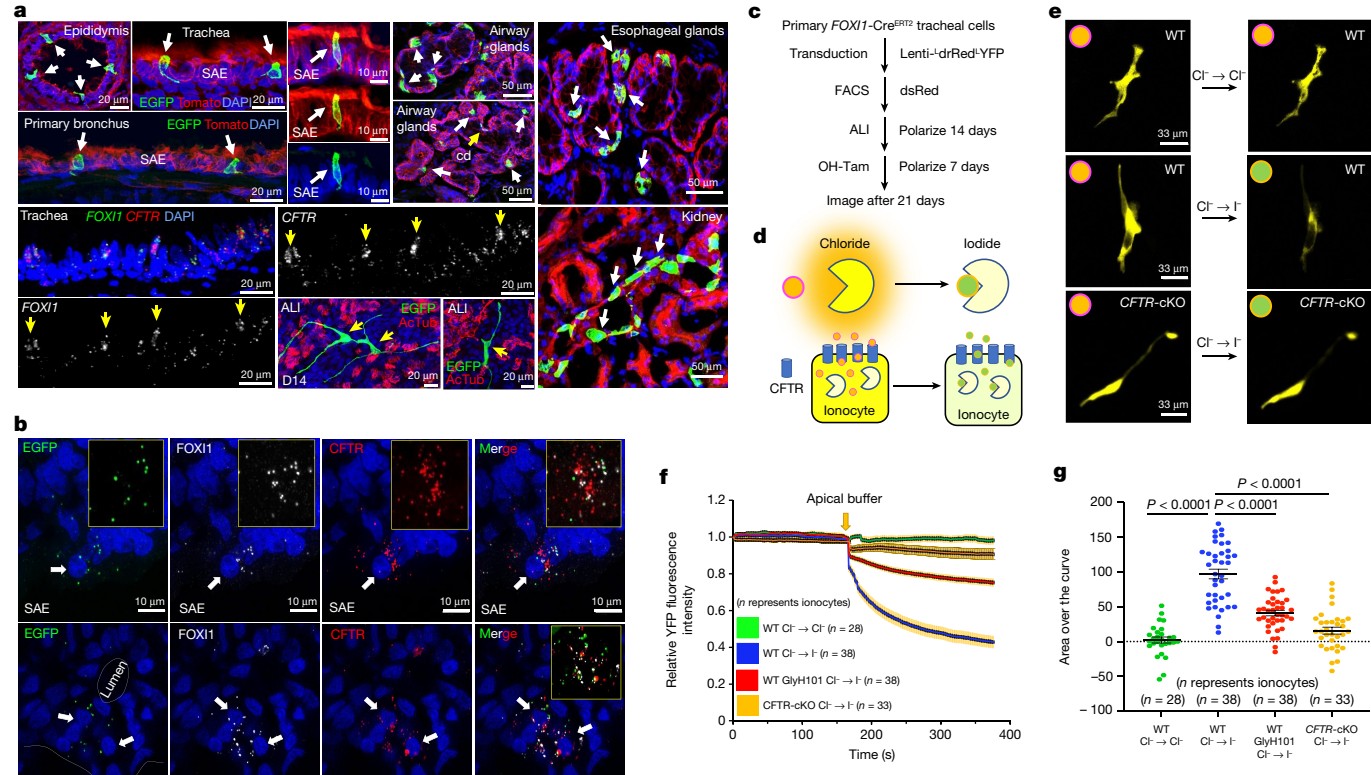

**Fig. 2 | Pulmonary ionocytes directly transport anions in a CFTR-dependent manner. a**, Ionocyte lineage tracing in tamoxifen-induced *FOXI1*-Cre[ERT2]::ROSA-TG ferrets showing EGFP[+] ionocytes (white arrows) in the seminiferous tubules of the epididymis, surface airway epithelium (SAE), airway submucosal glands (SMGs), kidney tubules, oesophageal glands and ALI cultures (14 days of differentiation). cd, airway SMG collecting duct. Bottom left panels show RNAscope for *FOXI1* and *CFTR* in the trachea. Representative image of *n* = 3 independent ferrets. **b**, RNAscope demonstrating colocalization of *EGFP*, *FOXI1* and *CFTR* transcripts in proximal tracheal SAE and SMGs of tamoxifen-treated *FOXI1*-Cre[ERT2]::ROSA-TG ferrets. Arrows mark traced ionocytes. Representative image of *n* = 3 independent ferrets. **c**, Workflow for halide quenching measurements in primary ALI cultures of YFP sensor-expressing pulmonary

ionocytes. **d**, Schematic of approach for evaluating apical halide movement into pulmonary ionocytes using the YFP sensor. **e**, Representative live-cell images showing pulmonary ionocyte YFP fluorescence following the indicated apical halide exchange in WT and *CFTR*-cKO ALI cultures. The halide colour scheme is from **d**. Representative image of *n* = 99 ionocytes. **f**, Relative single-cell ionocyte YFP fluorescence intensity data following apical halide exchange for the indicated ALI genotype (*n* represents ionocytes): *n* = 28 (WT, Cl[−]); *n* = 38 (WT, I[−]); *n* = 38 (WT, GlyH101, I[−]); *n* = 33 (*CFTR*-cKO, I[−]). Each group has three ferret donors. **g**, Area over the curve of relative single-cell ionocyte YFP fluorescence intensity for the data presented in **f**. Data are mean ± s.e.m. *P* values for the indicated comparisons were determined by one-way ANOVA and Tukey HSD posttest using R.

pulmonary ionocytes, we created *FOXI1*-Cre[ERT2] airway basal cells with an integrated Cre-activatable fluorescent halide sensor (herein called *FOXI1*-Cre[ERT2]::YFP[H148Q/I152L]), which provides rapid and quantitative assessment of halide transport in cells[20] via I[−] sensitive quenching of fluorescence relative to Cl[−] (Fig. 2c,d and Extended Data Fig. 2c). The frequency of ionocyte labelling following 4-hydroxytamoxifen (OH-Tam) activation was similar between *FOXI1*-Cre[ERT2]::YFP[H148Q/I152L] and *FOXI1*-Cre[ERT2]::ROSA-TG ALI cultures (Extended Data Fig. 2d versus Extended Data Fig. 8a).

To evaluate anion transport through ionocytes, OH-Tam-treated *FOXI1*-Cre[ERT2]::YFP[H148Q/I152L] ALI cultures were preincubated with chloride-containing buffer on the basolateral side during baseline measurements and then apically challenged with a small volume of I[−], Cl[−] or Na-gluconate (Cl[−] free) buffer containing forskolin and 3-isobutyl-l-methylxanthine (IBMX) to stimulate CFTR (Fig. 2c–e). Results from these experiments demonstrated that only apical I[−], but not Cl[−] or Na-gluconate, stimulated quenching of YFP fluorescence intensity (*P* < 0.0001) (Fig. 2e–g, Extended Data Fig. 2g–i and Supplementary Videos 3 and 4). Furthermore, addition of the CFTR inhibitor GlyH101 significantly (*P* < 0.0001) inhibited I[−] quenching of YFP and thus apical anion uptake into ionocytes (Fig. 2f,g and Extended Data Fig. 2j). Similar imaging experiments demonstrated that NKCC1 was required for ionocyte basolateral uptake of I[−], but

only when the ASL was dehydrated (Extended Data Figs. 2f, 3a–d and Supplementary Results).

To conclusively demonstrate that CFTR was required for I[−] uptake into ionocytes, we performed similar experiments on *FOXI1*-Cre[ERT2]::YFP[H148Q/I152L]::CFTR[L/L] (*CFTR*-cKO) ALI cultures (Extended Data Fig. 2a–c). Following apical addition of I[−], *CFTR*-cKO ionocytes had significantly (*P* < 0.0001) impaired YFP quenching compared with wild-type controls and achieved greater inhibition than that observed following CFTR inhibition with GlyH101 (Fig. 2e–g, Extended Data Fig. 2k and Supplementary Video 5). Last, positron emission tomography and computed tomography (PET/CT) imaging of tracheal MCC in *FOXI1*-Cre[ERT2]::CFTR[L/L] ferrets was markedly reduced following in vivo *CFTR*-deletion, compared with baseline MCC measurements in the same animals before deletion of *CFTR* in ionocytes (Extended Data Fig. 3e,f). Therefore, we conclude that ferret pulmonary ionocytes directly transport anions in a CFTR-dependent manner to facilitate MCC.

## Ferret proximal airway single-cell atlas

We profiled a total of 94,664 tracheal epithelial cells from wild-type (*n* = 4), *FOXI1*-KO (*n* = 4) and *FOXI1*-Cre[ERT2]::ROSA-TG (*n* = 8) differentiated ALI cultures derived from 12 independent donor ferrets

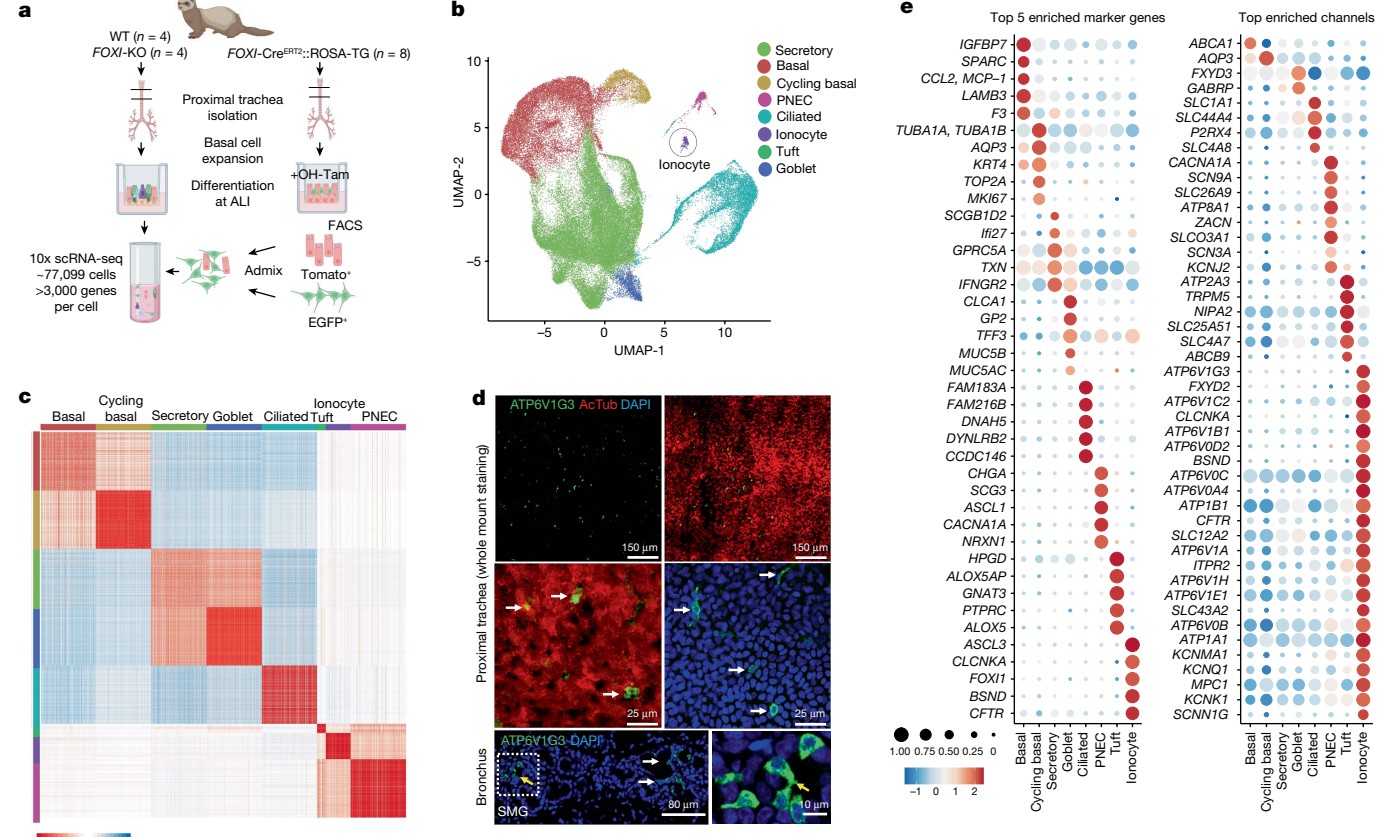

**Fig. 3 | Single-cell expression atlas of ferret proximal airway epithelial cells.** **a**, Study overview for scRNA-seq and ionocyte enrichment. Created with BioRender.com. **b**, UMAP of total tracheal epithelial cells captured across all ferret genotypes (WT, *FOXI1*-KO and *FOXI1*-Cre[ERT2]::ROSA-TG), coloured by broad cell type. **c**, Cell–cell Pearson correlation coefficient (*r*, colour bar) between each pair of cells (large clusters down-sampled to 200 cells for visualization). **d**, Top, ferret tracheal whole-mount immunostained for ATP6V1G3 (ionocyte) and AcTub (ciliated cells). Bottom, immunofluorescence staining of ferret intralobar bronchial SMGs for ATP6V1G3 (ionocyte). White arrows mark pulmonary ionocytes. Representative image of *n* = 3 independent ferrets. **e**, Top five enriched marker genes (left) and top enriched channels (right), showing the expression levels and fraction of each cell type that expresses them.

using 10X Chromium droplet-based 3′ single-cell RNA sequencing (scRNA-seq) (Fig. 3a). Methods for enriching rare ionocytes in our dataset involve upstream isolation of lineage-traced (EGFP[+]) cells by fluorescence-activated cell sorting (FACS) (Fig. 3a). After quality control filtering (Extended Data Fig. 4a), we retained 77,099 high-quality cells for further analysis. Unsupervised partitioning using the Louvain algorithm gave rise to eight distinct clusters which we annotated on the basis of known human cell-type gene expression signatures: basal, cycling basal, secretory, goblet, ciliated, pulmonary neuroendocrine cell (PNEC), tuft and ionocyte (Fig. 3b,c,e and Extended Data Fig. 4b,f,g), recapitulating the cell types we previously observed using scRNA-seq data obtained on mouse and human airways[1,2]. We then defined gene expression signatures for ferret proximal airway epithelial cells (Extended Data Fig. 4b,f and Supplementary Table 1), including cell-type-specific ion channels (Fig. 3e) and the complete channelome of all cell types (Extended Data Fig. 5 and Supplementary Table 9). As anticipated, pulmonary ionocytes were not identified in *FOXI1*-KO trachea and airway cultures (Extended Data Figs. 6e–g,j–m and 11g,h), consistent with FOXI1 being required for ionocyte specification[1,2]. Multilineage differentiation potential of ferret airway basal cells to ciliated and secretory cells was not affected by disruption of *FOXI1* (Extended data Fig. 6d–f); however, PNECs and tuft cells were significantly reduced (false discovery rate (FDR) < 0.05, 90% prediction interval) or increased (FDR < 0.05, 90% prediction interval), respectively, in *FOXI1*-KO cultures (Extended Data Fig. 6g). The reduction of PNEC numbers in *FOXI1*-KO cultures was further confirmed by immunostaining and quantification (Extended Data Fig. 11g,h).

We profiled 449 pulmonary ionocytes from wild-type and *FOXI1*-Cre[ERT2]::ROSA-TG ferret airway cultures and defined their consensus gene expression signature. Ferret pulmonary ionocytes were specifically enriched in four ATP6V0 and nine ATP6V1 family proton pumps, chloride channels (*CFTR* and *CLCNKA*), *BSND* (activator subunit of *CLCNKA*), NKCC1 basolateral Na-K-Cl symporter (*SLC12A2*), potassium channels (*KCNK1*, *KCNQ1*, *KCNK5* and *KCNMA1*), ENaC channel α and γ subunits (*SCNN1G* and *SCNN1A*) and the ATPase Na[+]/K[+] transporting channel (*ATP1B1*), consistent with unique functions in ion transport (Fig. 3e, Extended Data Fig. 4e and Supplementary Table 1).

We observed compensatory changes in gene expression of ion and water channels (83 genes upregulated and 53 genes downregulated) when comparing *FOXI1*-KO with wild-type cultures (Extended Data Fig. 6h and Supplementary Table 2). Among the most significantly upregulated genes in *FOXI1*-KO basal cells, ciliated cells, goblet cells, secretory cells and PNECs were aquaporins (*AQP3*, *AQP4*, *AQP5*). Notably, the ionocyte-enriched basolateral Na-K-Cl symporter (NKCC1/*SLC12A2*), which is required for CFTR-mediated salt and fluid secretion[17,21], was significantly upregulated in *FOXI1*-KO basal cells, goblet and secretory cells. *SLC26A9*, a gene that has been linked to cystic fibrosis disease severity and CFTR modulator responsiveness[22,23], was also significantly upregulated in *FOXI1*-KO goblet cells. Significantly downregulated genes in *FOXI1*-KO basal, secretory and ciliated cells included *ATP12A*, a proton pump involved in acidification of cystic fibrosis ASL[16], and *ATP1B1*, an Na[+]/K[+] ATPase.

## Ionocyte localization in ferret airways

We assessed the localization of functionally relevant ionocyte-enriched ion channels. To this end, we characterized CFTR, NKA (Na-K-ATPase) and ATP6V1G3 (H⁺ transporter) expression patterns in pulmonary ionocytes of ferret trachea, primary bronchi and intralobar bronchi (Extended Data Fig. 7b–k and Supplementary Video 7). Immunostaining for the ionocyte marker ATP6V1G3 confirmed their presence in the tracheal surface epithelium and SMGs (Fig. 3d). Notably, ionocytes were enriched in the airway surface epithelium within intercartilaginous zones above SMGs, compared with the membranous regions composed of trachealis muscle (Extended Data Fig. 6a–c). Consistent with other species, CFTR was most highly expressed in ferret pulmonary ionocytes and at lower levels in other cell types (Fig. 3e and Extended Data Figs. 4e,f and 6j). Similar to fish gill ionocytes, NKA, CFTR and ATP6V1G3 were enriched in pulmonary ionocytes. CFTR protein localized to what we have termed an ionocyte 'apical cap', whereas ATP6V1G3 localized to both the ionocyte apical cap and cytoplasm (Extended Data Fig. 7b,c,h and Supplementary Video 7). NKA protein was enriched within a basolateral rim around the apical cap (Extended Data Fig. 7d). Notably, single ionocytes can have more than one apical cap (Extended Data Fig. 7c), perhaps enhancing their capacity to direct salt and fluid movement.

We next defined how channel composition in ionocytes differs regionally at distinct locations in the airway. For example, proximal tracheal ionocytes express higher levels of the NKA transporter compared with primary bronchus (Extended Data Fig. 7d). Greater numbers of ionocytes were found in trachea and primary bronchus (Extended Data Fig. 7e,h), whereas ionocytes in secondary bronchi were primarily located in SMG primary ducts and collecting ducts (Extended Data Fig. 7i,f). Notably, we found ATP6V1G3⁺ ionocyte expansion in the intrapulmonary bronchial SMG ducts of $CFTR^{G551D/G551D}$ cystic fibrosis ferrets removed from treatment with a CFTR modulator (VX-770) (Extended Data Figs. 7j,k and 8c,d). ALI cultures derived from cystic fibrosis ferret tracheal airway basal cells demonstrated a lack of CFTR-mediated currents (Extended Data Fig. 7l) and a significant increase in pulmonary ionocytes as assessed by the expression of transcription factors $FOXI1$ and $ASCL3$ (Extended Data Fig. 7m,n).

## Ionocyte fragility enhances ambient RNA

Our single-cell data significantly underestimated the frequency of ionocytes in proximal airway epithelia. Ionocyte-enriched 10X samples loaded 14,966 EGFP⁺ cells and 77,145 Tomato⁺ cells, which were admixed following FACS isolation before running on eight 10X lanes. Cell viability following FACS isolation was 61.8 ± 6.3 for Tomato⁺ cells and 73.0 ± 2.5% EGFP⁺ cells ($n = 4$). Following sequencing, only 434 ionocytes were identified compared with 37,299 cells in the Tomato⁺ population. This produced an apparent recovery rate that was about 16-fold lower for EGFP⁺ ionocytes compared with other Tomato⁺ cell types. This raised the possibility that ambient mRNAs derived from fragmented ionocytes could be problematic when attempting to use scRNA-seq data to assign the proportion of cells that express a transcript such as $CFTR$. Indeed, we observed significant reductions in the percentages of basal cells (FDR < 0.0001), secretory cells (FDR < 0.0001) and ciliated cells (FDR < 0.05) that expressed $CFTR$ in $FOXI1$-KO cultures lacking ionocytes (Extended Data Fig. 6i), supporting the notion that ionocyte damage during 10X capture may raise ambient RNA for genes highly expressed in ionocytes. Consistent with this notion, there was a significant correlation of apparent secretory cell expression of ionocyte marker genes ($CFTR$, $ASCL3$, $FOXI1$, $BSND$, $CLCNKA$ and $ATP6V1C2$) with the number of ionocytes detected in each ALI culture captured by 10X (Extended Data Fig. 9a). A subset of these genes also demonstrated significant correlations with apparent ciliated and basal cell expression (Extended Data Fig. 9a).

To expand this analysis, we compared ionocyte proportion correlations for all cell types with the top 30 most ionocyte-specific transcripts and used the top 30 basal cell-specific transcripts as a control set of non-ionocyte-associated transcripts (Extended Data Fig. 9b). We reasoned that if ionocyte damage or loss was contributing to spurious detection of ionocyte-associated transcripts in non-ionocytes, then we would observe correlations between ionocyte proportion and ionocyte marker transcripts, but not for basal cell marker transcripts. This trend was indeed clearly observable, particularly for secretory cells, which showed a significant positive correlation for their expression of 21 of 30 ionocyte markers. By contrast, secretory cell expression of only 1 of 30 basal cell markers was positively correlated. Putative spurious expression of ionocyte transcripts was also observed in ciliated cells (13 of 30) and to a lesser extent in basal cells (5 of 30), implicating cell-type-specific characteristics that are more prone to binding ambient RNA. Notably, the transcript type also seemed to affect its involvement as an ambient RNA target (for example, $ASCL3$ versus $ATP1B1$ in secretory cells; $ATP6V1C2$ in four cell types) (Extended Data Fig. 9b). Together, this analysis provides evidence that damage during 10X capture may raise ambient RNA for genes highly expressed in ionocytes, particularly in secretory cells. These findings may partially explain recent claims that secretory cells are the predominant cell type expressing $CFTR$ (ref. 24).

## FOXI1-lineage rare cell progenitor

Our single-cell data suggested a large difference in the recovery rate for EGFP⁺ ionocytes compared with other Tomato⁺ cells. Further investigation into this finding showed that enriched EGFP⁺ cells contained tuft cells (approximately 7%), PNECs (approximately 59%) and ionocytes (approximately 34%), demonstrating that $FOXI1$-Cre^ERT2 lineage-labelled all rare cell populations (Extended Data Fig. 4c–e). Of note, lineage tracing in these scRNA-seq studies was performed using an approach analogous to pulse-seq, by treating actively differentiating basal cell cultures with OH-Tam on days 1–17 of moving to ALI. This was done because a greater number of EGFP⁺ cells were observed at full differentiation (day 21), as opposed to OH-Tam treatment initiating on day 14 as performed for functional halide sensor assays. To confirm the differences in rare cell labelling using these two $FOXI1$-Cre^ERT2 tracing protocols, we colocalized the pan ionocyte-specific marker ATP6V1G3 with EGFP. In cultures treated with OH-Tam on days 14–22 of differentiation, 100% of EGFP⁺ cells also expressed ATP6V1G3 (Extended Data Fig. 10d,e). By contrast, OH-Tam labelling on days 1–17 produced 40 ± 3.2% EGFP⁺ATP6V1G3⁺ and 60 ± 3.2% EGFP⁺ATP6V1G3⁻ cells (Extended Data Fig. 10c,e). Taking into account that around 40% of the 14,966 EGFP⁺ cells loaded onto the 10X were ATP6V1G3⁺ ionocytes, we estimated the recovery rate of viable ionocytes was 7.3% compared with 48% for the Tomato⁺ population, emphasizing the limitation of 10X for quantifying the frequency of ionocytes in mixed populations of airway epithelial cells.

Previous working models for rare cell types in the mouse trachea have suggested that tuft cells, PNECs and ionocytes are independently specified by Krt5⁺ basal cells[1]. More recently, it has been suggested that tuft cells are a common precursor of human PNECs and ionocytes[25]. Our unintended $FOXI1$-Cre^ERT2 pulse-seq results demonstrate that $FOXI1$ is expressed in a common rare cell progenitor of tuft cells, PNECs and ionocytes (Extended Data Fig. 4c–e). We further confirmed the existence of EGFP⁺SYP⁺ PNECs by immunostaining in $FOXI1$-Cre^ERT2 cultures treated with OH-Tam on days 1–17 (Extended Data Fig. 10f,g). To assess the in vivo relevance of this finding during airway development, we performed fate mapping in $FOXI1$-Cre^ERT2 neonatal and adult ferrets (Extended Data Fig. 11). Findings from the study in neonates confirmed the existence of EGFP⁺SYP⁺ PNECs and EGFP⁺TRPM5⁺ tuft cells by whole-mount tracheal staining and demonstrated that $FOXI1$-Cre traced EGFP⁺ cells included ionocytes (74.5%,

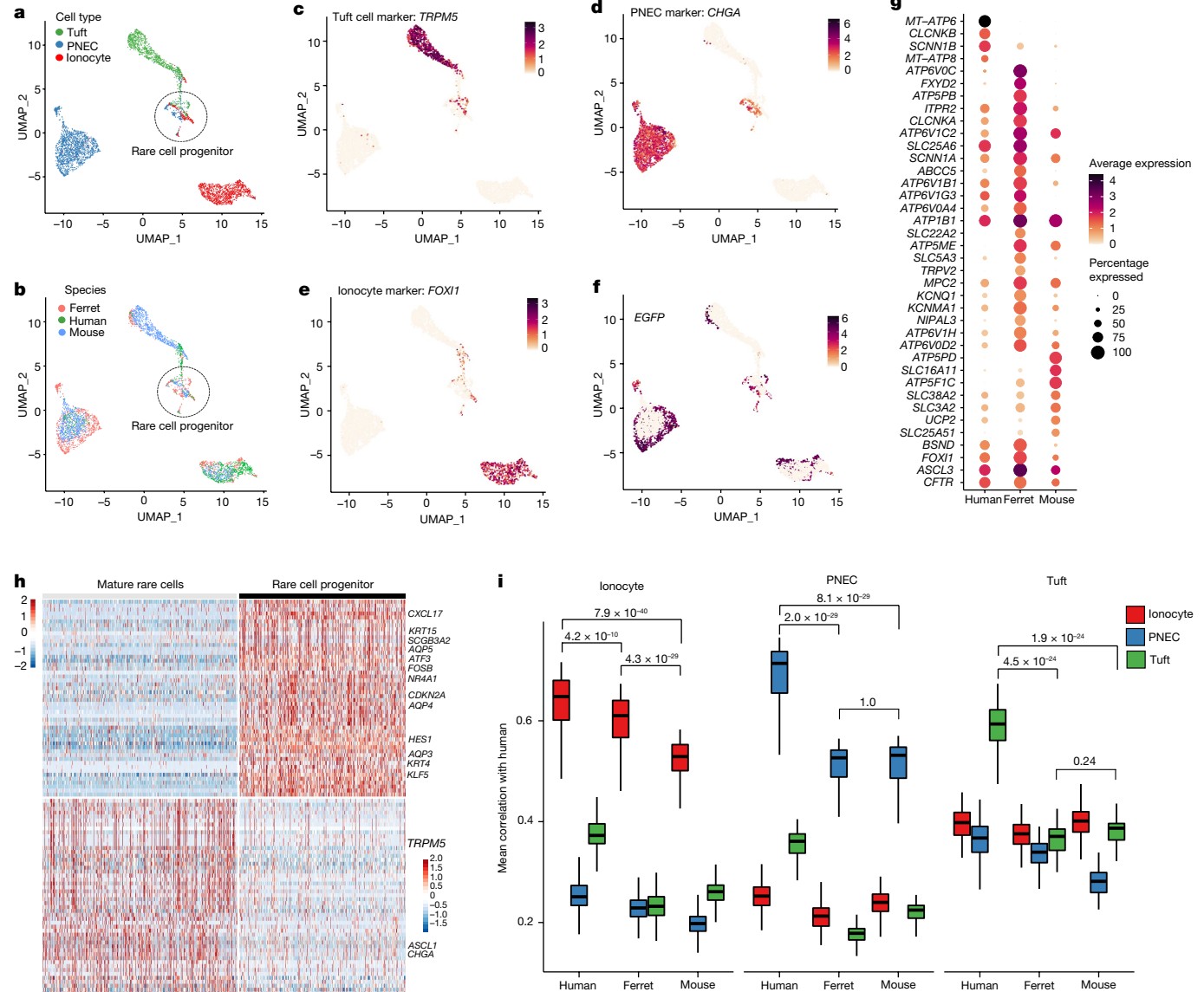

**Fig. 5 | Rare cell-type comparisons from the proximal airway epithelium of human, ferret and mouse. a**, UMAP of rare cell type transcriptomes across human, ferret and mouse, coloured by rare cell type (tuft, neuroendocrine/PNEC, ionocyte). **b**, UMAP of rare cell types across human, ferret and mouse, coloured by species. **c**, Expression of rare cell-type markers across rare cell clusters. UMAP plot shows cells coloured by expression ($\log_2(TPM+1)$, colour bar) of tuft marker *TPRM5*. **d**, UMAP plot shows cells coloured by expression of PNEC marker *CHGA*. **e**, UMAP plot shows cells coloured by expression of ionocyte marker *FOXI1*. **f**, UMAP plot of *EGFP* expression marks *FOXI1*-Cre$^{ERT2}$:: ROSA-TG lineage-labelled cells from ferret ALI scRNA-seq experiments including the common rare cell progenitor. **g**, Ion channel gene expression levels and fraction of ionocytes that express each gene across human, ferret and mouse (*MT-ATP6*, *MT-ATP8*, *ATP5PD* and *CLCNKB* are not annotated in ferret

genome and thus show no expression). **h**, Gene expression signatures of rare cell progenitors compared with mature rare cells. **i**, Interspecies comparison of mouse and ferret rare cell-type transcriptional signatures with those of human. Ferret ionocytes are transcriptionally more similar to human ionocytes. Boxplots are standard: lower and upper hinges correspond to the first and third quartiles (the 25th and 75th percentiles), and the upper and lower whiskers extend from the hinge to the largest or smallest values, respectively, no further than $1.5 \times IQR$ from the hinge where IQR is the interquartile range or distance between the first and third quartiles. Centre shows the mean. Statistical significance was determined by Wilcoxon test for the marked comparisons. $n = 1,655$ cells from 12 ferret donors, $n = 1,640$ cells from 9 mice donors, $n = 885$ cells from 60 human donors.

ATP6V1G3$^+$), PNECs (8.6%, SYP$^+$) and tuft cells (6.5%, TRPM5$^+$) (Extended Data Fig. 11a,b,d,e). *FOXI1*-Cre traced PNECs were found interacting with nerve fibres (Extended Data Fig. 11d). By contrast, lineage tracing of adult *FOXI1*-Cre$^{ERT2}$::ROSA-TG ferrets demonstrated that 95.7% of EGFP$^+$ cells were ATP6V1G3$^+$ ionocytes (Extended Data Fig. 11a–c) and no traced PNECs or tuft cells could be found (Extended Data Fig. 11f). The observed 4.3% of EGFP$^+$ATP6V1G3$^-$ cells in adult trachea is consistent with a small fraction of ionocytes (approximately 4%) demonstrating no *ATP6V1G3* expression by scRNA-seq (Fig. 4c). Thus, we conclude

that a shared *FOXI1*-lineage progenitor gives rise to ionocytes, PNECs and tuft cells during ferret airway development.

## Three subtypes of pulmonary ionocytes

Ionocytes have been called 'mitochondrial-rich cells' and 'chloride secreting cells' in the fish gill[26]. Subtypes of ionocytes have been identified in rainbow trout, killifish, tilapia and zebrafish on the basis of differing transporting functions[27]. Pisam and colleagues have described two subtypes

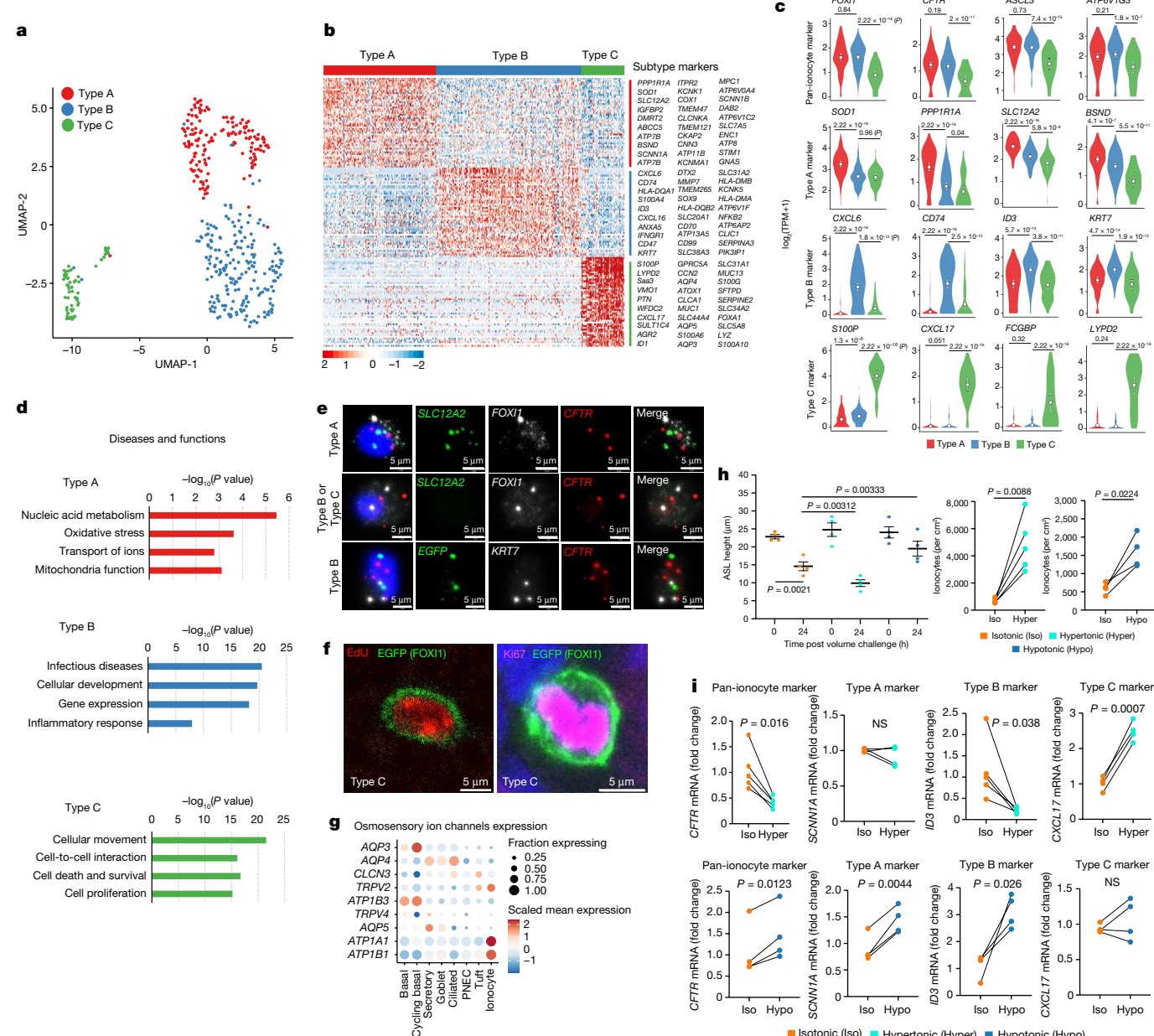

**Fig. 4 | Distinct subtypes of pulmonary ionocytes exist and respond to osmotic stress. a**, UMAP of 449 ionocytes coloured by subcluster. **b**, Type A, B and C ionocyte gene expression signatures showing relative expression (Z-score of $\log_2(\text{TPM} + 1)$). **c**, Distribution of expression levels ($\log_2(\text{TPM} + 1)$) for ionocyte subtype markers (white circle, mean; error bars, 95% confidence interval; $n = 449$ ionocytes from 8 donors; P value, Wilcoxon). **d**, Ingenuity pathway analysis of differentially expressed ionocyte subtype genes showing top significantly associated diseases and functional pathways (right-tailed Fisher's exact test). **e**, RNAscope validation of ionocyte subtypes using cytospun samples from ALI cultures. Representative image of $n = 36$ ionocytes. **f**, Differentiated *FOXI1*-Cre[ERT2]::ROSA-TG ALI cultures induced with OH-Tam and later pulsed-labelled with EdU. Cultures were then stained for Ki67 or EdU. Representative image of $n = 20$ ionocytes. **g**, Osmosensory ion and water channel gene expression in different cell types. **h**, Hyperosmotic and hypoosmotic conditions induce opposing forces on ASL hydration. Left, differentiated ALI cultures were exposed to hyperosmotic (+77 mOsm) and hypoosmotic (−77 mOsm) basolateral media for 24 h and ASL height was measured at 0 and 24 h following small volume addition to the apical surface. Data are mean ± s.e.m. Statistical significance by paired two-tailed Student's *t*-test ($n = 4$ donors, each using 2 cultures). Right, quantification of ionocyte numbers in 21 days ALI cultures maintained under hyperosmotic and hypoosmotic conditions throughout basal cell differentiation. Data are mean ± s.e.m. Statistical significance by paired two-tailed Student's *t*-test (hyperosmotic: $n = 5$ donors, 1 culture per donor, quantified as *FOXI1*-Cre[ERT2] EGFP⁺ cells; hypoosmotic: $n = 4$ donors, each using 2–3 cultures, quantified as ATP6V1G3⁺ cells). **i**, Changes in subtype marker gene expression by RT–qPCR in ALI cultures maintained under hypoosmotic or hyperosmotic stress as in **h**. mRNA fold change was calculated by the $\Delta\Delta C_t$ method. Statistical significance was determined by paired two-tailed Student's *t*-test ($n = 4$ donors, each using 3 cultures). TPM, transcripts per million.

of mitochondrial-rich cells called 'α-chloride cells' and 'β-chloride cells' in freshwater trout with electron-dense and -light appearance[28]. Mammalian kidney intercalated cells are functionally related to ionocytes, and intercalated cells have three described subtypes, type A (A-IC), type B (B-IC) and nonA-nonB. To evaluate whether subtypes of pulmonary ionocytes also exist, we re-clustered pulmonary ionocyte transcriptional signatures. Ionocytes partitioned into three clusters (Fig. 4a) defined by distinct transcriptional programs (Fig. 4b,c).

Type A pulmonary ionocytes contained 127 upregulated genes (adjusted $P < 0.05$, $\log_2$-transformed fold change > 0.50), including the Na-K-2Cl cotransporter-1 (NKCC1; encoded by *SLC12A2*), the amiloride sensitive epithelial sodium channel (ENaC) α and γ subunits (*SCNN1A* and *SCNN1G*), the CLC voltage gated chloride channel CLC-Ka (*CLCNKA*) and the Barttin CLCNK type accessory beta subunit (*BSND*) required for CLC-K channel activation (Supplementary Table 3). Type A ionocytes also shared expression of the tilapia gill type III ionocyte marker *SLC12A2* (ref. 27), which we confirmed with RNAscope (Fig. 4e), and several other markers of type III ionocytes were also expressed in type A pulmonary ionocytes (Supplementary Table 3). We also observed two phenotypes of ionocytes expressing BSND and/or FOXI1 in human ALI cultures (Extended Data Fig. 10b). Ingenuity pathway analysis (IPA) of the differentially upregulated genes in type A ionocytes revealed associated pathways including oxidative phosphorylation, mitochondrial function and tricarboxylic acid (TCA) cycle signalling pathways (Fig. 4d, Extended Data Fig. 10a and Supplementary Table 4). Each of these pathways is closely associated with mitochondrial energy production, which is consistent with type A pulmonary ionocytes having similar biology to mitochondrial-rich cells in other species. The activation of the aldosterone signalling pathway in type A ionocytes (Extended Data Fig. 10a) is notable, given its known role in increasing ENaCα expression and proteolytic processing of the ENaCγ subunit required for channel activation[29]. These features of type A pulmonary ionocytes align with known ion/osmoregulatory roles of ionocytes in other systems[10].

Type B pulmonary ionocytes contained 165 upregulated genes (adjusted $P < 0.05$, $\log_2$-transformed fold change > 0.5) (Supplementary Table 3), including *KRT7*, proton pumps genes *ATP6AP2* and *ATP6V1F*, a chemotactic gene *CXCL6* that recruits neutrophils, and MHC II antigen presentation genes *CD74*, *HLA-A*, *HLA-DQA1* and *HLA-DR1B* (Fig. 4b,c). RNAscope confirmed co-expression of *KRT7*, *CFTR* and *EGFP* in *FOXI1*-Cre^ERT2^ lineage-labelled type B ionocytes (Fig. 4e). The type B ionocyte was also characterized by a unique set of enriched genes functionally relevant to inflammatory response, IL-8 signalling and infectious disease pathways (Fig. 4d, Extended Data Fig. 10a and Supplementary Table 5).

Type C pulmonary ionocytes contained 500 upregulated genes (adjusted $P < 0.05$, $\log_2$-transformed fold change > 0.5) (Fig. 4a–c and Supplementary Table 3) having biological functions closely associated with cellular movement and cell proliferation (Fig. 4d, Extended Data Fig. 10a and Supplementary Table 6). Pulmonary ionocytes have been thought to be terminally differentiated[1]; however, we found that a small subset of *FOXI1*-Cre^ERT2^ traced ionocytes in ALI cultures were EGFP⁺Ki67⁺ and also incorporated the 5-ethynyl-2′-deoxyuridine (EdU) nucleotide after tracing (Fig. 4f). Further supporting enriched genes with biologic functions in cellular movement pathways, a subset of ionocytes was observed to have highly dynamic appendages under time-lapse live ionocyte imaging of lineage-traced cultures (Supplementary Video 6). The highly expressed *CXCL17* gene in type C ionocytes has potent antimicrobial activity and also functions as a chemoattractant to recruit immature dendritic cells and monocytes to the lung[30], whereas enrichment in aquaporin gene expression (*AQP3*, *AQP4* and *AQP5*) suggests this subtype may play a unique role in water transport (Fig. 4b,c and Supplementary Table 3). Notably, these same aquaporins were significantly upregulated in basal and luminal cell types of *FOXI1*-KO epithelia (Extended Data Fig. 6h and Supplementary Table 2), suggesting compensatory expression in the absence of ionocytes.

### Osmotic stress alters ionocyte phenotype

Osmoregulation, which maintains the osmolarity of fluid surrounding cells, is a key feature of ionocytes across species and extensively studied in fish adaptation to environmental changes in salinity[10,31]. In the mammalian kidney, osmoregulation is coordinated by principal

cells and intercalated cells of the collecting ducts, which control water movement, acid–base regulation, and Na⁺, Cl⁻, K⁺ and Ca²⁺ homeostasis[32]. Similar processes are thought to be important in maintaining the osmolality of ASL, which at homeostasis in the mouse trachea is isosmotic (330 ± 36 mOsm) with serum, but under evaporative stress can increase to greater than 400 mOsm (ref. 33).

Given that ionocytes expressed high levels of several osmosensory channel genes (*ATP1A1*, *ATP1B1* and *TRPV2*) (Fig. 4g), we proposed that pulmonary ionocytes may also participate in osmoregulation by airway epithelia. To investigate this possibility, we exposed fully differentiated ferret ALI cultures to slightly hypertonic or hypotonic basolateral media (±77 mOsm l⁻¹ NaCl) and observed ASL heights that were significantly lower ($P = 0.00312$) or higher ($P = 0.00333$), respectively, than isosmotic conditions (Fig. 4h). Continuous exposure of actively differentiating basal cells at ALI to hypertonic and hypotonic conditions significantly increased (hypertonic: 7.3-fold, $P = 0.0088$; hypotonic: 2.7-fold, $P = 0.0224$) the number of ionocytes at full differentiation (21 days), compared with isosmotic controls (Fig. 4h and Extended Data Fig. 8a,b). Hyperosmotic stress decreased the expression of *CFTR* and the type B ionocyte marker *ID3*, whereas it increased the expression of the type C ionocyte marker *CXCL17* (Fig. 4i). By contrast, hypoosmotic stress increased the expression of *CFTR*, type B ionocyte marker *ID3* and type A ionocyte marker *SCNN1A* (ENaC) (Fig. 4i). The observed changes in *CFTR* expression under hyperosmotic or hypoosmotic stress (Fig. 4i) are consistent with the changes in abundance of type C and/or type B ionocytes, which have significantly different ($P = 1.8 \times 10^{-7}$) levels of *CFTR* expression (Fig. 4c).

### Interspecies comparisons of rare cell types

To better understand functional differences between ferret and mouse pulmonary ionocytes, we performed an interspecies comparison of human, ferret and mouse rare cell transcriptomes using publicly available scRNA-seq datasets (Fig. 5 and Extended Data Fig. 10h). These studies demonstrated that the human pulmonary ionocyte transcriptional signature was significantly more similar to ferret than to mouse ionocytes (Fig. 5i). This similarity extended to human and ferret ion channels including *BSND*, *ATP6V1G3* and five other channels not expressed in mouse ionocytes (Fig. 5g). By contrast, tuft cells and PNECs from ferret and mouse were equally divergent at the transcriptional level from their human counterparts (Fig. 5i).

Uniform manifold approximation and projection (UMAP) clusters of rare cell type from human, ferret and mouse revealed a new population of cells (Fig. 5a,b) with overlapping PNEC, ionocyte and tuft cell transcriptional signatures. This unique population contained a subset of cells expressing lower levels of tuft (*TRPM5*), PNEC (*CHGA*) and ionocyte (*FOXI1*) marker genes (Fig. 5c–e) and thus seemed to be a rare cell progenitor population at an intermediate stage of differentiation. This finding is somewhat consistent with previous work that proposes tuft cells are the precursors of ionocytes and PNECs[25]. Top differentially expressed genes in this rare cell progenitor, compared with fully differentiated transcriptomes of tuft cells, PNECs and ionocytes, included *SCGB3A2*, *CXCL17*, *KRT4* and *HES1*, among others (Fig. 5h and Supplementary Table 11), and its most specific transcription factor was *FOXQ1* (Extended Data Fig. 12d). Furthermore, a small subset of cells in this rare cell progenitor cluster expressed *FOXI1* (Fig. 5e) and also contained a small fraction of ferret EGFP⁺ *FOXI1*-Cre^ERT2^ lineage-labelled cells (Fig. 5f), consistent with in vivo and in vitro *FOXI1*-Cre lineage tracing of the three rare cell types during airway development or basal cell differentiation, respectively.

We next isolated the 1,497 rare cell types (PNECs, tuft cells, ionocytes and rare cell progenitors) from our scRNA-seq profiling of ferret airway epithelial cells and fitted the partition-based graph abstraction (PAGA) algorithm[34] (Extended Data Fig. 12a–c). The low-dimensional PAGA embedding recapitulated the expected cellular topology, with

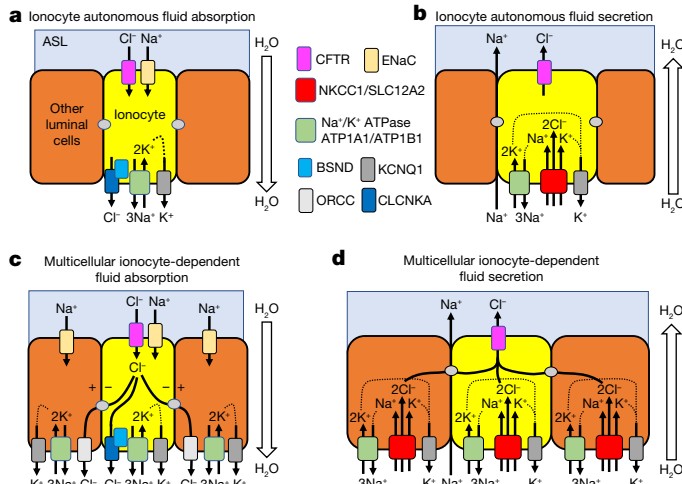

**a** Ionocyte autonomous fluid absorption

**b** Ionocyte autonomous fluid secretion

**c** Multicellular ionocyte-dependent fluid absorption

**d** Multicellular ionocyte-dependent fluid secretion

**Fig. 6 | Models for pulmonary ionocyte anion transport function in a multicellular airway epithelium. a,b,** Pulmonary ionocytes (yellow) function in a cell-autonomous manner to facilitate anion movement across airway epithelia required for fluid absorption (**a**) and fluid secretion (**b**). **c,d,** Multicellular ionocyte-dependent anion movement utilizing electric coupling through gap junctions to facilitate fluid absorption (**c**) and fluid secretion (**d**). The second models propose that Na⁺ and K⁺ electrical driving forces in cells coupled to ionocytes collectively drive Cl⁻ absorption and secretion through CFTR in pulmonary ionocytes. In both models, the ionocyte channels shown were differentially enriched in the pulmonary ionocyte transcriptome. ORCC, outward rectifying Cl⁻ channel.

putative progenitor cells linked with each of the mature cell-type clusters (Extended Data Fig. 12d). We fitted an elastic principal graph[35] to the data, which identified a branching trajectory consistent with interpretation of the progenitor cluster as the precursor for all three of the mature rare types (Extended Data Fig. 12c). Our trajectory inference described above provided a framework to assess the relationship of ionocyte subtypes. By projecting the cells in each ionocyte subcluster onto the principal graph topology (Extended Data Fig. 12a,b), we observed that the type C ionocytes are most similar to the rare cell progenitor, which would be consistent with this subtype giving rise to both type A and B ionocytes. Supporting this notion are our data (Fig. 4f) demonstrating that type C ionocytes can replicate and have a proliferative transcriptomic signature (Fig. 4d). Type C ionocytes were also the least abundant of the ionocyte subtypes, which is consistent with a progenitor cell state.

## Conclusions

Here we applied conditional genetics and fate mapping in ferrets to dissect the biology and function of pulmonary ionocytes. Functional studies and the channelome of pulmonary ionocytes support cell-autonomous movement of Cl⁻ and HCO₃⁻ to regulate ASL volume, pH, viscosity and airway clearance (Fig. 6 and Supplementary discussion). Notably, ionocytes maintain homeostatic ASL height of the airway epithelium by controlling both absorption and secretion. Single-cell transcriptomic profiling of lineage-traced ionocytes using pulse-seq revealed previously unknown diversity, defining three ionocyte subtypes and a common rare cell progenitor during airway development that specifies ionocytes, tuft cells and PNECs. Type C ionocytes seem most closely related to this common rare cell progenitor and thus seem to be a committed progenitor of type A and B ionocytes.

Collectively, these studies provide a path for using precision genome editing in ferrets to study the function of evolutionarily divergent cell types between humans and rodents and a proximal airway single-cell

transcriptional atlas of utility for studying lung disease in ferret models. The genetic approaches applied here in the ferret can be used to study individual cell-type contributions to genetic disease states in any organ and the stem cell compartments that mediate tissue repair.

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

## Methods

### Generation and use of transgenic ferret models

All animal experimentation was approved by the Institutional Animal Care and Use Committee of the University of Iowa. Existing transgenic models used in these studies included *CFTR*-KO ferrets[3,36] and *CFTR*[G551D/G551D] ferrets[4,37]. Animals were distributed into experimental groups based on genotype and were not randomized. Blinding was not necessary in this study since assays used unbiased quantification methods. Sex was not considered a variable due to the difficulty in obtaining an equal distribution of genders when using transgenic ferrets. No statistical methods were used to predetermine sample size. Ferrets were outbred on a sable coat colour background. The age of animals tested is given in the Reporting Summary.

**Zygote manipulation and adoptive transfer of embryos.** Ferret zygotes were collected as previously described from sable ferret matings[38]. Cas9 ribonucleoprotein complex, at a final concentration of 400 ng μl$^{-1}$, was injected into zygote pronuclei (around 3–5 pl) using a FemtoJet (Eppendorf). When a DNA template was used to facilitate homologous recombination, linear DNA was added at a final concentration of 40 ng μl$^{-1}$. To assemble Cas9/sgRNA ribonucleoproteins, 1 μM sgRNA was incubated with 1 μM Cas9 protein in IDT Duplex Buffer for 30 min at room temperature. Injected embryos were cultured in TCM-199 + 10% FCS medium overnight to the two-cell stage before being transferred into primipara pseudopregnant jills[39]. The kits were naturally delivered 42 days after adoptive transfer (full-term gestation).

**Generation of sgRNA and DNA templates for homologous recombination in ferret zygotes.** The sgRNAs targeting *FOXI1* exon-1, *FOXI1* 3′ UTR and *CFTR* exon-16 were designed using the Broad Institute GPP and/or CRISPOR tool. All sgRNAs used in this project are listed in Table 1. sgRNAs were generated from in vitro transcribed gBlocks synthesized by Integrated DNA Technologies and included all components necessary for sgRNA production (that is, T7 promoter, target sequence, guide RNA scaffold and termination signal). gBlocks were PCR amplified using primers gRNA-fwd and gRNA-rev (Table 1). The T7-sgRNA PCR products were gel purified and used as the template for in vitro transcription using MEGAshortscript T7 kit (Ambion). All sgRNAs were then purified using MegaClear Kit (Ambion) and eluted in RNase-free water.

*FOXI1* gene homology arms for insertion of the IRES-Cre[ERT2] cassette were generated as gBlocks with unique restriction sites and cloned into a plasmid flanking the IRES-Cre[ERT2]. Similarly, *CFTR* exon-16 homology arms containing *loxP* sites were synthesized as gBlocks and cloned into a plasmid flanking exon-16. In both cases, silent blocking mutations were introduced into the sgRNA binding sequence to prevent cleavage of the donor fragment following HDR. sgRNA sequences, genomic target sites and length of homology arms with genomic positions are listed in Table 1 for each of the models created.

**Molecular characterization of transgenic ferret models.** Genomic DNA was generated from tail clips as previously described[4] and used for initial genotyping of founders by PCR. Putative *FOXI1*-KO founder DNA was screened using a set of primers that resided external to the sgRNA cut sites, whereas *FOXI1*-Cre[ERT2] and *CFTR*[L/L] candidate founders were screened using two sets of primers to capture genomic flanking sequences outside the right and left homology arms (that is, one primer internal and another primer external to the homology arm) (Supplementary Table 7). To confirm integrity of the targeted sequences in candidate founders, two further assays were performed: (1) the entire region of the donor fragment and flanking genomic sequences were subjected to nested PCR and the products were Sanger sequenced; and (2) Southern blotting was performed using restriction enzyme digests and locus-specific probes that mapped the length of endogenous and transgene-derived fragments for both arms flanking the insertion. A Cre probe was also used to map the *FOXI1*-Cre[ERT2] locus. Finally, primary fibroblasts were generated from *CFTR*[L/L] and wild-type ferrets and treated with TAT-CRE (or untreated) to induce deletion of the 482 base pairs (bp) *CFTR* exon-16 fragment. Deletion was confirmed by PCR of the region and analysis of PCR products on agarose gels: wild-type (1,230 bp), intact *loxP* allele (1,298 bp) and deleted *loxP* allele (816 bp).

**Husbandry and specialized care of *FOXI1*-KO ferrets.** *FOXI1*-KO kits are very fragile and show a head tilt/circling phenotype shortly after birth that can affect their ability to feed. *FOXI1*-KO ferrets have balance disturbances early in life, probably due to a lack of pendrin (SLC26A4) expression in the inner ear and the expansion of the endolymphatic compartment. Impaired kidney function due to cyst formation also affected overall health. A rearing protocol was developed to increase *FOXI1*-KO kit survivability. From birth, *FOXI1*-KO kits were weighed every 24 h. Those animals that demonstrated a decline in weight gain were supplemented with oral Elecare baby formula (Abbott Laboratories) every 4–6 h. The frequency and amount of Elecare hand fed was adjusted on the basis of the weight and size of the developing kit and veterinarian directed. When kits demonstrated the ability to eat solid food (about 5 weeks of age), they were also offered a canned supplemental diet (Purina) in addition to hand feeding. After weaning at 8 weeks of age, they were transitioned onto a solid food diet consisting of solid chow (Marshall Farms), a hydrated solid chow mash and canned cat food (Purina).

### Lineage tracing pulmonary ionocytes in *FOXI1*-Cre[ERT2]::ROSA-TG ferrets and ALI cultures

*FOXI1*-Cre[ERT2] ferrets were crossed to ROSA-TG Cre reporter ferrets[19] to obtain hemizygous offspring for lineage tracing. In vivo *FOXI1*-Cre[ERT2] lineage tracing was performed by five sequential daily intraperitoneal injections of tamoxifen (20 mg kg$^{-1}$) to 1-month-old and 5-month-old adult ferrets. At 7–10 days later, tissues were fixed in 4% PFA for 24 h, cryoprotected in sucrose and embedded into OCT block or processed for whole-mount staining. Cryosections (8 μm) were used for immunofluorescence and mounted with DAPI solution for confocal imaging. For lineage tracing in vitro, ALI cultures derived from *FOXI1*-Cre[ERT2]::ROSA-TG ferret basal cell cultures were treated with 2 μM 4-hydroxytamoxifen (OH-Tam) in ethanol or vehicle alone (ethanol) using two different experimental procedures: (1) For scRNA-seq experiments, cultures were treated with OH-Tam during differentiation at ALI from day 1 to day 17 (medium change every other day) and cells were collected at 21 days of ALI. This was done because it gave rise to a greater number of EGFP$^+$ cells in the cultures following full differentiation. (2) For functional halide transport studies using the YFP sensor, ALI cultures were treated with OH-Tam starting on day 14 after moving to an ALI and cultures were moved to a new plate in the absence of OH-Tam 2 days before functional imaging studies (on days 22 to 28 of ALI).

### Immunofluorescence and antibodies

Paraffin sections (5 μm) and cryosections (8 μm) were treated for epitope retrieval with 10 mM citrate buffer at 95 °C for 20 min and permeabilized with 0.1% Triton X-100 in PBS. Sections were then blocked in 10% donkey serum/PBS for 1 h at room temperature and primary antibodies were applied and incubated at 4 °C overnight. Slides were then washed three times for 15 min and incubated with secondary antibodies for 1 h at room temperature. Slides were washed and counterstained with DAPI for confocal imaging. The following antibodies were used: anti-BSND (1/500; ab196017, Abcam), anti-FOXI1 (1/500; ab20454, Abcam), anti-Keratin 5 (1/500; 905501, Biolegend), anti-SYP (1/200; sc-17750, Santa Cruz Biotechnology), anti-acetylated Tubulin (1/1,000, T7451, Sigma-Aldrich), anti-ATP6V1G3 (1/500, HPA028701, Sigma-Aldrich), anti-NKA (1/100, a5, DHSB, UIOWA), anti-Ki-67 (1/500, 14-5698-82, eBioscience), anti-EGFP (1/300, ab13970, Abcam),

anti-CFTR (1/100 to 1/300, CFTR antibody 596, cftrantibodies.web.unc. edu), anti-TRPM5 (1/300, ACC-045, Alamone), anti-p63 (1/300, clone poly6190, STEMCELL Technologies), anti-Muc5B (1/300, HPA008246, Sigma), anti-Muc5AC (1/300, ab3649, Abcam), Alexa Fluor 647 donkey anti-mouse IgG (1/250, A31571, Molecular Probes), Alexa Fluor 488 donkey anti-goat IgG (1/250, A11055, Invitrogen), Alexa Fluor 488 donkey anti-chicken IgG (1/250, 703-546-155, Jackson ImmunoResearch), Alexa Fluor 488 donkey anti-rabbit IgG (1/250, A21206, Invitrogen), Alexa Fluor 568 donkey anti-goat IgG (1/250, A-11057, Jackson ImmunoResearch), Alexa Fluor 647 donkey Anti-Rabbit IgG (1/250, 711-606-152, Jackson ImmunoResearch), Alexa Fluor 555 donkey Anti-mouse IgG (1/250, A31570, Life Technologies). EdU staining was performed according to the manufacturer's instructions (C10340, Click-iT EdU Cell Proliferation Kit for Imaging, Alexa Fluor 647 Dye).

## Whole-mount ferret trachea immunofluorescence staining for ionocytes, tuft cells and PNECs
Ferret tracheae and dissected intralobar airways were fixed overnight in 4% paraformaldehyde, then washed three times in PBS for 30 min each. Fixed ferret tracheae could then be placed in 70% ethanol at −20 °C for extended storage. Staining for ionocyte markers used CFTR (cftrantibodies.web.unc.edu, CFTR antibody 596, working dilutions: 1:100 to 1:300), ATP6V1G3 (Millipore Sigma, HPA028701, working dilution: 1:300) and NKA (dshb.biology.uiowa.edu, ATP1A1, a5, working dilutions: 1:100 to 1:300). Staining for tuft cell marker used TRPM5 (Alamone, ACC-045, working dilution: 1:300) and for PNEC marker used SYP/Synaptophysin (Santa Cruz Biotechnology, sc-17750, working dilution: 1:100). Samples required antigen retrieval in citrate buffer (10 mM sodium citrate, 0.05% Tween 20, pH 6.0) at 55 °C overnight, with agitation, before immunostaining. Ferret tracheae and intralobar airways were then washed with PBS three times for 20 min each and incubated in blocking buffer (20% donkey serum, 0.1% Triton X-100 and 1 mM $CaCl_2$ dissolved in PBS) overnight at 37 °C. Ferret tissues were then incubated with primary antibodies (dissolved in diluent buffer: 1% donkey serum, 0.1% Triton X-100 and 1 mM $CaCl_2$ dissolved in PBS) for 3 days at 37 °C, with agitation. Tracheae were then washed three times for 30 min in PBS and incubated with appropriate secondary antibodies for 2 days at 37 °C, with agitation. After secondary antibody incubation, samples were washed three times for 30 min each in PBS and transferred to Ce3D tissue clearing solution (Biolegend catalogue no. 427704) for 2–3 h at room temperature. After tissue clearing, samples were mounted onto microscope slides (Fisherbrand Superfrost Plus) under 0.33 mm coverslips and edges were sealed with Gorilla Glue and clamped with binder clips for 30 min to ensure glue fixation. Zeiss LSM 880 or 980 confocal microscopes were used for imaging acquisition.

## In vivo MCC measurements
MCC measurements in wild-type, *FOXI1*-KO, and *FOXI1*-Cre[ERT2]::*CFTR*[L/L] ferrets were performed using positron emission tomography and computed tomography (PET/CT) and [68]Ga-macro aggregated albumin ([68]Ga-MAA) as previously described with modifications[40]. *CFTR*[G551D/G551D] ferrets[4] reared on or off the CFTR modulator (VX-770/ivacaftor) served as controls for CFTR-dependent MCC. Ferrets were anaesthetized with ketamine/xylazine and then intubated. After intubation, the animal was placed in the gantry of a PET/CT scanner (GE Discovery MI, GE Healthcare). An initial scout computed tomography was acquired to confirm placement of the endotracheal tube at the distal end of the trachea. The dynamic positron emission tomography acquisition (15 min) was initiated and during the first minute of image acquisition 50 µl of saline containing 50 µCi [68]Ga-Macro Aggregated Albumin (about 1.85 MBq) and 600 µM methacholine was rapidly administered into the distal trachea through a catheter. The syringe, catheter and endotracheal tube were then removed, and images were acquired continuously for 15 min. The positron emission tomography acquisition was followed by computed tomography for attenuation

correction and anatomical coregistration. List mode data were reconstructed with the GE Discovery scanner's software using three methods: (1) static image of 15 min; (2) dynamic image with 15 ×1 min frames; (3) dynamic image with a 20 s delay to eliminate delay before dose administration, followed by 60 ×10 s frames. We performed data analysis using PMOD software v.4.2 (PMOD Technologies) and clearance of the [68]Ga-MAA was quantified as the PET/CT volume of interest at minute intervals after tracer deposition. Data were normalized to the volume of interest of the first full minute after tracer deposition. A plateau of clearance was typically reached by 10.5 min in wild-type animals. This timepoint was used to calculate percentage clearance.

## Ferret tracheal basal cell isolation, expansion and differentiation
Ferret tracheal airway basal stem cells were isolated using an enzymatic digestion method similar to previous reports[41]. All primary cells tested negative for mycoplasma contamination. The cells were cultured in PneumaCult-Ex Plus medium (STEMCELL Technologies) on plastic plates precoated with laminin-enriched 804G-conditioned medium. For passaging, the cells were detached with Accutase (STEMCELL Technologies) and re-seeded at a 1:4 split on 804G-coated plates as previously described[42]. For differentiation at ALI, ferret basal cells were seeded onto Transwell membranes coated with 804G in PneumaCult-Ex Plus medium for 24 h and then lifted to an ALI with PneumaCult-ALI medium (STEMCELL Technologies) placed only on the basal side of the Transwell. Cultures were then used for experiments at 21–28 days.

## Droplet-based scRNA-seq
Fully differentiated ferret airway epithelia ALI cultures were dissociated using Accumax (STEMCELL Technologies) followed by DNase treatment. Cells were filtered through a 20 µM strainer and pelleted in 0.04% BSA PBS at 500*g* for 10 min. Non-viable dead cells were removed by using MACS Dead Cell Removal Kit following 10X Genomics recommendations (Document CG00039). Single cells were counted on a Thermo Countess cell counter and 0.04% BSA/PBS was added to achieve a targeted concentration of 1,000 cells per microlitre. Ionocyte enrichment was performed on OH-Tam-treated *FOXI1*-Cre[ERT2]::ROSA-TG ALI cultures followed by FACS isolation of EGFP- and Tomato-positive cells. Airway epithelial cells were sorted using a Becton Dickinson Aria instrument and BD FACSDiva 8.0.1 software. Cells were identified on the basis of FSC and SSC gating (Supplementary Fig. 2). Tomato- and EGFP-positive epithelial cells were identified on the basis of comparison with non-reporter ferret airway basal cells. Single cells were identified on the basis of forward scatter and forward pulse width. FACS-isolated EGFP-positive cells were then mixed with Tomato-positive epithelial cells to achieve around 10,000 total cells for 10X sequencing. The ratio of EGFP- to Tomato-positive cells varied for each experiment depending on the yield of lineage-labelled cells. Sequencing libraries were generated by following 10X Genomics recommendations (Document CG000315). Briefly, single cells and reverse transcription master mix were partitioned into Gel Beads in partitioning oil in the 10X Chromium controller. After reverse transcription, complementary DNA libraries were amplified and fragmented, followed by adaptor ligation and sample index PCR reaction. Libraries were sequenced on the NovaSeq 6000 platform by the University of Iowa Genomics Division.

## Short circuit current measurements of CFTR-mediated Cl[−] and HCO$_3^-$ transport in ALI cultures
Short circuit current (Isc) measurements were made using an epithelial voltage clamp and an adapted Ussing chamber system (Physiologic Instruments). Symmetrical buffer systems were used for measuring both the chloride and bicarbonate currents. The chloride buffer consisted of 135 mM NaCl, 2.4 mM $K_2HPO_4$, 0.6 mM $KH_2PO_4$, 1.2 mM $CaCl_2$, 1.2 mM $MgCl_2$, 10 mM dextrose and 5 mM HEPES (pH 7.4) gassed with air at 37 °C. The bicarbonate buffer consisted of 118.9 mM sodium gluconate, 25 mM $NaHCO_3$, 2.4 mM K$_2$HPO4, 0.6 mM $KH_2PO_4$, 5 mM

calcium gluconate, 1 mM magnesium gluconate and 5 mM dextrose (pH 7.4) gassed with 5% $CO_2$ at 37 °C. The following chemicals were sequentially added to the apical chamber: (1) 100 µM amiloride (to inhibit ENaC); (2) 100 µM DIDS (to inhibit non-CFTR anion channels); (3) 100 µM IBMX and 10 µM forskolin (cAMP agonists that stimulate CFTR); and (4) 10 µM GlyH101 (to block CFTR). The difference in the average plateau measurement for the Isc from 45 s before to 45 s after each stimulation was calculated and represented as the change of Isc ($\Delta$Isc) in response to the corresponding drug[12]. Data were collected using the software Acquire and Analyze v.2.3.

## Whole-mount ferret tracheal tissue electrophysiology
An Ussing chamber system (Physiologic Instruments) was used for measuring the electrophysiological properties of intact ferret tracheae. Ferret tracheae were maintained under warm F-12 medium (Gibco) during dissection. Briefly, ferret proximal tracheae were collected and mounted on pins in a 'slider' (P2304 slide) that fits between two halves of the chamber, being careful to only handle the edges of tissue. The tissue was then assembled into the Ussing chamber and secured by the pressure clamps. Symmetrical chloride buffer (135 mM NaCl, 2.4 mM $K_2HPO_4$, 0.6 mM $KH_2PO_4$, 1.2 mM $CaCl_2$, 1.2 mM $MgCl_2$, 10 mM dextrose and 5 mM HEPES, pH 7.4) was used and samples were maintained at 37 °C and bubbled with air. The following chemicals were sequentially added to the apical chamber: (1) 100 µM amiloride (to inhibit ENaC); (2) 100 µM DIDS (to inhibit non-CFTR anion channels); (3) 100 µM IBMX and 10 µM forskolin (cAMP agonists that stimulate CFTR); and (4) 10 µM GlyH101 (to block CFTR). Data were collected using the software Acquire and Analyze v.2.3.

## mRNA quantification using RT–qPCR
TaqMan Real-Time PCR was used for quantification of mRNA. All primers and probes were synthesized by Integrated DNA Technologies and primer sequences are provided in Supplementary Table 8. Total RNA isolation was performed using the RNeasy Plus miniKit (Qiagen) and RNA concentration was measured using a Nanodrop or the Qubit assay. cDNA was then synthesized using a High-Capacity cDNA reverse transcription kit (Applied Biosystems).

## Measurements of ASL height, pH, viscosity and fluid absorption rates in differentiated ferret ALI cultures
**Fluid absorption rates and ASL height measurements.** Fully differentiated ALI cultures were derived from wild-type, *CFTR*-KO (ref. 36) and *FOXI1*-KO primary tracheal basal cells. ASL height and fluid absorption rates were evaluated as previously described[12]. Briefly, excess mucus was removed from the apical surface of ALI cultures by washing in an excess of PBS and then they were equilibrated for about 16 h in a humidified, 5% $CO_2$, 37 °C incubator before initiating the experiment. Then, 10,000 Da Texas red–dextran dye was applied (in 18 µl of PBS) to the apical surface of ALI cultures. *XZ* (line) images were then taken at five different rotational axes around the centre of the culture, immediately after dye was added and then again at various time points up to 24 h. While imaging on the confocal, the chamber remained humidified in a 5% $CO_2$ atmosphere at 37 °C. For absorption studies in the first hour after fluid addition, the chamber was not moved and one location was imaged on five directional axes. For ASL height measurements at 4 and 24 h, five locations around the centre of the Transwell were imaged on five rotational axes for each sample. The mean ASL height and volume/unit area of the Texas red–dextran dye were calculated for each *XZ* scan, and then the 25 values for each measurement were averaged for each Transwell. Fluid absorption rates were calculated as previously described[12] by converting the ASL height into a uniform cylinder with defined volume and diameter. Graphs of the ASL volume versus time were then used to calculate the fluid absorption rates (nl min$^{-1}$ cm$^{-2}$) as the linear slope of the line generated during the first 20 min following fluid addition. Equilibrated ASL height was at 24 h.

**ASL pH measurements.** Fully differentiated ALI cultures were derived from wild-type, *CFTR*-KO (ref. 36) and *FOXI1*-KO primary tracheal basal cells. ASL pH was measured with slight modifications to that previously described[15]. In brief, the ratiometric pH indicator SNARF-conjugated dextran dye (ThermoFisher Scientific) was used to generate pH standard curves and directly measure apical pH on differentiated ALI cultures. ALI cultures were maintained in basolateral $HCO_3^-$-containing buffer and the microscope chamber was humidified in a 5% $CO_2$ atmosphere at 37 °C. SNARF-conjugated dextran powder was directly applied to the apical surface through a 5 µm mesh. A confocal microscope (Zeiss LSM 880) was used to excite the SNARF dye at 488 nm and measure fluorescence intensity at 580 nm and 640 nm from 6–8 areas of interest in each ALI culture. Fluorescence ratios were converted into pH values by using the standard curves as previously described[43].

**ASL viscosity measurements by fluorescence recovery after photobleaching.** Fully differentiated ALI cultures were derived from wild-type, *CFTR*-KO (ref. 36) and *FOXI1*-KO primary tracheal basal cells. ASL viscosity was measured as previously described with minor modifications[14]. In brief, the apical surface of the cultured epithelium was washed with PBS 48 h before fluorescence recovery after photobleaching (FRAP) was performed (19 days of differentiation at ALI). FITC-dextran powder (70 kDa, Sigma-Aldrich) was then directly applied to the apical surface of ALI cultures using a 100 µm mesh. After 2 h, epithelial ASL viscosity was measured in a humidified chamber in a 5% $CO_2$ atmosphere at 37 °C using a confocal microscope (Zeiss 880). An $8 \times 8$ µm$^2$ square region was photobleached by increasing the 488 nm laser intensity to 100%. Images were then acquired until maximal recovery was reached. Six to nine regions were selected for recovery curves from different locations in each Transwell. The fluorescence recovery half time ($t_{1/2}$) was determined using Zeiss software FRAP.

## Single-cell measurements of anion movement through pulmonary ionocytes
A previously described halide-sensitive YFP-H148Q/I152L/F46L cDNA[20] was used to replace EGFP in the pLenti-LoxP-sdRED-LoxP-EGFP plasmid (Addgene) for the generation of lentivirus (Extended Data Fig. 2c). *FOXI1*-Cre$^{ERT2}$ and *FOXI1*-Cre$^{ERT2}$::*CFTR*$^{L/L}$ ferret tracheal basal cells were virally transduced with Lenti-LoxP-dsRED-LoxP-YFP and selected by FACS for dsRED$^+$ transduced cells. ALI cultures were established using these basal cells and then treated with 2 µM OH-Tam (Sigma-Aldrich) starting on day 14 and terminating 2 days before imaging. YFP-labelled ionocytes were then used for imaging studies at days 22–28 (Extended Data Fig. 2d). Anion transport measurements were adapted from previously described methods using this YFP sensor[20] and used to study both anion absorption (apical→basolateral movement of iodide) and secretion (basolateral→apical movement of iodide). For all experiments, the microscope chamber was maintained in a 5% $CO_2$ environment at 37 °C with or without humidification. Anion absorption studies: The basolateral side of ALI cultures was immersed in PBS (137 mM NaCl, 2.7 mM KCl, 0.7 mM $CaCl_2$, 1.1 mM $MgCl_2$, 1.5 mM $KH_2PO_4$, 8.1 mM $Na_2HPO_4$, pH 7.4) while maintaining a humidified apical ALI, and baseline fluorescence intensity measurements were obtained. Cl$^-$ to I$^-$ exchange was then initiated by adding 18 µl of I$^-$ PBS buffer (137 mM NaI, 2.7 mM KCl, 0.7 mM $CaCl_2$, 1.1 mM $MgCl_2$, 1.5 mM $KH_2PO_4$, 8.1 mM $Na_2HPO_4$, pH 7.4) containing 100 µM IBMX/10 µM forskolin onto the apical surface of the culture as for fluid absorption studies. Similar conditions were used with the addition of 10 µM GlyH101 (to inhibit CFTR) or the exchange of Na-gluconate for Cl$^-$ to the apical 18 µl of PBS buffer. Ionocyte fluorescence intensity was obtained continuously using a confocal microscope (Zeiss LSM 880) with HQ filter set (488 nm excitation, 514 nm emission). Anion secretion studies under humidified conditions: The basolateral side of ALI cultures was immersed in Cl$^-$ PBS and baseline measurements were obtained. Cl$^-$ to I$^-$ exchange was then initiated by perfusing the basolateral side with

I⁻ PBS containing 100 μM IBMX/10 μM forskolin. These experiments were performed under humified conditions. Anion secretion studies with dehydrated ASL: Cultures were perfused with non-humidified 5% $CO_2$ for 20 min to dehydrate the ASL in the presence of basolateral Cl⁻ PBS and baseline measurements were obtained. Cl⁻ to I⁻ exchange was then initiated by perfusing the basolateral side with I⁻ PBS containing 100 μM IBMX/10 μM forskolin. Cl⁻ PBS (18 μl) was then added to the apical chamber in the absence or presence of basolateral bumetanide (100 μM) to block NKCC1. To assess the apical Cl⁻ dependence for basolateral I⁻ uptake by ionocytes, 18 μl of gluconate PBS (137 mM sodium gluconate, 2.7 mM KCl, 0.7 mM $CaCl_2$, 1.1 mM $MgCl_2$, 1.5 mM $KH_2PO_4$, 8.1 mM $Na_2HPO_4$, pH 7.4) was added to the apical surface in place of Cl⁻ PBS. Quantification of I⁻ transport was assessed as area over the curve of the YFP fluorescence intensity traces normalized to the starting YFP intensity before buffer exchange. To quantify the differences in ion transport between different conditions and genotypes, we fitted an area under the curve calculation using the 'pKNCA' package in R and modified this calculation to area over the curve. Modified R scripts can be obtained upon request.

### Ferret trachea μOCT imaging and quantitative analysis

The methods for μOCT and quantitative image analysis have been described previously in detail[44,45]. μOCT measurements were performed on wild-type and *FOXI1*-KO trachea tissue shipped overnight to the University of Alabama. In brief, the ASL depth, PCL depth, CBF and mucociliary transport rates were directly measured by μOCT without exogenous dyes. Real-time μOCT images were then processed for quantitative analysis. ASL depth and PCL depth were determined by geometric measurement in ImageJ. CBF was measured by Fourier analysis. Mucociliary transport rate was quantified by projecting a cross-sectional line through the mucus and using time elapsed over multiple layers. μOCT images were obtained at 5–8 randomly chosen locations on the mucosal surface of ferret proximal trachea.

### Identification of unannotated ferret genes using mouse and human orthologues

In the NCBI *Mustela putorius furo* Annotation Release 102, only 16,579 (59.4%) of all 27,912 genes are properly annotated with gene symbols. We used the NCBI Entrez database to identify a further 825 genes that had annotated gene names or aliases, increasing the number of labelled genes to 17,404 (62.4%). Next, single gene sequences of ferret reference genome (MusPutFur1.0) from the Ensemble database were aligned with the human (GRCh38.p13) and mouse (GRCm39) genomes using the 'msaClustalOmega' function from the multiple sequence alignment (msa) package in R. Gene names of sequence alignments with identity greater than 40% were adopted to label any ferret gene that was unannotated in the NCBI *Mustela putorius furo* Annotation Release 102. EggNOG-mapper was run using the diamond algorithm on the protein sequences in the current reference. Orthologues mapping to ferret, mouse and human were written to an output file. In cases for which multiple orthologues from one species mapped, the orthologue with the highest score was picked. To combine the multiple species comparison file run on the Ensemble genome with the file run on the NCBI genome, Ensemble protein IDs were added to the multiple species comparison file using the GTF file. Next, the two files were combined using 'merge' in R on the Ensemble protein ID. Annotating ferret genes with their orthologues in this manner enabled identification of a further 1,655 genes, taking the total to 19,059 (68.3%) (Supplementary Table 10). As a result of these changes, the median read assignment in the scRNA-seq studies was 71.3% (Extended Data Fig. 4a).

### Preprocessing of droplet (10X) scRNA-seq data

To generate a digital gene expression matrix, we first performed demultiplexing of the raw sequencing data. Subsequently, we conducted pseudo-alignment of these demultiplexed reads to a custom reference genome. This reference genome was assembled by combining sequences for reporter proteins EGFP and tdTomato with NCBI *Mustela putorius furo* annotation release 102. During the process, unannotated genes were partially renamed as described above. Pseudo-alignment and unique molecular identifier (UMI)-collapsing were performed using the Kallisto toolkit (v.0.48)[46]. We estimated the number of non-empty droplets using the KneePlot function from the 'DropletUtils' package, which detected a total of 94,664 cells. For each cell, we quantified the number of detected genes (with at least one UMI), and then excluded all cells with fewer than 2,000 genes detected, resulting in 77,099 high-quality cells from $n = 16$ ALI cultures from $n = 12$ donor ferrets. Expression values $E_{i,j}$ for gene $i$ in cell $j$ were calculated by dividing UMI count values for gene $i$ by the sum of the UMI counts in cell $j$, to normalize for differences in coverage, and then multiplying by 10,000 to create TPM-like values, and finally calculating $\log_2(\text{TPM} + 1)$ values, implemented using the NormalizeData function in the 'Seurat' R package. To merge all datasets together, batch correction was performed using the built-in data integration tool in Seurat v.3, using the 'IntegrateData' function[47]. The output was a corrected expression matrix, which was used as input for further analysis.

### Data visualization, dimensionality reduction and clustering

Highly variable genes were selected using a logistic regression fitted to the sample detection fraction, using the log of total number of UMIs for each gene as a predictor. Outliers from this curve are expressed in a lower fraction of samples than would be expected given the total number of reads mapping to that gene, that is, they are specific to a cell type, treatment, condition or state. The 2,000 most variable genes with greatest deviance were selected, both for analysis of the full dataset and for the subset of ionocytes. Principal component analysis was then computed using these variable genes, and scores for the top ten components were used to compute a nearest neighbour graph ($k = 20$) which was the input to clustering. To cluster single epithelial cells by their expression, we used the Louvain unsupervised clustering algorithm, as implemented with Seurat's 'FindClusters' function. We used a resolution parameter of $R = 1$ on the main dataset of 77,099 cells. Clusters were mapped to cell types using known marker genes on the basis of human and mouse tracheal epithelial subsets (Extended Data Fig. 4b). Pulmonary ionocytes were subclustered to examine possible heterogeneity of mature types (Fig. 4). For subclustering of pulmonary ionocytes, we use $R = 0.25$ for ionocyte subtypes and defined three groups, which we annotated as type A, B and C ionocytes.

### Differential expression and identification of cell-type markers

All differential expression testing was performed using a two-part 'hurdle' model to control for both technical quality and ferret-to-ferret variation, implemented using the R package MAST[48], and likelihood-ratio test was used to assess the significance of differential expression. Multiple hypothesis testing correction was performed by controlling the FDR using the R function 'p.adjust'[49]. To identify cell-specific genes, we used the procedure we have previously described[50]. Briefly, differential gene expression tests were performed between all pairwise combinations of clusters. For a given cell type, putative marker genes were ranked using two stringent criteria: maximum FDR $Q$-value ($Q_{max}$) and the minimum $\log_2$-transformed fold change ($FC_{min}$), which represent the weakest effect and significance across all comparisons. Cell type signature genes (Fig. 4, Extended Data Figs. 4 and 6 and Supplementary Tables 1 and 3) were obtained using a $Q_{max} = 0.05$ and $FC_{min} = 0.25$. To define signature genes for the more similar ionocyte subtypes, a more lenient criterion was used, an adjusted Fisher's combined $P$ value ($Q_{Fisher}$) across the pairwise tests $Q_{max} = 0.05$ and $FC_{min} = 0.25$ (Fig. 4 and Supplementary Table 1). Ion channel lists were obtained from the *Guide to Pharmacology* (www.guidetopharmacology.org), University of Edinburgh, UK[51].

### Testing for differences in cell-type proportions

To assess the significance of changes in the fraction of cells under different conditions, we used Bayesian negative binomial regression,

estimated using the R package 'brms'. This enabled us to model the number of each cell type detected in each donor and to test the effect of genetic perturbations while controlling for variability among biological replicates (donor ferrets). For each cell type, we modelled the number of cells detected in each donor as a random count variable using a negative binomial distribution. The rate of detection (that is, the relative proportion of that cell type) was modelled by using the natural log of the total number of all cells profiled from a given donor as an offset term (Extended Data Fig. 4d). To test the effect of *FOXI1* deletion, donor genotype was added to regression models as a categorical covariate (Extended Data Fig. 6g). Significance of changes in cell-type proportions was assessed using the 90%, 95% and 99% posterior credible intervals for the main effect of the genotype covariate.

## Proportion of CFTR-expressing airway epithelial cells

To assess the significance of changes in proportion of CFTR-expressing airway epithelial cells between wild-type (both *FOXI1*-Cre[ER] FACS-enriched and unenriched samples) and *FOXI1*-KO samples (Extended Data Fig. 6i), we aggregated cells from all samples together rather than averaging the proportion over samples. Some samples showed low proportions of certain cell types. This causes samples with low numbers of certain cell types to spuriously bring down the average proportion of CFTR-expressing cells (for example, a single ionocyte in a non-enriched sample, resulting in a CFTR-expressing proportion of 0%). The R package 'prop.test' v.3.6.2 was used to calculate statistical significance. Because of the statistical approach used, it was not possible to plot the individual data points on the graphs in Extended Data Fig. 6i.

## Interspecies comparison of rare cell-type transcriptomes

To compare rare cell types (Fig. 5), single-cell data from cultured ferret airway epithelial cells (this study) were merged with published single-cell data from mouse trachea[1] and human airway epithelial cells[52], and merged using Seurat data integration as described above. Before running dimensionality reduction, all genes that were strongly different by species were identified using a one-way ANOVA. Genes with an *F* statistic over the 90th percentile (indicating high confidence of differential expression by species) were removed. Cell-type labels (Fig. 5a) were taken from the respective studies, along with species annotation (Fig. 5b). To compute the transcriptional similarity between rare cell types from each species, all pairwise cell–cell Pearson correlations across the set of variable genes (defined as above) were computed, and then aggregated using the mean in each cell type and species combination (Fig. 5i).

## Pseudotime analysis using PAGA and elastic principal graphs

We isolated the 1,497 tuft cells, ionocytes and PNECs from our ferret scRNA-seq data to examine the progression from their putative common progenitor to mature rare epithelial subsets. The PAGA algorithm[34], implemented using scanpy[53], was used to project cells into a low-dimensional manifold, after defining unsupervised clusters generated using the Leiden algorithm[54]. Elastic principal graphs[35] were then used to fit a branching tree through the PAGA co-ordinate space. Spurious single-node branches were removed. The node in the rare progenitor cluster was manually selected as the root node for pseudotime calculation, computed using the ElPiGraph R package.

## Validation of transcriptional signatures using RNAscope

The ferret trachea and lung were postfixed in 4% paraformaldehyde for 18–24 h. After postfixation, tissues were cryoprotected in sucrose, embedded as frozen OCT blocks and then cut into 7–15-μm-thick sections. RNAscope Multiplex Fluorescent Kit (Advanced Cell Diagnostics) was used per the manufacturer's recommendations. The multiplex RNAscope assay uses three probe sets against three target molecules (*CFTR*, *FOXI1*, *EGFP*) and markers of ionocytes subtypes in various combinations (type A: *CFTR*, *FOXI1*, *SLC12A2* or *BSND*; type B: *CFTR*, *KRT7*, *EGFP*); probes are referred to as channel 1, channel 2 and channel 3 probes, respectively. All amplification and detection steps were done using the kit instructions. Finally, the multiplex assay probe sets were detected with Fluorescein (channel 1), Cyanine3 (channel 2) and Cyanine5 (channel 3). Images of tissues were acquired with a confocal microscope (Zeiss 880 or Zeiss 980). Scale bars were added to each image using ImageJ. Images were visualized using ImageJ software. Probes used for RNAscope (Advanced Cell Diagnostics): *CFTR* (C1), *EGFP* (C2), *SLC12A2* (C2), *BSND* (C2), *KRT7* (C3) and *FOXI1* (C3).

## Statistical analysis

All non-bioinformatic experimental results are presented as the mean ± s.e.m., and R 4.2.0 (www.r-project.org) and Prism 9 (GraphPad) were used for statistical analysis. Student's *t*-test and one-way ANOVA were performed when appropriate. *P* values of less than 0.05 were considered statistically significant. Statistical analysis of scRNA-seq data is described in the bioinformatics section.

## Materials availability

All unique and stable reagents and transgenic ferrets generated in this study are available under institutional MTA without restriction to non-for-profit institutions from the corresponding authors.

## Reporting summary

Further information on research design is available in the Nature Portfolio Reporting Summary linked to this article.

## Data availability

Single-cell sequencing data are available in the GEO, accession no. GSE233654. Source data are provided with this paper.

## Code availability

Customized code for the computation of cross-species similarity between rare cells is available on GitHub (https://github.com/yifand64/ferret_ionocyte). The remainder of the code used for the analysis is standard.

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

**Acknowledgements** We acknowledge the infrastructure and expertise of the Small Animal Imaging Core facility and the Iowa Institute for Biomedical Imaging, which are supported by the University of Iowa and the Carver College of Medicine. We acknowledge that BioRender granted a license (agreement number RO25OM5CTF) that permits BioRender content to be sublicense for use in journal publication. This work was funded by the following grants from the National Institutes of Health (grant P01 HL152960, NHLBI contract 75N92019R00010; grants P30 DK054759 and R01 HL165404 to J.F.E.), the Cystic Fibrosis Foundation (grants ENGELH20XX0 and 001478XX220 to J.F.E.) and the Roy J. Carver Chair in Molecular Medicine (to J.F.E.). A.L.H. was supported by a Parker B. Francis Fellowship.

**Author contributions** F.Y., X.S., A.L.H. and J.F.E. conceived the study. X.S., A.L.H. and J.F.E. supervised research. F.Y., Z.Y., M.Y. and X.S. designed and generated transgenic ferrets supervised by J.F.E. F.Y. designed and analysed electrophysiology data, ASL properties data, tissue whole-mount staining data and molecular biology data with G.N.G., Y.Z., I.E., V.I., N.S.J., J.J.W., B.H.R. and W.S.. F.Y., G.N.G. and D.K.M. collected and analysed tissue section staining data. F.Y., G.N.G., E.L., D.T.M., K.J. and Y.D. designed and performed ferret airway single-cell experiments. X.L. and Q.C. designed and performed double transgenic ferrets lineage tracing. K.L.W. and P.G.R. revised the ferret genome build. M.W., S.A.W., M.R.A., D.N.P., V.A., D.W.D. and T.J.W. performed mucociliary clearance experiments. A.S.V. and P.M.H. supervised halide sensor experiments. S.B., C.M.F., H.M.L., G.J.T. and S.M.R. collected, prepared and interpreted trachea μOCT data. G.N.G. and K.S. performed *FOXI1*-KO ferret rearing. D.B. provided breeding management and genotyping on all transgenic ferret models. G.N.G., Y.Z., S.M.R. and J.J.W. helped to edit the manuscript. F.Y., A.L.H., X.S. and J.F.E. wrote the manuscript with input from all authors.

**Competing interests** The authors declare no competing interests.

**Additional information**
**Correspondence and requests for materials** should be addressed to Adam L. Haber, Xingshen Sun or John F. Engelhardt.

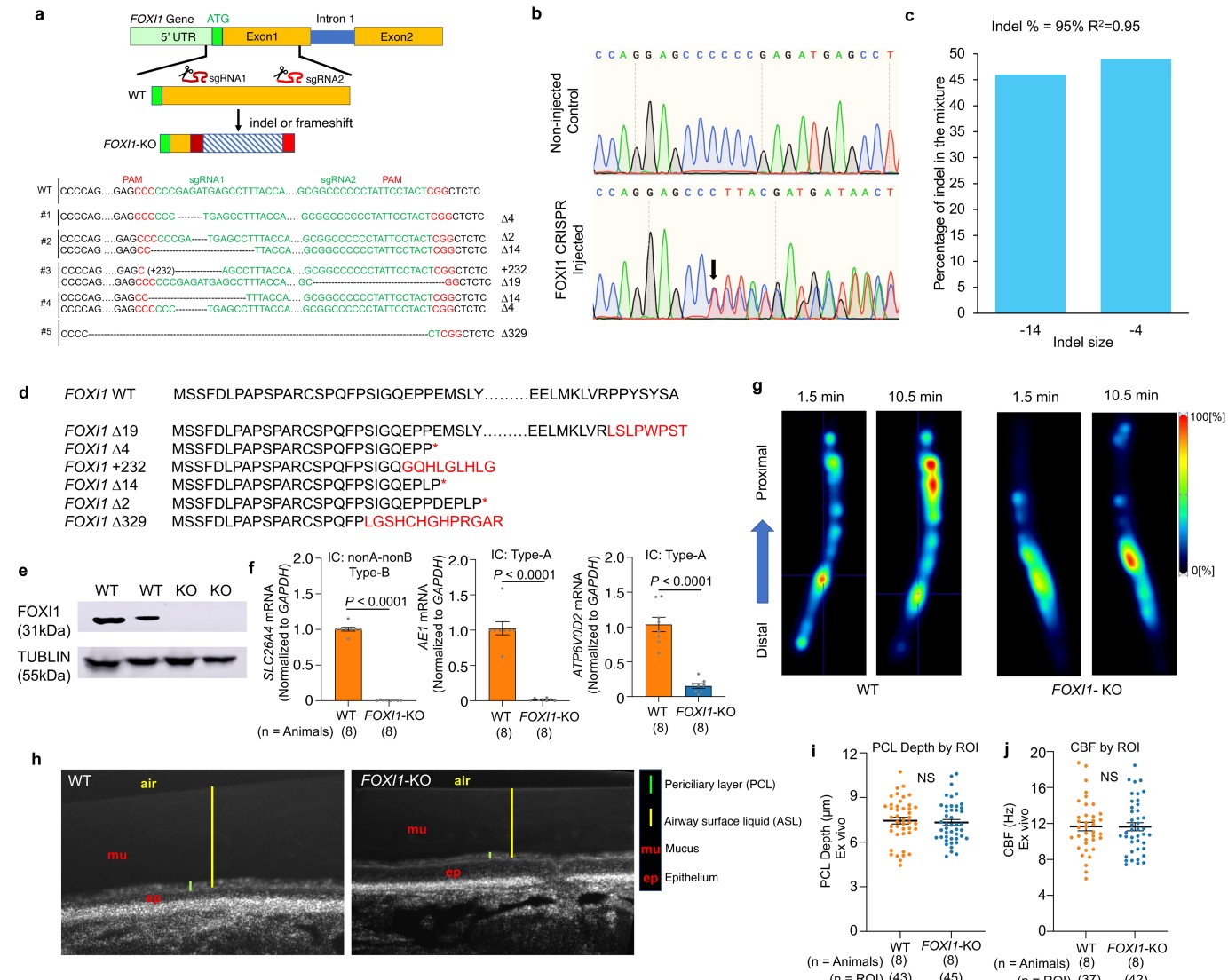

**Extended Data Fig. 1 | *FOXI1* deletion impairs kidney intercalated cell formation and mucociliary clearance in ferrets. a**, Schematic of the approach to generate *FOXI1*-KO ferrets by Cas9/sgRNA RNP injection of zygotes. Bottom, sequence of the founder indel insertions (+) and deletions (Δ) with gRNA sequence (green) and PAM sequence (red) shown. **b**, Representative sequence chromatograms of the DNA target site from non-injected (control) and *FOXI1* RNP-injected zygotes (black arrow indicates cleavage site). **c**, Representative TIDE analysis from a compound heterozygous *FOXI1*-KO (−14/−4) founder ferret showing percentage of DNA editing for each indel size. **d**, Predicted amino acid sequence of ferret FOXI1 in wild-type (top) and *FOXI1*-KO founders (bottom). Asterisks indicates a premature stop codon found in the coding sequence of FOXI1 mutants. Red letters indicate frameshift mutation caused by indels. **e**, Kidney tissue lysates from *FOXI1*-KO and WT animals were collected and subjected to Western blotting with FOXI1 antibody. Representative samples

show the absence of FOXI1 protein in KO animal kidney tissues. **f**, RT-qPCR of kidney mRNA showing absent or reduced intercalated cell (IC) marker (*SLC26A4*, *AE1*, *ATP6V0D2*) expression in *FOXI1*-KO ferrets. Mean ± s.e.m.; n = 8 animals in each group. **g**, Representative $^{68}$Ga-MAA PET/CT images of *FOXI1*-KO and WT ferret trachea at the indicated time points showing reduced clearance in *FOXI1*-KO. **h**, Micro-optical coherence tomography (μOCT) imaging of *FOXI1*-KO and WT ferret tracheal explants. Images show airway surface liquid (ASL; yellow bar), mucus layer (mu), and periciliary liquid (PCL; green bar) on the luminal surface. Analysis of μOCT images from explanted ferret trachea yields numerical values for functional and anatomic parameters. **i,j**, PCL depths (i) and ciliary beat frequency (CBF) (j) were analyzed geometrically as shown in bar graph. Mean ± s.e.m.; n = 8 animals in each group. Statistical significance was determined by: (f) two-tailed Student's t-test and (i,j) ROI using t-tests with pooled SD by R, statistical test was two-sided.

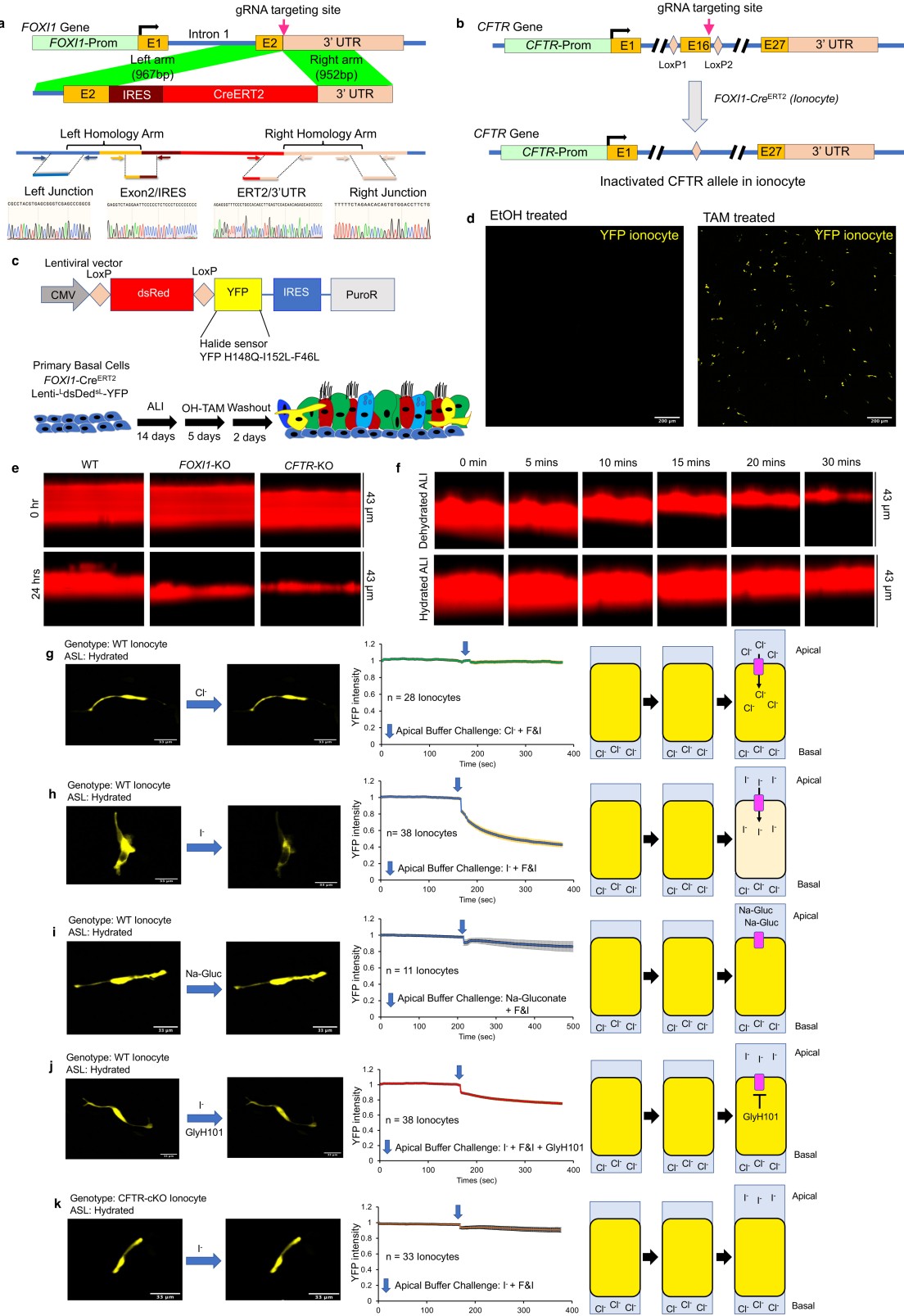

**Extended Data Fig. 2** | See next page for caption.

**Extended Data Fig. 2 | *FOXI1*-Cre^ERT2 and *CFTR* conditional KO (*CFTR*^L/L) ferret models demonstrate CFTR is required for ionocyte apical uptake of anions.** CRISPR homology-directed repair (HDR) was used to generate *FOXI1*-Cre^ERT2 and *CFTR*^L/L ferrets. **a**, Schematic of the strategy for generating transgenic ferrets with an IRES-CreERT2 insertion in the *FOXI1* 3'-UTR. **b**, Schematic of the floxed exon-16 in *CFTR*^L/L ferrets and strategy for deletion of *CFTR* in *FOXI1*-Cre^ERT2::*CFTR*^L/L (*CFTR*-cKO) ferrets. **c**, Primary *FOXI1*-Cre^ERT2 airway basal cells transduced with a lentivirus encoding LoxP-dsRED-stop-LoxP-YFP-H148Q/I152L cassette (herein called *FOXI1*-Cre^ERT2::YFP^H148Q/I152L) and differentiated at ALI, treated with hydroxy-tamoxifen (OH-Tam) and then used for functional studies of halide transport. **d**, Scattered YFP-positive ionocytes were observed in the pseudostratified airway epithelium of only OH-Tam treated differentiated *FOXI1*-Cre^ERT2::YFP^H148Q/I152L ALI airway cultures. Representative images from 3 independent ferret donors. **e**, Representative images of ASL height from differentiated WT, *CFTR*-KO, and *FOXI1*-KO ALI cultures challenged with 18 µl of Alexa-dye containing buffer (time zero) and following equilibration 24 hrs later. Representative images from n = 11 (WT), n = 10 (*FOXI1*-KO) and n = 9 (*CFTR*-KO) independent cultures. **f**, Dehydration experiment on WT ALI cultures monitoring the ASL height following apical perfusion of non-humidified 5% $CO_2$ for the indicated times. 20 min of dehydration was chosen for basolateral halide sensor assays (Extended Data Fig. 3) since the ASL height approached that observed *CFTR*-KO and *FOXI1*-KO cultures. Representative images from 3 independent experiments. **g–k**, Representative images and traces of apical I^- uptake in YFP halide sensor expression ionocytes of ALI cultures. **g**, No YFP quenching is observed in WT ionocytes after the addition of apical Cl^- buffer (18 µl) with Forskolin/IBMX (F&I) to stimulate CFTR (negative control). Mean ± s.e.m.; n = 28 ionocytes. **h**, YFP quenching is observed in WT ionocytes following the addition of apical I^- buffer with F&I. Mean ± s.e.m.; n = 38 ionocytes. **i**, YFP quenching, as shown in (h), is not observed following the addition of apical Na-Gluconate (Na-Gluc) buffer with F&I (negative control). Mean ± s.e.m.; n = 11 ionocytes. **j**, YFP quenching, as shown in (h), is reduced following the addition of apical I^- buffer, F&I, and GlyH101 CFTR inhibitor (negative control). Mean ± s.e.m.; n = 38 ionocytes. **k**, YFP quenching is not observed in *CFTR*-cKO ionocytes (*FOXI1*-Cre^ERT2::*CFTR*^L/L) following application of apical I^- buffer with F&I. Mean ± s.e.m.; n = 33 ionocytes.

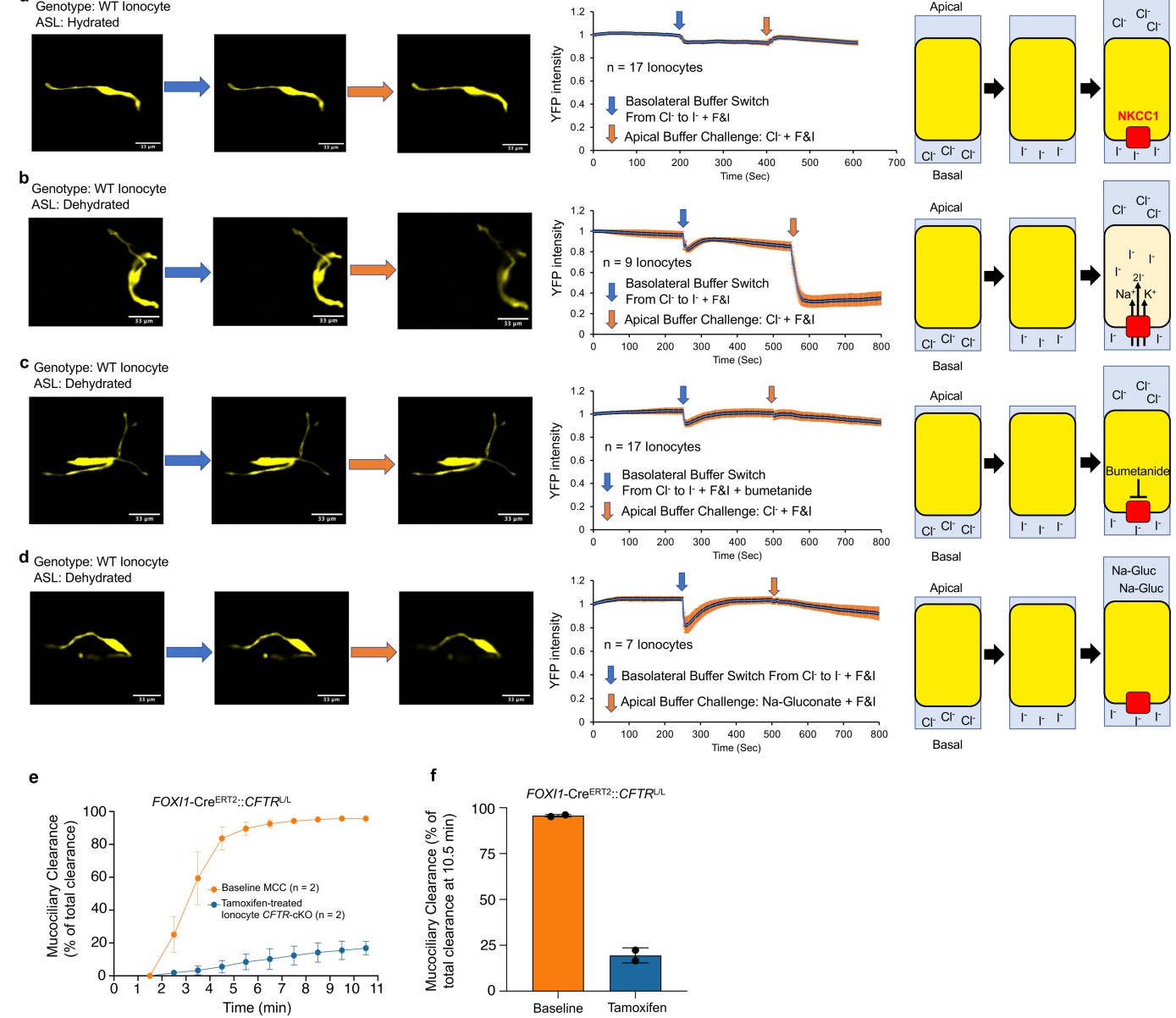

**Extended Data Fig. 3 | NKCC1 is required for ionocyte basolateral uptake of anions.** Representative images and traces of basolateral I⁻ uptake in YFP halide sensor expression ionocytes of ALI cultures. **a**, Basolateral Cl⁻ to I⁻ buffer exchange in the presence of forskolin/IBMX (F&I) does not lead to ionocyte YFP quenching in homeostatic apically-hydrated cultures at baseline or following apical Cl⁻ buffer addition (18 μl) with F&I. Mean ± s.e.m.; n = 17 ionocytes. **b**, In apically-dehydrated cultures, basolateral Cl⁻ to I⁻ + F&I buffer exchange leads to basolateral I⁻ uptake, but only after apical Cl⁻ buffer addition with F&I. Mean ± s.e.m.; n = 9 ionocytes. **c**, Basolateral I⁻ uptake by ionocyte, as shown in (b), requires the NKCC1 channel and is blocked by bumetanide. Mean ± s.e.m.; n = 17 ionocytes. **d**, Basolateral I⁻ uptake by ionocyte, as shown in (b), requires apical Cl⁻ and is not observed after apical Na-gluconate (Na-Gluc) buffer addition with I&F. Mean ± s.e.m.; n = 7 ionocytes. **e,f**, In vivo tracheal mucociliary clearance (MCC) measured by ⁶⁸Ga-MAA PET-CT in *FOXI1*-Cre^ERT2^::*CFTR*^L/L^ ferrets prior to and following CFTR deletion with tamoxifen (n = 2 animals, range in values is shown on both graphs).

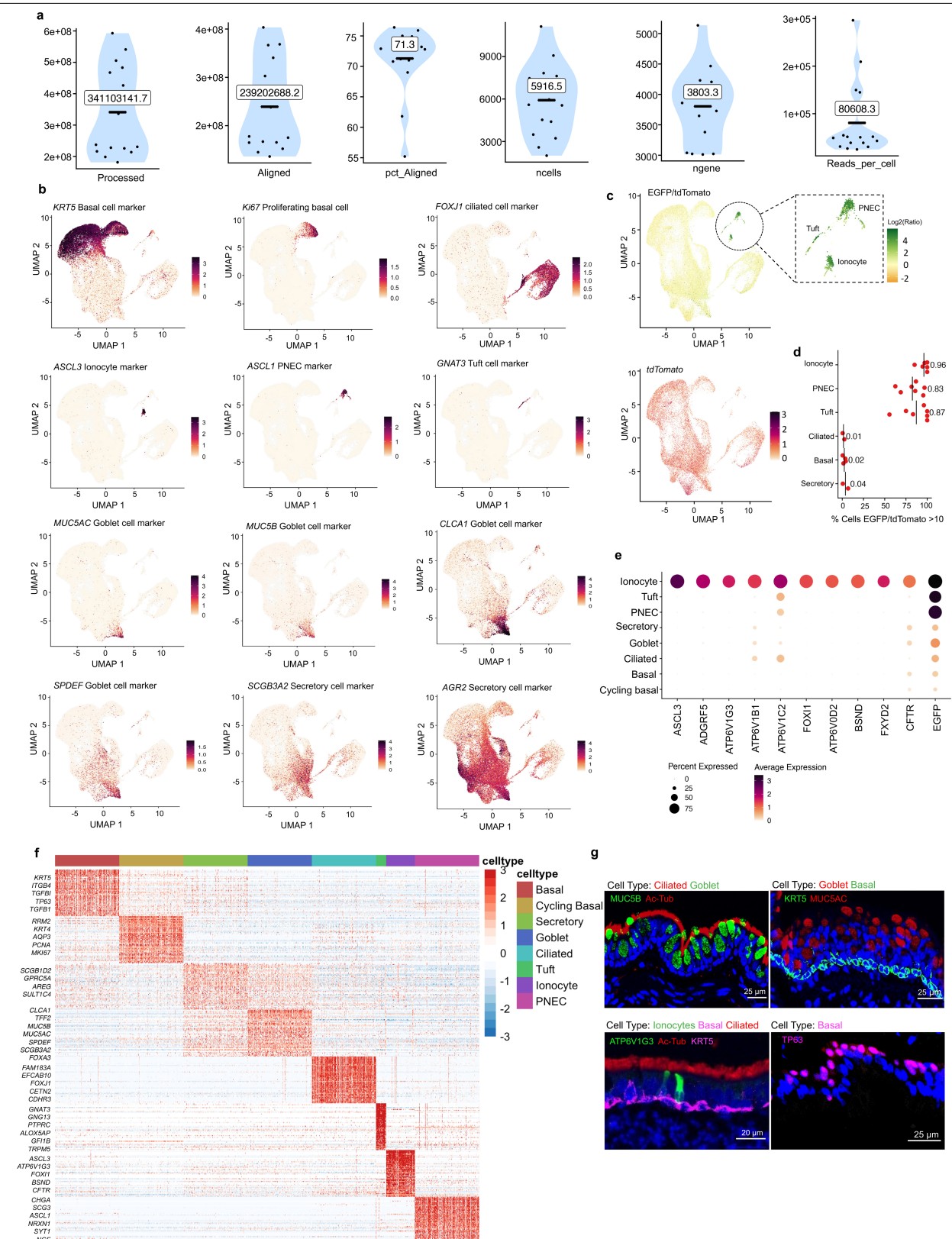

**Extended Data Fig. 4** | See next page for caption.

**Extended Data Fig. 4 | Single-cell expression atlas of ferret proximal airway epithelial cells. a**, Quality metrics for droplet-based 3′ scRNA-seq data. **b**, Cluster annotation by the expression of known cell type markers. UMAP visualization of 77,099 single cells where individual points correspond to single cells. **c**, UMAP visualization of EGFP/tdTomato (ratio) and tdTomato expression for all cells sequenced. Note, *FOXI1*-Cre[ERT2]::ROSA-TG ALI cultures used for ionocyte enrichment were treated with OH-Tam throughout basal cell differentiation at ALI. This leads to lineage labeling (EGFP expression) and enrichment by FACS of all rare cell types (ionocytes, tuft, and PNECs). Thus, FOXI1 is expressed in common early precursor of these rare cell types. **d**, Percentage of cells with a ratio of EGFP/tdTomato greater than 10. **e**, Ferret pulmonary ionocyte markers. Expression level of ionocyte marker genes (columns) in each airway epithelial cell type (rows). Note that EGFP⁺ *FOXI1*-lineage labeled tuft and PNECs extinguish FOXI1 and other ionocyte markers when fully differentiated. **f**, Ferret airway epithelial cell gene expression signatures. The relative expression of genes (rows) across cell types (columns) is shown, sorted by cell types. **g**, Cell type marker immunostaining of ferret tracheal sections for different cell clusters. Representative images of n = 3 independent ferrets.

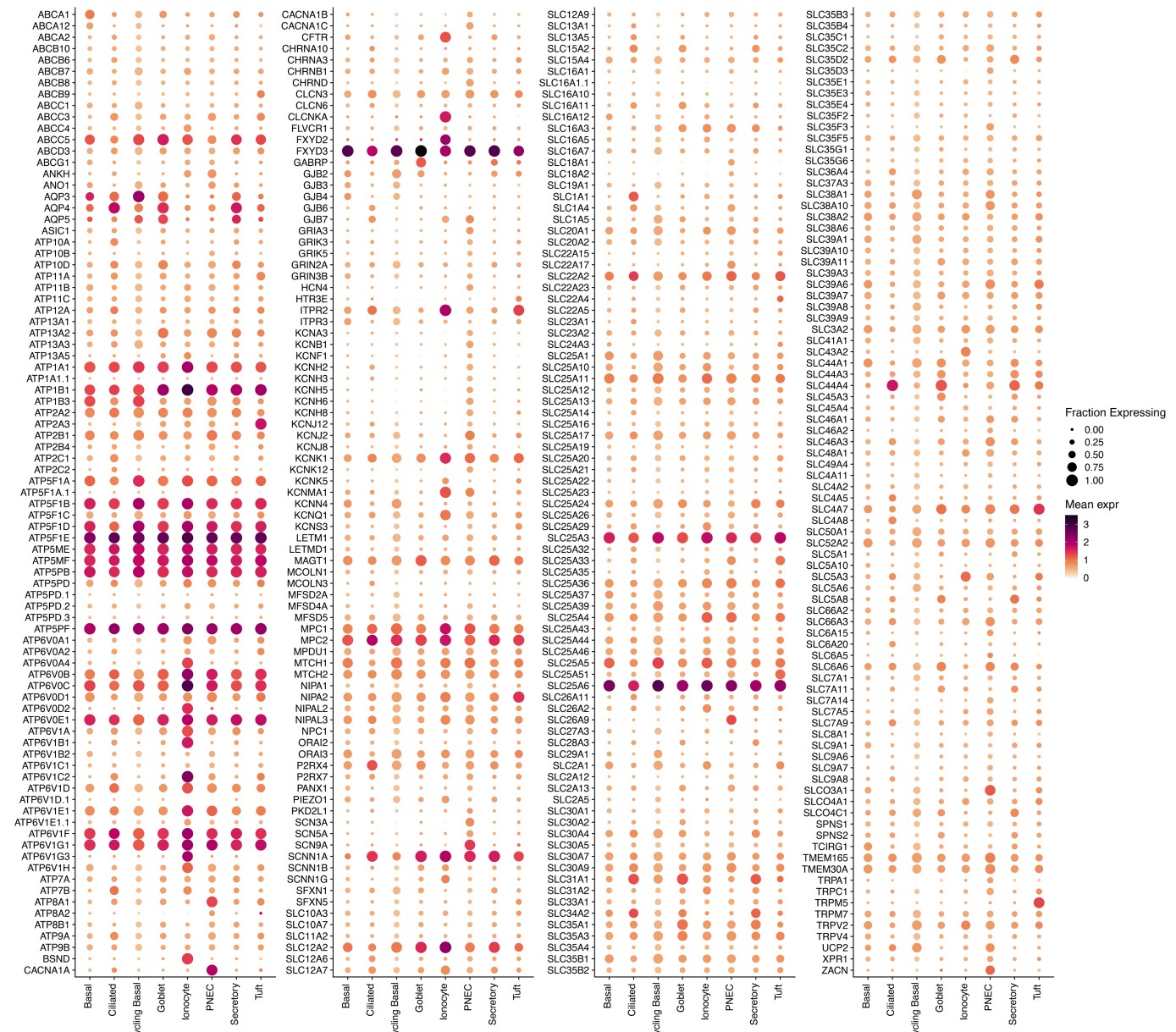

**Extended Data Fig. 5 | Channelome of all ferret proximal airway cell types.** mRNA expression levels for all ion and water channels and gap junctional genes are listed for the various airway cell types regardless of whether they are differentially expressed. Plots show the mean relative expression level (unnormalized between genes) and the percentage of cells that express each gene.

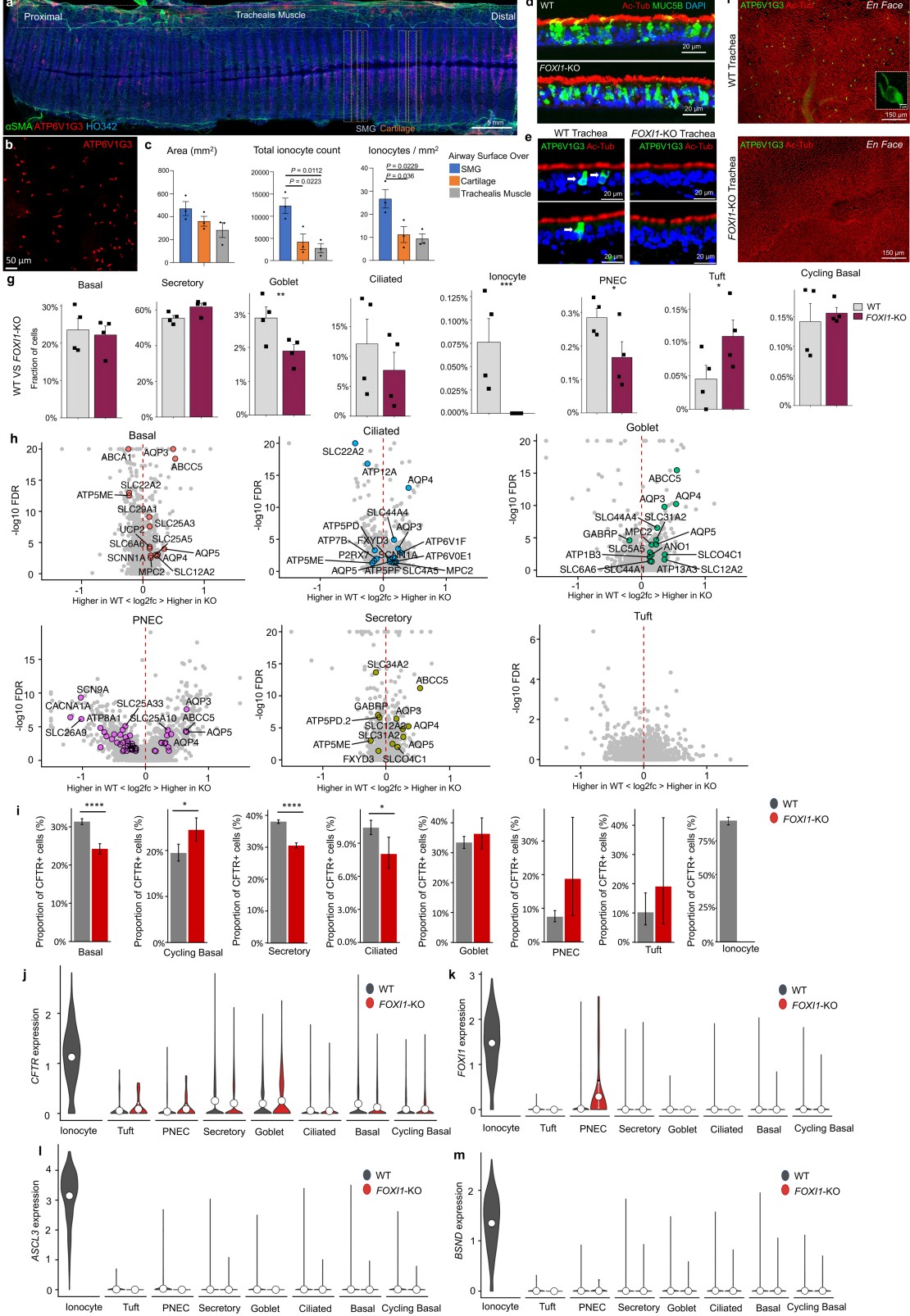

**Extended Data Fig. 6** | See next page for caption.

**Extended Data Fig. 6 | Ion and water channels are transcriptionally altered in *FOXI1*-KO airway epithelia. a**, Cleared WT ferret trachea was whole mount immunostained for ionocyte (ATP6V1G3) and submucosal gland (α-SMA) markers. Boxed areas show examples of regions quantified for ATP6V1G3+ ionocyte numbers in the surface airway epithelia above submucosal glands (SMG), cartilage, and trachealis muscle. **b**, Higher magnification of the tracheal surface showing single channel ATP6V1G3+ ionocytes. Representative images of n = 3 donors. **c**, Quantification of ionocyte numbers in the surface airway plane in different regions of ferret trachea from WT animals (the entire trachea was quantified, n = 3 WT donors). Bar graph showing the total area quantified, the total number of ionocytes, and the number of ionocytes per mm². Graphs show the mean ± s.e.m. Statistical significance was determined by two-tailed Student's *t*-test. **d**, Sections of ALI cultures of WT and *FOXI1*-KO airway epithelia immunostained for Ac-Tub (ciliated cells) and MUC5B (goblet cells). Representative images of n = 4 ferrets from each group. **e,f**, Immunostaining of (e) tracheal sections and (f) whole-mount trachea from WT and *FOXI1*-KO ferrets for ATP6V1G3 (ionocytes) and Ac-Tub (ciliated cells). Ionocytes (arrows) are depleted in *FOXI1*-KO ferrets. Representative images of n = 4 ferrets from each group. **g**, Cell type proportions across WT and *FOXI1*-KO scRNA-seq runs (Mean +/− s.e.m.; n = 4 donors each genotype, excluding *FOXI1*-Cre^ERT2^::ROSA-TG ionocyte enriched samples). Statistical significance was determined by Bayesian analysis: goblet cells, ** FDR < 0.01, 95% PI; ionocyte, *** FDR < 0.001, 99% PI; Tuft and PNEC, * FDR < 0.05, 90% PI. **h**, Volcano plots of differential channel expression within various epithelial cell types derived from WT and *FOXI1*-KO ALI cultures. **i**, The proportions of *CFTR* expressing basal, secretory, and ciliated cells is reduced in *FOXI1*-KO cultures as compared to WT culture (including *FOXI1*-Cre^ERT2^::ROSA-TG ionocyte enriched samples). This may be due to lower levels of ambient *CFTR* RNA from lysed ionocytes. Data shows the mean, n = 4 donors in KO, n = 12 donors in WT, error bars represent 95% confidence interval, **** FDR < 0.0001, * FDR < 0.05. The proportion of CFTR expressing cells per cell type were calculated by aggregating cells from all samples together (see Methods). **j-m**, Expression of ionocyte markers *CFTR*, *ASCL3*, *FOXI1* and *BSND* in different cell types from WT and FOXI1-KO cultures. Data shows the mean, n = 18,664 cells from 4 *FOXI1*-KO donors, n = 58,435 cells from 8 WT donors. Large white point shows the mean, error bars represent 95% confidence interval. Of note, deletion of a portion of *FOXI1* exon-1 led to enhanced mRNA expression from the *FOXI1* locus only in PNECs, suggesting a potential functional role for *FOXI1* in PNECs. The *FOXI1* gene has only two exons and thus its mRNA likely has minimal non-sense mediated decay.

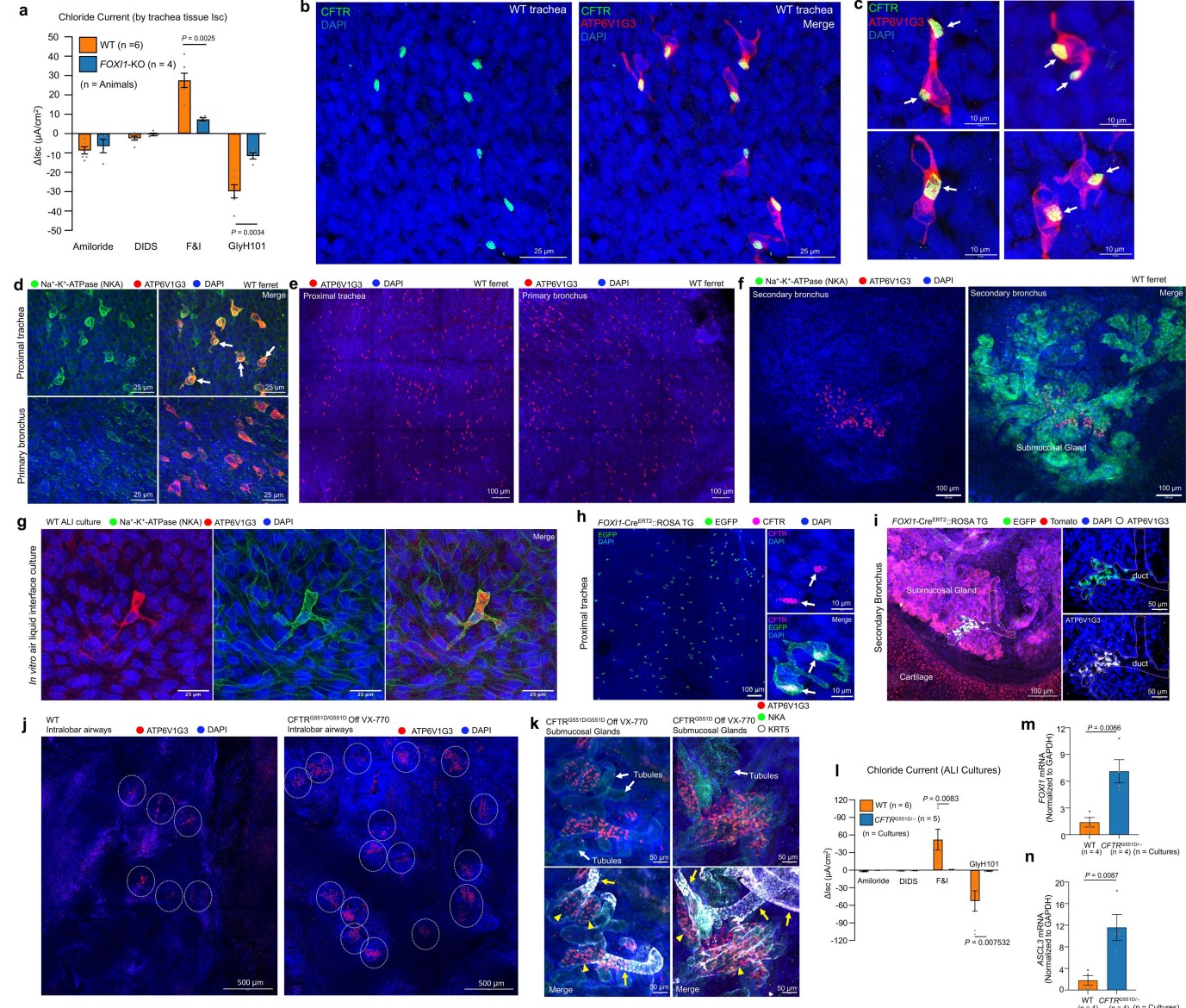

**Extended Data Fig. 7 | Pulmonary ionocyte function and localization pattern at different levels of the airway. a**, Change in chloride short circuit current (ΔIsc) from *FOXI1-KO* and WT trachea following the addition of the indicated channel antagonists and agonists (n = 6 donors for WT, 4 donors for KO, with 2-3 tracheal samples quantified from each donor and averaged). F&I, forskolin and IBMX. Graph shows the mean ± s.e.m. *P* value determined by two-tailed Student's t-test. **b**, CFTR protein localization in ATP6V1G3⁺ ionocytes following whole mount staining of WT ferret trachea. Representative images from n = 4 ferrets. **c**, High magnification images of ATP6V1G3⁺ ionocytes showing CFTR protein localizes to an "apical cap" (arrows). Representative images of ionocytes with two apical caps (upper panels) or one apical cap (lower panels) are shown. Images from n = 4 ferrets. **d**, Na⁺-K⁺-ATPase (NKA) staining of ATP6V1G3⁺ ionocytes in the proximal trachea of WT ferret by whole mount staining. NKA staining in ionocytes localized to an apparent basolateral rim around the apical cap (marked by arrows). Proximal trachea showed more abundant NKA expression in ionocytes as compared to primary bronchus. Representative images from n = 4 ferrets. **e**, Representative whole mount images showing the frequency of ATP6V1G3⁺ ionocytes in the proximal trachea and primary bronchus of WT ferrets. Images from n = 4 ferrets. **f**, Representative whole mount images showing clusters of ATP6V1G3⁺ ionocytes in NKA⁺ submucosal glands in the secondary bronchus of WT ferrets. Images from n = 4 ferrets. **g**, Co-localization of ATP6V1G3 and NKA in ionocytes of WT ferret ALI

cultures. Representative images from cultures derived from n = 3 donors. **h**, Representative whole mount image of a tamoxifen-induced adult *FOXI1*-Cre^ERT2^::ROSA-TG ferret trachea demonstrating CFTR staining within an apical cap (arrows) of EGFP⁺ ionocytes. Images from n = 3 ferrets. **i**, Representative image of a tamoxifen-induced adult *FOXI1*-Cre^ERT2^::ROSA-TG ferret secondary bronchus demonstrating traced EGFP⁺ATP6V1G3⁺ ionocytes in a submucosal gland collecting duct (marked by dotted lines). Images from n = 3 ferrets. **j**, Localization of ATP6V1G3⁺ ionocytes in whole mount stained intralobar airways from a WT and *CFTR*^G551D/G551D^ (reared off VX-770) ferrets, demonstrating expansion of ionocyte clusters (circles) in the CF airway. Images from n = 4 (WT) and n = 3 (CF) ferrets. **k**, Localization of ATP6V1G3⁺ ionocytes in KRT5⁺-enriched gland primary ducts (yellow arrows), adjacent NKA⁺ collecting ducts (yellow arrowheads), and NKA⁺ glands tubules (white arrows) in *CFTR*^G551D/G551D^ (reared off VX-770) ferrets (whole mount staining of intralobar bronchus). Images from n = 3 (CF) ferrets. **l**, CFTR-mediated chloride current from WT and *CFTR*^G551D/−^ ALI cultures (n = 5-6 ALI cultures derived from one donor). Graph shows the mean ± s.e.m. *P* value determined by one-tailed Student's t-test. **m** and **n**, RT-qPCR detection of mRNA for ionocyte-specific transcription factors (*FOXI1*, *ASCL3*) in differentiated WT and *CFTR*^G551D/−^ ALI cultures demonstrating expansion of ionocytes in CF epithelium (n = 4 samples from one donor, each sample with 3 ALI cultures pooled for mRNA). Graphs show the mean ± s.e.m for the n indicated in each graph. *P* value determined by by two-tailed Student's t-test.

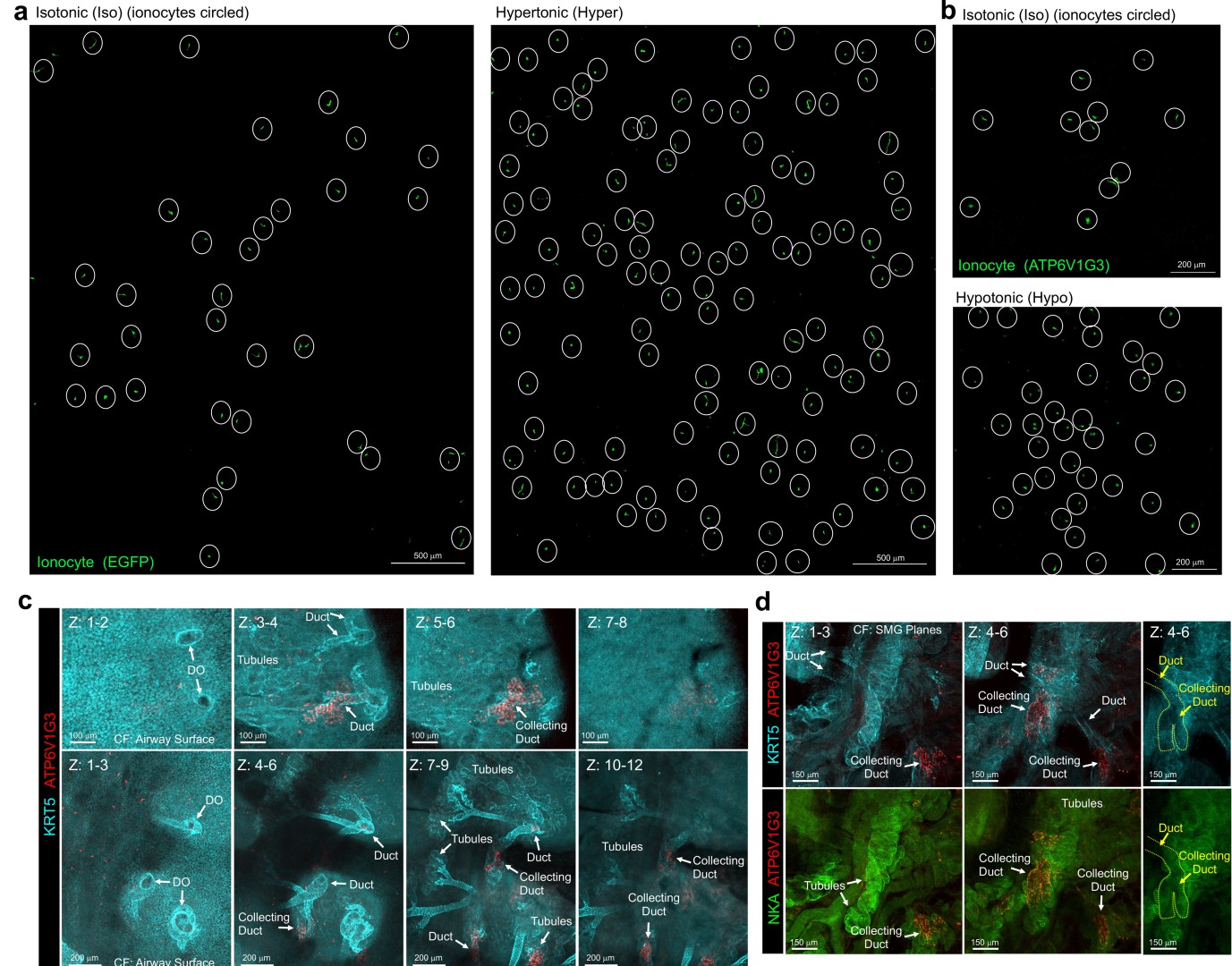

**Extended Data Fig. 8 | Osmotic stress in ALI cultures and a CF disease state in vivo induced ionocyte expansion. a and b**, Hyperosmotic (a) and hypoosmotic (b) ALI culture conditions induce ionocyte expansion. Ionocytes were localized either by lineage tracing using *FOXI1*-Cre[ERT2] or ATP6V1G3 staining. Representative images of cultures derived from n = 5 donor ferrets for hyperosmotic stress and n = 4 ferrets for hypoosmotic stress. **c and d**, Additional panels related to Extended Data Fig. 7k. Whole mount trachea immunostaining demonstrating expanding ATP6V1G3[+] ionocytes in KRT5[+]-enriched submucosal gland (SMG) ducts and NKA[+] tubules of *CFTR*[G551D/G551D] ferrets (reared off VX-770). Confocal Z planes are indicated and show apical surface to SMG planes (c) or just the SMG plane (d). DO, duct opening. Representative images from n = 3 CF ferrets.

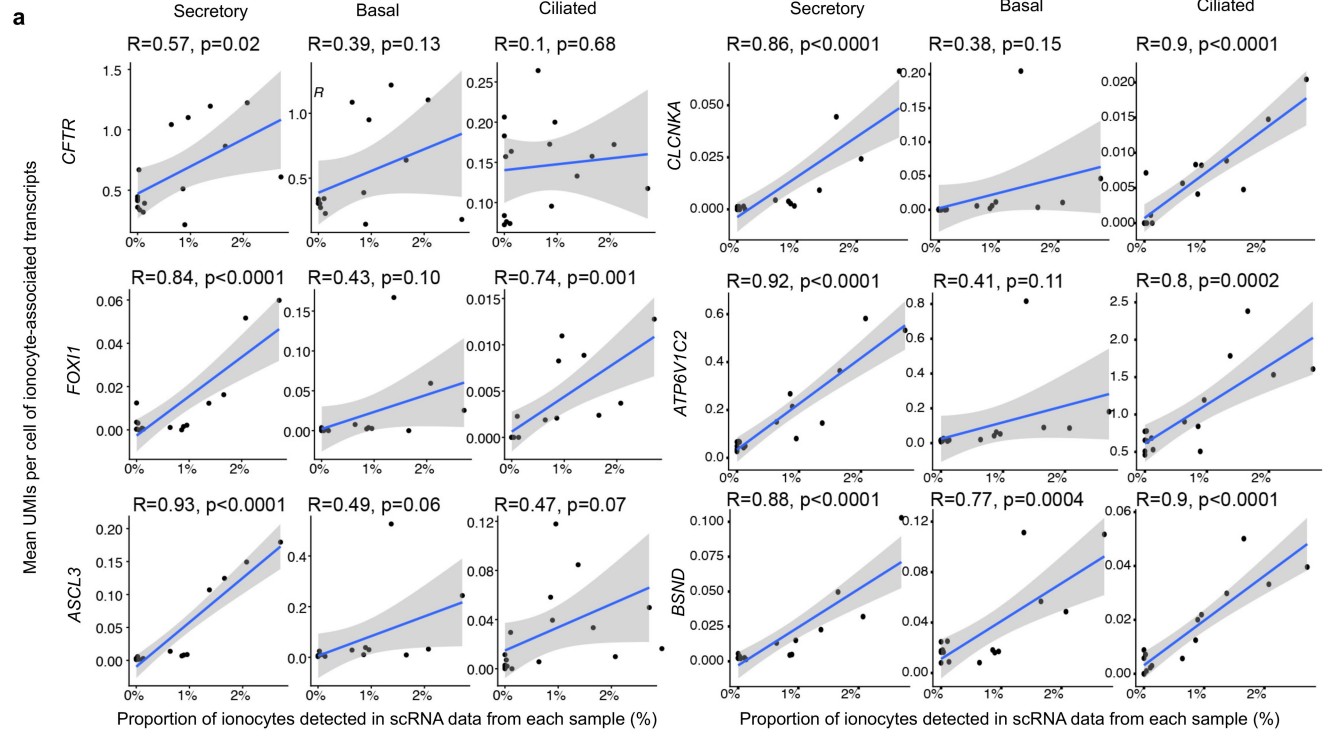

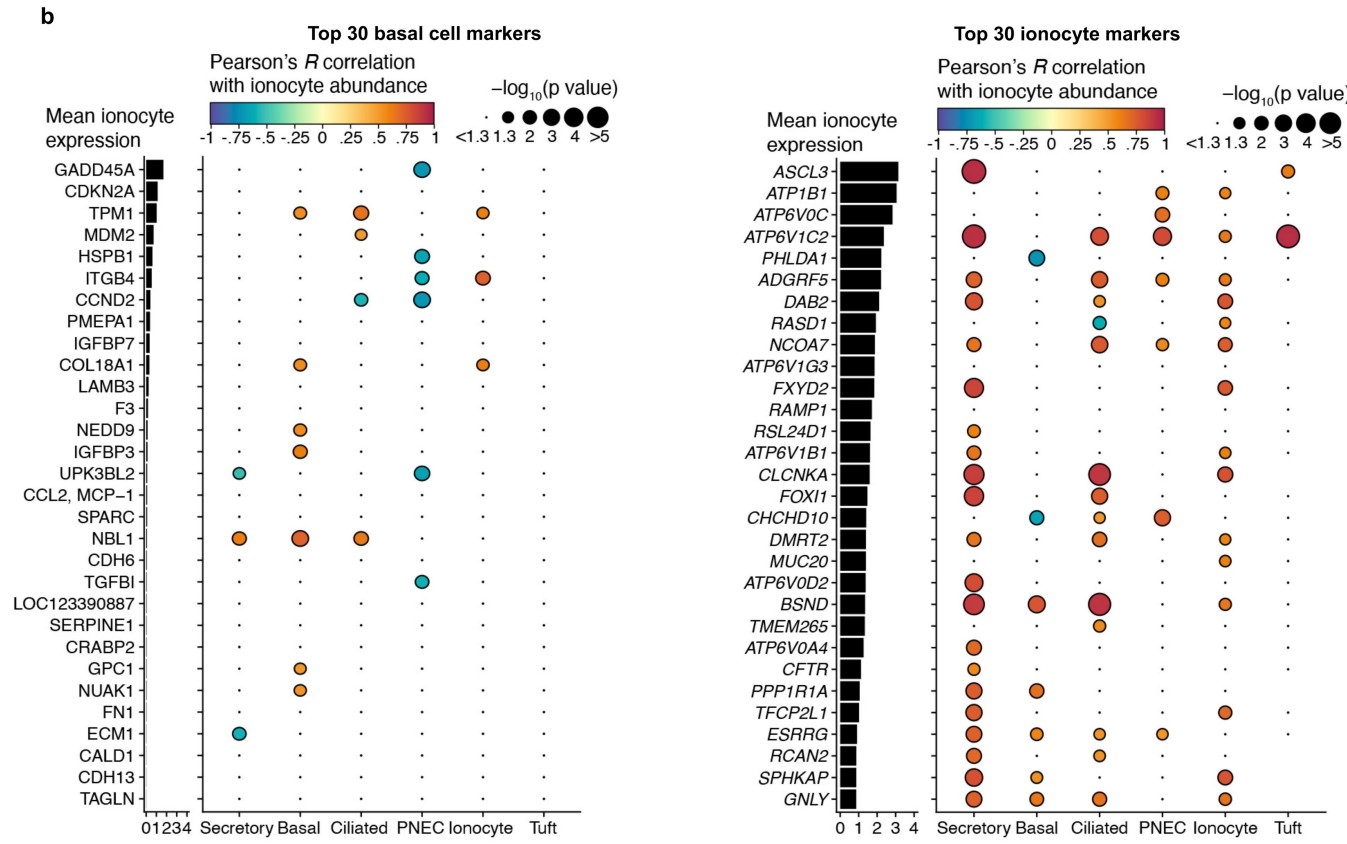

**Extended Data Fig. 9** | See next page for caption.

**Extended Data Fig. 9 | Ambient RNA from fragile ionocytes enhances apparent *CFTR* expression in other cell types. a**, Expression (mean UMI counts, y-axis) of ionocyte markers *CFTR*, *FOXI1*, *ASCL3*, *CLCNKA*, *BSND*, *ATP6V1C2* (rows) in cells of each type (columns) from each ALI culture (points) correlated with the number of ionocytes detected in each culture (x-axis), particularly in secretory cells. These data include *FOXI1*-KO and *FOXI1*-Cre[ERT2]:: ROSA-TG FACS-enriched sequence runs, which had the smallest and largest percentage of ionocytes, respectively. Blue line and shaded area show line of best fit and 95% confidence interval, respectively, from linear regression models fit independently for each gene. *P* values by Wald test. **b**, Dotplots show the Pearson's R correlation (dot color, legend) and significance (dot size, legend) between proportion of ionocytes detected and expression level (as in Extended Data Fig. 6i) for the top 30 basal cell markers (rows, left) and top 30 ionocyte markers (rows, right). Genes are ordered by expression level, which is shown on left hand barplots with $\log_2(TPM + 1)$ on the x-axis. In secretory cells, 21 of the top 30 ionocyte markers show a significant positive correlation, whereas only 1 of 30 basal cell markers was significant. *P* values: Wald test on linear regression models fit independently for each gene, no multiple hypothesis adjustment was performed since only 30 genes were tested for each panel (left, right).

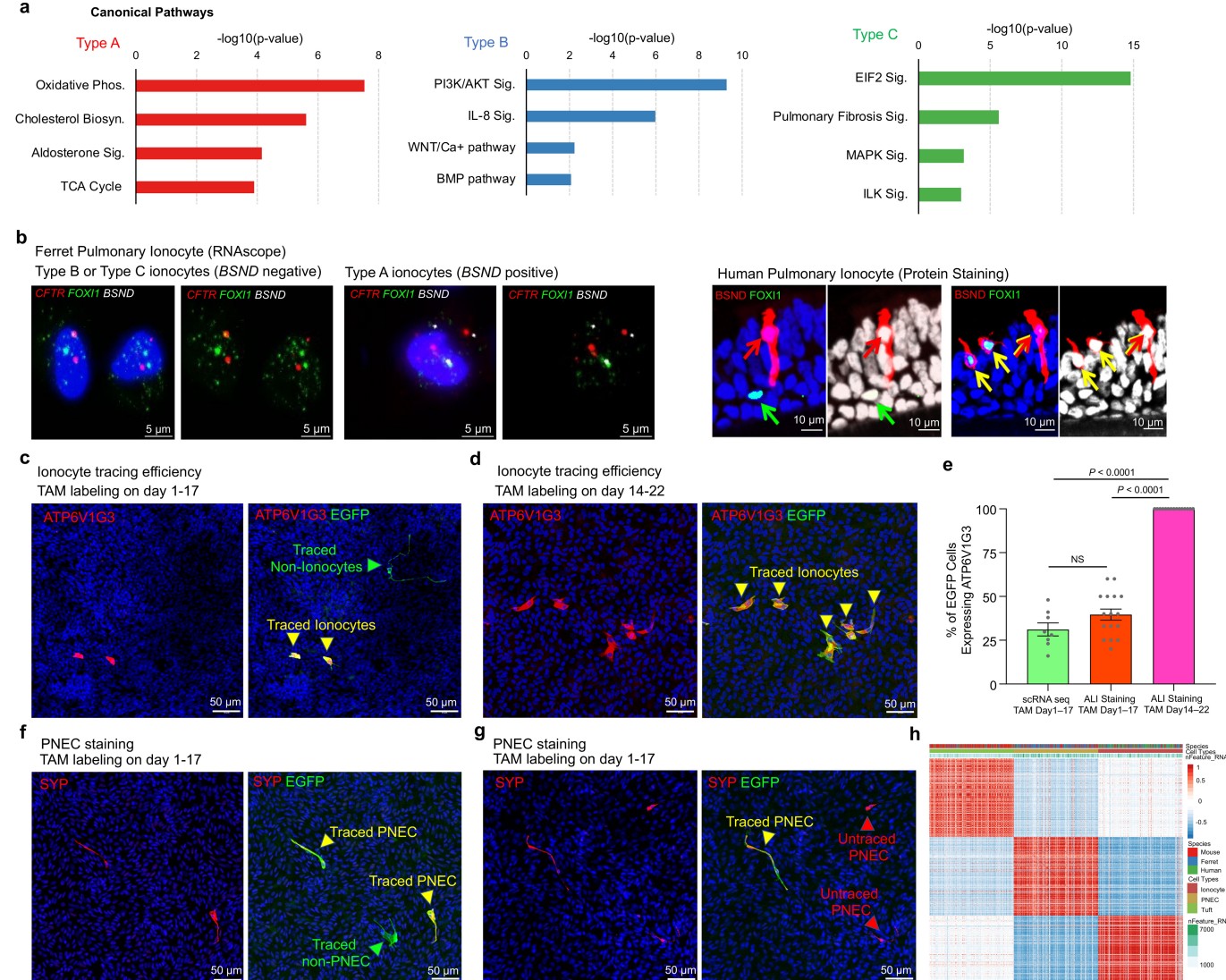

**Extended Data Fig. 10 | *FOXI1* lineage basal cells differentiate into multiple rare cell types in vitro. a**, Ingenuity Pathway Analysis (IPA) of differential gene expression data from ionocyte subtypes. The charts represent the top significantly associated canonical pathways. *P* value was calculated using a right-tailed fisher's exact test. **b**, (Left) RNAscope validation of Type A, Type B or Type C ionocytes from a cytospun ferret ALI culture. Representative images of n = 10 ionocytes. (Right) Immunofluorescence validation of Type A, Type B or Type C ionocytes in a section of human ALI culture. BSND positive ionocytes are Type A subtype, some of which co-express FOXI1 (yellow arrows) or lack FOXI1 expression (red arrows). BSND negative ionocytes that express FOXI1 (green arrows) are Type B or Type C subtype. Intermediated expression of FOXI1 in BSND positive cells is marked by overlapping yellow and red arrows. Right panels show nuclei in white to demarcate the cell more clearly. Representative images of n = 7 ALI cultures. **c,d**, Lineage tracing of *FOXI1*-Cre^ERT2^::ROSA-TG ALI culture following hydroxytamoxifen (TAM) treatment on (c) day 1–17 or (d) day 14–22 produce different proportions of EGFP+ATP6V1G3+ ionocytes and EGFP+ATP6V1G3− other rare cell types. Yellow and green

arrowheads indicate traced ionocytes or traced non-ionocytes, respectively. Representative image from n = 16 independent samples from TAM day 1–17 group and n = 15 independent samples from TAM day 14–22 group. **e**, Percentage of *FOXI1*-Cre^ERT2^ traced EGFP+ cells that also expressed ATP6V1G3 from scRNA-seq and the two TAM treatment protocols (Mean ± s.e.m.; n = 8 independent samples in scRNA-seq group, n = 16 independent samples from 4 different animals in ALI TAM Day 1–17 group, n = 15 independent samples from 4 different animals in ALI TAM Day 14–22 group; *P* value determined by one-way ANOVA and Tukey's multiple comparisons test). **f,g**, Lineage tracing of PNECs in actively differentiating *FOXI1*-Cre^ERT2^::ROSA-TG basal cells treated with TAM on day 1–17 of ALI culture. Cultures were immunostained with PNEC marker SYP on day 21. Yellow, red, and green arrowheads indicate traced PNEC, untraced PNEC, or traced non-PNEC, respectively. Representative image of 11 independent cultures. **h**, Cell-cell Pearson correlation coefficient matrix (r, color bar) across species (human, mouse, ferret) ordered by cluster assignment (tuft, PNEC, ionocyte).

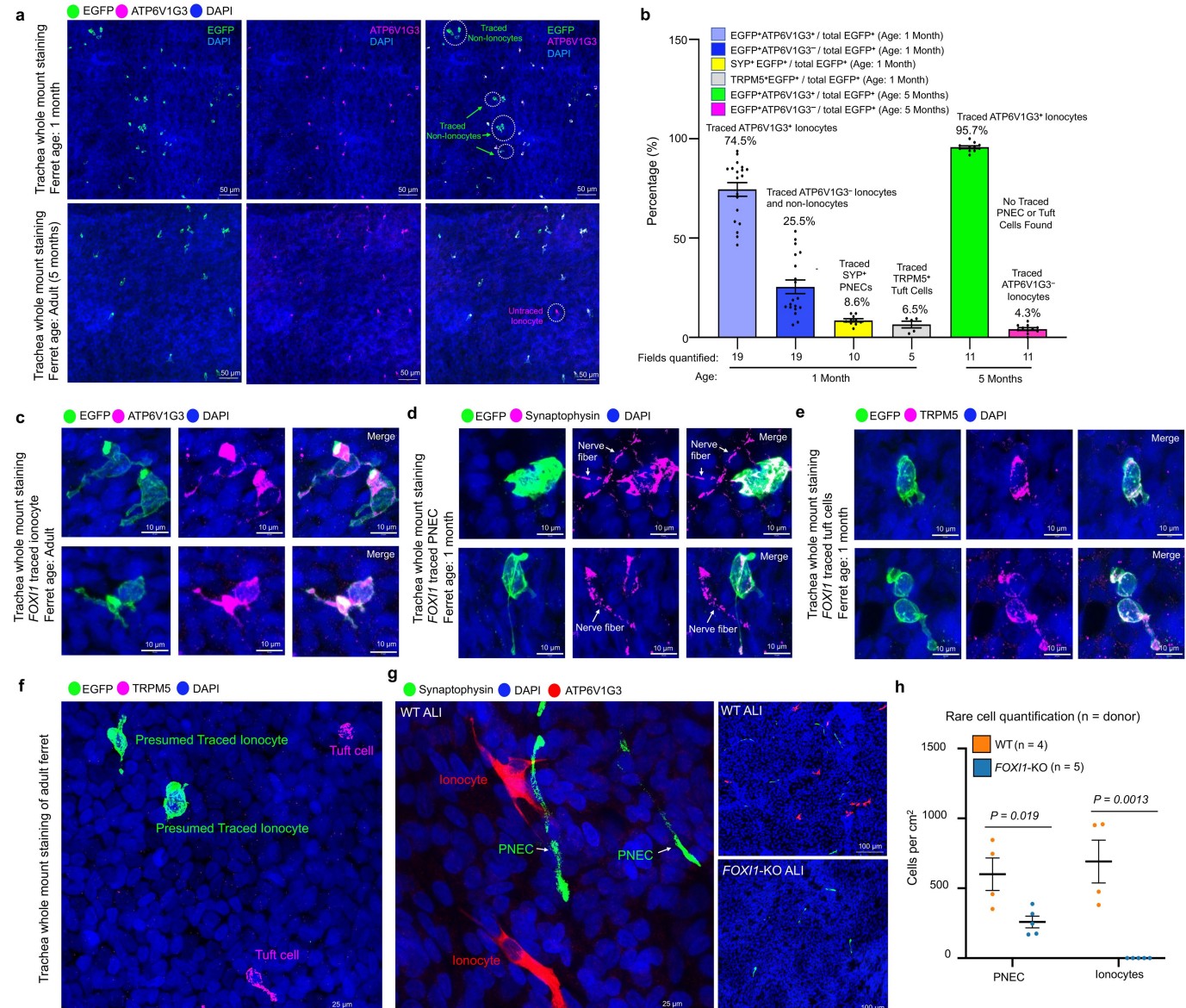

**Extended Data Fig. 11 | *FOXI1*-lineage rare cell progenitor has the capacity to generate ionocytes, PNECs, and tuft cells in vivo. a**, Confocal images of lineage-traced trachea following 5 sequential tamoxifen injections in *FOXI1*-Cre[ERT2]::ROSA-TG ferrets at 1 or 5 months of age (harvested one week after the last tamoxifen injection). Traced EGFP+ATP6V1G3+ ionocytes and EGFP+ATP6V1G3− non-ionocyte were identified in one month old trachea (upper panel). In adult *FOXI1*-Cre[ERT2]::ROSA-TG ferret trachea, only traced EGFP+ATP6V1G3+ ionocytes were observed (lower panel). Representative images from n = 3 independent ferrets were traced in each group. **b**, Percentage of *FOXI1*-Cre[ERT2] labeled EGFP+ cells in the trachea for the indicated phenotypic classifications following colocalization with markers for ionocytes (ATP6V1G3), PNEC (synaptophysin/SYP), and tuft cells (TRPM5) at 1 and 5 months of age (Mean ± s.e.m.; n = 3 ferrets quantified for each age; each datapoint represents quantification from a 1.41 mm² area for the number of fields indicated). Lineage tracing of 1 month old *FOXI1*-Cre[ERT2]::ROSA-TG ferrets demonstrated *FOXI1*-lineage cells (EGFP+) were composed of ionocytes (74.5%), PNEC (8.6%) and tuft cells (6.5%). Lineage tracing of 5 month old (adult) *FOXI1*-Cre[ERT2]::ROSA-TG ferrets demonstrated 95.7% of *FOXI1*-lineage cells (EGFP+) were ATP6V1G3+ ionocytes. The 4.3% EGFP+ATP6V1G3− cells in adult animals is consistent with a

small fraction of mature ionocytes (~4%) expressing no *ATP6V1G3* by scRNAseq. **c**, Maximum intensity projection of traced EGFP+ATPV61G3+ pulmonary ionocytes from adult *FOXI1*-Cre[ERT2]::ROSA-TG ferret trachea. Representative image from n = 3 ferrets. **d**, Maximum intensity projection of traced EGFP+SYP+ PNECs from one month old *FOXI1*-Cre[ERT2]::ROSA-TG ferret trachea. Images show sensory nerves associated with PNECs. Representative image from n = 3 ferrets. **e**, Maximum intensity projection of traced EGFP+TRPM5+ tuft cells from one month old *FOXI1*-Cre[ERT2]::ROSA-TG ferret trachea. Representative image from n = 3 ferrets. **f**, Representative image of adult *FOXI1*-Cre[ERT2]::ROSA-TG ferret trachea following lineage tracing demonstrating the lack of TRPM5 staining in EGFP+ presumed ionocytes. Representative image from n = 3 ferrets. **g**, Representative confocal image of ionocytes (ATP6V1G3+) and PNECs (SYP+) in epithelia from WT and *FOXI1*-KO air-liquid interface (ALI) cultures. n = 4 ferrets (WT), n = 5 ferrets (*FOXI1*-KO). **h**, Frequency of ionocytes (ATP6V1G3+) and PNECs (SYP+) in epithelia from *FOXI1*-KO and WT ALI cultures (n = ferret donors, 3-5 independent cultures quantified from each donor and averaged). Graphs show the mean ± s.e.m. Statistical significance was determined by two-tailed Student's *t*-test.

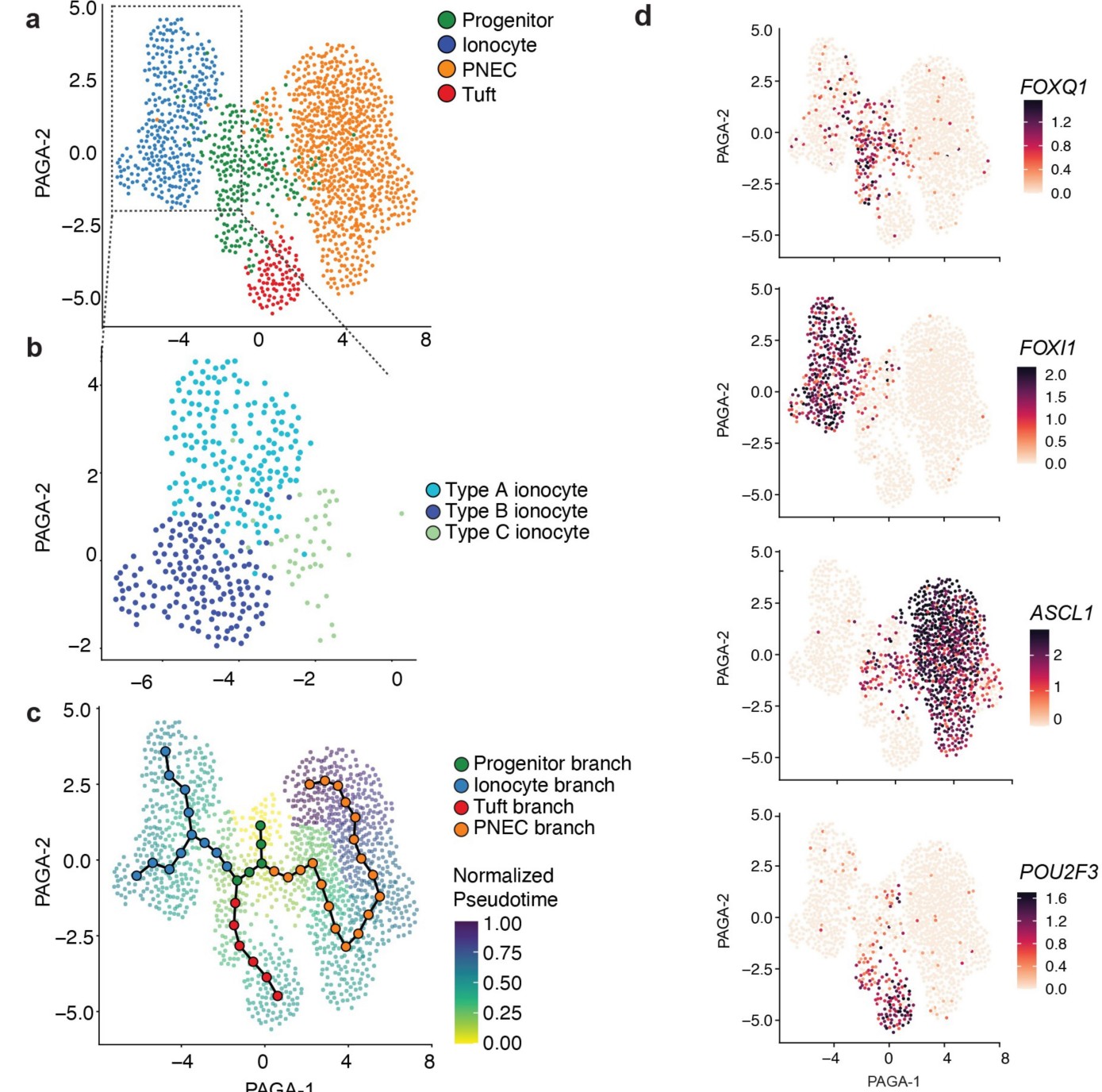

**Extended Data Fig. 12 | Computational lineage inference of rare ferret airway epithelial cell types using pseudotime. a**, Partition-based graph abstraction (PAGA) embedding of 1,497 rare airway epithelial cells (PNEC, tuft cells, ionocytes) and putative progenitor group (color legend), all previously identified by unsupervised clustering (Fig. 3 and Fig. 5). **b**, Subset of ionocytes, colored by their assignment to novel ionocyte subtypes, also by unsupervised clustering (Fig. 4). **c-d**, PAGA embedding of 1,497 rare airway epithelial cells, colored pseudo-time coordinate (c), and lineage transcription factors for each rare cell type (d). Trajectory (large points) and branches (color legend) in (c) were fit using elastic principal graphs.

# Reporting Summary

## Statistics

For all statistical analyses, confirm that the following items are present in the figure legend, table legend, main text, or Methods section.

| n/a | Confirmed | |
|---|---|---|
| ☐ | ☒ | The exact sample size (*n*) for each experimental group/condition, given as a discrete number and unit of measurement |
| ☐ | ☒ | A statement on whether measurements were taken from distinct samples or whether the same sample was measured repeatedly |
| ☐ | ☒ | The statistical test(s) used AND whether they are one- or two-sided *Only common tests should be described solely by name; describe more complex techniques in the Methods section.* |
| ☐ | ☒ | A description of all covariates tested |
| ☐ | ☒ | A description of any assumptions or corrections, such as tests of normality and adjustment for multiple comparisons |
| ☐ | ☒ | A full description of the statistical parameters including central tendency (e.g. means) or other basic estimates (e.g. regression coefficient) AND variation (e.g. standard deviation) or associated estimates of uncertainty (e.g. confidence intervals) |
| ☐ | ☒ | For null hypothesis testing, the test statistic (e.g. *F*, *t*, *r*) with confidence intervals, effect sizes, degrees of freedom and *P* value noted *Give P values as exact values whenever suitable.* |
| ☐ | ☒ | For Bayesian analysis, information on the choice of priors and Markov chain Monte Carlo settings |
| ☐ | ☒ | For hierarchical and complex designs, identification of the appropriate level for tests and full reporting of outcomes |
| ☐ | ☒ | Estimates of effect sizes (e.g. Cohen's *d*, Pearson's *r*), indicating how they were calculated |

*Our web collection on statistics for biologists contains articles on many of the points above.*

## Software and code

Policy information about availability of computer code

| Data collection | Zen software (Immunofluorescence imaging collection): version 2.3; Acquire and Analyze 2.3 (electrophysiology data collection); Novaseq control software v1.7.5 (scRNA seq data collection); Bio-Rad Real-Time PCR (Relative gene expression data collection): version 2.3; Zeiss software FRAP (airway viscosity data collection): version 2.3. |
|---|---|
| Data analysis | Zen software (Immunofluorescence imaging analysis): version 2.3; Acquire and Analyze 2.3 (electrophysiology data analysis); Metamorph (imaging analysis): version 7.0; Ingenuity Pathway Analysis (IPA): version 01-21-03; GraphPad Prism 9; R 4.2.0; Image J: version 2.3.0/1.53q; PMOD (Mucociliary Clearance analysis): version 4.2; Bio-Rad Real-Time PCR: version 2.3; Zeiss software FRAP: version 2.3; BD FACSDiva 8.0.1 Software; Kallisto toolkit: version 0.48; DropletUtils package: version 1.20.0; R package 'brms': version 2.17.0, MAST: version 1.22.0; EIPiGraph.R: version 1.0.0, R package "prop.test": version 3.6.2. R markdown scripts are available upon request. Customized code for the computation of cross-species similarity between rare cells is available on GitHub (https://github.com/yifand64/ferret_ionocyte). |

For manuscripts utilizing custom algorithms or software that are central to the research but not yet described in published literature, software must be made available to editors and reviewers. We strongly encourage code deposition in a community repository (e.g. GitHub). See the Nature Portfolio guidelines for submitting code & software for further information.

## Data

Policy information about availability of data

All manuscripts must include a data availability statement. This statement should provide the following information, where applicable:
- Accession codes, unique identifiers, or web links for publicly available datasets
- A description of any restrictions on data availability
- For clinical datasets or third party data, please ensure that the statement adheres to our policy

Single-cell sequencing data is available in GEO, accession (GSE233654): https://www.ncbi.nlm.nih.gov/geo/query/acc.cgi?acc=GSE233654.
Data for Figures and Extended Data Figures are available as Source Data.
The publicly available Genome assembly MusPutFur1.0 (https://www.ncbi.nlm.nih.gov/datasets/genome/GCF_000215625.1/) was used as ferret reference genome in this study. The publicly available Genome assembly GRCh38.p13 (https://www.ncbi.nlm.nih.gov/datasets/genome/GCF_000001405.39/) was used for identification of unannotated ferret genes by using human orthologs. The publicly available Genome assembly GRCm39 (https://www.ncbi.nlm.nih.gov/datasets/genome/GCF_000001635.27/) was used for identification of unannotated ferret genes by using mouse orthologs.

## Human research participants

Policy information about studies involving human research participants and Sex and Gender in Research.

| | |
|---|---|
| Reporting on sex and gender | None |
| Population characteristics | None |
| Recruitment | None |
| Ethics oversight | None |

Note that full information on the approval of the study protocol must also be provided in the manuscript.

# Field-specific reporting

Please select the one below that is the best fit for your research. If you are not sure, read the appropriate sections before making your selection.

☒ Life sciences ☐ Behavioural & social sciences ☐ Ecological, evolutionary & environmental sciences

For a reference copy of the document with all sections, see nature.com/documents/nr-reporting-summary-flat.pdf

# Life sciences study design

All studies must disclose on these points even when the disclosure is negative.

| | |
|---|---|
| Sample size | No predetermined sample size calculation was performed in this study. Based on data we collected, we think the number of samples in each group were adequate to confirm our findings. Regarding to scRNA-seq experiments, we have excellent cell diversity coverage of ferret epithelial cell of interest and already have shown in our study that all cells are well covered by each ferret donor in each group. Given the complexity, size and reproducibility within each figures, we felt that the number of samples shown was appropriate to address the questions in this study. |
| Data exclusions | No data were excluded from analysis. |
| Replication | All attempts of replication were successful as shown in our Figures. |
| Randomization | Samples/organisms were allocated into experimental groups based on genotype. |
| Blinding | Blinding was not necessary in this study since assays used unbiased quantification methods. |

# Reporting for specific materials, systems and methods

We require information from authors about some types of materials, experimental systems and methods used in many studies. Here, indicate whether each material, system or method listed is relevant to your study. If you are not sure if a list item applies to your research, read the appropriate section before selecting a response.

## Materials & experimental systems

| n/a | Involved in the study |
|-----|----------------------|
| ☐ | ☒ Antibodies |
| ☐ | ☒ Eukaryotic cell lines |
| ☒ | ☐ Palaeontology and archaeology |
| ☐ | ☒ Animals and other organisms |
| ☒ | ☐ Clinical data |
| ☒ | ☐ Dual use research of concern |

## Methods

| n/a | Involved in the study |
|-----|----------------------|
| ☒ | ☐ ChIP-seq |
| ☐ | ☒ Flow cytometry |
| ☒ | ☐ MRI-based neuroimaging |

# Antibodies

**Antibodies used**

Primary Antibodies Used (dilution factor; catalogue number; company):
anti NKA (1/100-1/300; a5, DSHB UIOWA, dshb.biology.uiowa.edu )
anti CFTR (1/100-1/300; CFTR antibody 596, cftrantibodies.web.unc.edu)
anti-Keratin 5 (1/500; 905501, Biolegend)
anti-TRPM 5 (1/300; ACC-045, Alamone)
anti-SYP (1/200; sc-17750, Santa Cruz Biotechnology)
anti-acetylated Tublin (1/1000, T7451, Sigma Aldrich)
anti-ATP6V1G3 (1/500, HPA028701, Sigma Aldrich)
anti-Ki-67 (1/500, 14-5698-82, eBioscience)
anti-BSND (1/500; ab196017, Abcam)
anti-FOXI1 (1/500; ab20454, Abcam)
anti-EGFP (1/300; ab13970, Abcam)
anti-p63 (1/300; Clone Poly6190, Stemcell technology )
anti-Muc5B (1/300; HPA008246,  Sigma)
anti-Muc5AC (1/300; ab3649, Abcam)
Secondary Antibodies Used (dilution factor; catalogue number; company):
Alexa Fluor 647 donkey anti-mouse IgG (1/250, A31571, Molecular Probes)
Alexa Fluor 488 donkey anti-goat IgG (1/250, A11055, Invitrogen)
Alexa Fluor 488 donkey anti-chicken IgG (1/250, 703-546-155, Jackson ImmunoResearch)
Alexa Fluor 488 donkey anti-rabbit IgG (1/250, A21206 Invitrogen)
Alexa Fluor 568 donkey antigoat IgG (1/250, A-11057, Jackson ImmunoResearch)
Alexa Fluor 647 donkey Anti-Rabbit IgG (1/250,711-606- 152, Jackson ImmunoResearch)
Alexa Fluro 555 donkey Anti-mouse IgG (1/250, A31570, Life Technologies).

**Validation**

anti-NKA (Confirmed Species reactivity: Avian, Drosophila, Fish, Frog, Honeybee, Human, Insect, Mackerel, Mammal, Mosquito, Zebrafish; Applications: FFPE, Immunofluorescence, Immunohistochemistry, Immunoprecipitation, Western Blot; Validation: Manufacturer - https://dshb.biology.uiowa.edu/a5)
anti-CFTR (Confirmed Species reactivity: Human; Applications: Immunofluorescence, Western Blot; Validation: Manufacturer - https://cftrantibodies.web.unc.edu/cell-staining/; https://cftrantibodies.web.unc.edu/western-blot/)
anti-Keratin 5 (Confirmed Species reactivity: Human; Applications: Immunohistochemistry; Validation: Manufacturer - https://www.biolegend.com/Files/Images/media_assets/pro_detail/datasheets/905501-keratin-5-polyclonal-antibody-purified-10956-IFU-Rev-7.pdf?v=20220914082417)
anti-TRPM 5 (Confirmed Species reactivity: Human, mouse, rat; Applications: Immunofluorescence, Western Blot, Immunohistochemistry; Validation: Manufacturer - https://www.alomone.com/p/anti-trpm5/ACC-045)
anti-SYP (Confirmed Species reactivity: Human, mouse, rat; applications: Immunofluorescence, Western Blot, Immunohistochemistry, Elisa; Validation: Manufacturer - https://www.scbt.com/p/syp-antibody-h-8?gclid=CjwKCAjwt52mBhB5EiwA05YKo4ePEOMyMIrqZzCaYKzlU_gx2A-wSZh0uQLJoBeVyQfX5oHfjp8eyxoCrcwQAvD_BwE )
anti-acetylated Tublin (Confirmed Species reactivity: bovine, frog, invertebrates, human, hamster, mouse, protista, pig, monkey, chicken, rat, plant;  Applications: electron microscopy, immunohistochemistry, Immunofluorescence, western blot; Validation: Manufacturer - https://www.sigmaaldrich.com/US/en/product/sigma/t7451?gclid=CjwKCAjwt52mBhB5EiwA05YKowLtMY3KWAxOJZNqKiqoIubAK-awWidzUJv_yNP1GRyOl8sn0wqYNxoCs-UQAvD_BwE&gclsrc=aw.ds)
anti-ATP6V1G3 (Confirmed Species reactivity: Human; Applications: immunohistochemistry; Validation: Manufacturer - https://www.sigmaaldrich.com/US/en/product/sigma/hpa028701)
anti-Ki-67 (Confirmed Species reactivity: Mouse; Applications: immunocytochemistry; Validation: Manufacturer - https://www.thermofisher.com/antibody/product/Ki-67-Antibody-clone-SolA15-Monoclonal/14-5698-82)
anti-BSND (Confirmed Species reactivity:  Mouse, Rat, Human; Applications: immunohistochemistry, western blot; Validation: Manufacturer - https://www.abcam.com/products/primary-antibodies/bsnd-antibody-epr14270-c-terminal-ab196017.html)
anti-FOXI1 (onfirmed Species reactivity: Human; Applications: Western Blot, Immunohistochemistry; Validation: Manufacturer - https://www.abcam.com/products/primary-antibodies/foxi1-antibody-ab20454.html )
anti-EGFP (Confirmed Species reactivity: Species independent; Applications: Western Blot, Immunohistochemistry; Validation: Manufacturer - https://www.abcam.com/products/primary-antibodies/gfp-antibody-ab13970.html)
anti-p63 (Confirmed Species reactivity: human; Applications: immunohistochemistry, Immunofluorescence, western blot; Validation: Manufacturer -  https://www.stemcell.com/products/anti-human-p63-deltan-antibody-clone-poly6190.html )
anti-Muc5B (Confirmed Species reactivity: human; Applications:  immunohistochemistry, Immunofluorescence; Validation: Manufacturer -  https://www.sigmaaldrich.com/US/en/product/sigma/hpa008246?gclid=Cj0KCQjw2qKmBhCfARIsAFy8buIYdkjTn8ZEHkC66iUYoh13yOwm_A8Q8SUGxYNVfWEvD-Ya4q9GxAwaAkHjEALw_wcB&gclsrc=aw.ds)
anti-Muc5AC (Confirmed Species reactivity: Mouse, Rat, Human; Applications: immunohistochemistry, Immunofluorescence;

# Eukaryotic cell lines

Policy information about cell lines and Sex and Gender in Research

| Cell line source(s) | Primary airway basal cells were derived from WT, FOXI1-CreERT2::ROSA-TG, FOXI1-KO, CFTR-KO, and FOXI1-CreERT2::CFTR-L/L ferrets. |
|---|---|
| Authentication | All primary airway basal cell lines were validated for genotype and performed from multiple donor as specified in the text. |
| Mycoplasma contamination | All primary cells tested negative for mycoplasma contamination prior to experimentation. |
| Commonly misidentified lines (See ICLAC register) | n/a |

# Animals and other research organisms

Policy information about studies involving animals; ARRIVE guidelines recommended for reporting animal research, and Sex and Gender in Research

| Laboratory animals | Transgenic ferret models were used in this study including: WT ferrets, FOXI1-CreERT2::ROSA-TG ferrets, FOXI1-KO ferrets, CFTR-KO ferret, FOXI1-CreERT2::CFTRL/L ferrets and CFTR-G551D CF ferrets.<br>MCC studies were conducted on 5-6 month old FOXI1-CreERT2::CFTR-L/L ferrets.<br>MCC studies were conducted on 6-38 month old WT ferrets and FOXI1-KO ferrets.<br>MCC studies were conducted on 4-6 month old CFTR-G551D CF ferrets.<br>Tissue electrophysiology studies were conducted on 18-36 month old FOXI1-KO ferrets  and age match WT ferrets.<br>Ionocyte Lineage tracing studies were conducted on 1-5 month old FOXI1-CreERT2::ROSA-TG ferrets.<br>Proximal airway stem cells differentiation studies were conducted on 1-5 month old FOXI1 KO ferrets strian and age match WT ferrets, FOXI1-CreERT2::ROSA-TG ferrets, FOXI1-CreERT2::CFTR-L/L ferrets, CFTR-KO ferret.<br>Ionocyte localization studies (trachea whole mount staining studies) were conducted on 6-12 month old WT ferret and CFTR-G551D CF ferrets. |
|---|---|
| Wild animals | This study did not involve wild animals. |
| Reporting on sex | Sex was not considered as a variable in our studies since it is hard to control equal distribution of sexes with transgenic ferrets. |
| Field-collected samples | This study did not involve samples collected from field. |
| Ethics oversight | All animal experimentation was approved by the Institutional Animal Care and Use Committee of the University of Iowa (Protocol: 0031945). |

Note that full information on the approval of the study protocol must also be provided in the manuscript.

# Flow Cytometry

## Plots

Confirm that:

☐ The axis labels state the marker and fluorochrome used (e.g. CD4-FITC).

☐ The axis scales are clearly visible. Include numbers along axes only for bottom left plot of group (a 'group' is an analysis of identical markers).

☐ All plots are contour plots with outliers or pseudocolor plots.

☒ A numerical value for number of cells or percentage (with statistics) is provided.

## Methodology

| Sample preparation | As indicated in the methods, fully differentiated ferret airway epithelia ALI cultures were dissociated using Accumax (Stem Cell Technologies) followed by DNase treatment. Cells were filtered through a 20 μM strainer and pelleted in 0.04% BSA PBS at 500g for 10 min. Nonviable dead cells were removed by using MACS Dead Cell Removal Kit following 10X Genomics recommendations (Document CG00039). |
|---|---|
| Instrument | Becton Dickinson Aria II |

| Software | BD FACSDiva 8.0.1 Software |
|---|---|
| Cell population abundance | Lineage traced pulmonary ionocytes (EGFP+) were <1% of total population. |
| Gating strategy | Cells were identified based on FSC and SSC gating. Tomato and EGFP positive epithelial cells were identified as positive based based on comparison to non-reporter ferret airway basal cells.  Single cells were identifies based on forward scatter and forward pulse width. |

☒ Tick this box to confirm that a figure exemplifying the gating strategy is provided in the Supplementary Information.

