## [Peer Review File · Nature]

Manuscript Title: Transgenic Ferret Models Define Pulmonary Ionocyte Diversity and Function

Reviewer Comments & Author Rebuttals

Reviewer Reports on the Initial Version:

Referees' comments:

Referee #1 (Remarks to the Author):

Transgenic ferrets were produced in which FOXI1 was deleted resulting in loss of ionocyte. FOXI1-CreERT2 was used for lineage tracing. In vivo and in vitro studies demonstrated that ionocytes play primary roles in controlling airway liquid volume mucociliary clearance and other known CFTR-associated processes and phenocopy findings in CFTRG551D. Numbers of ionocytes and associated cell lineages were determined in the ferret airways. Single cell transcriptomics were used to predict distinct ionocytes subtypes from in vitro cultures. Data support the concept that ionocytes share a common airway epithelial progenitor with other rare airway cells, (tuft, and neuroendocrine cells) and that the numbers of ionocytes are influenced by sensing airway surface liquid (ASL). Taken together, the work supports a critical role of ionocyte and ionocyte progenitors in CFTR related lung biology.

Originality and significance:

This is a landmark paper addressing a number of critical questions regarding the control of airway fluid volume and mucociliary clearance. The work is of clear relevance to the pathogenesis of CF and other chronic lung diseases. Ionocytes, a relatively rare subset of lung epithelial cells, were previously shown to be highly enriched in CFTR mRNA. This previous work raised a major controversy regarding the potential role of these "rare cells" including their primary role in CF pathogenesis, since CFTR is also expressed in other airway epithelial cells, albeit at lower levels. The present study in a new animal model (with lung structure more similar to that in humans), convincingly shows that deletion of FOXI1 expressing cells phenocopied findings in ferrets bearing CFTRG551D mutations. The work carefully identified the role of ionocytes (or other FOXI1-lineage traced cells) in the regulation of CF-associated lung epithelial cell functions, including mucociliary clearance (MCC), pH control, ASL, and chloride transport. In vitro and in vivo experimental data support the major conclusions of the work.

Data and Methodology:

Appropriate in vitro and in vivo experiments, using state-of-the-art approaches support the major conclusions. Data quality and documentation are rigorous. The paper is logically presented, figures, imaging data, and quality are excellent. Statistical approaches are appropriate.

Conclusions:

Are generally strongly supported by experimental data. Multiple in vivo and in vitro approaches provide orthogonal validity to the work. A critical issue regarding the major conclusions relates to the interpretation of the impact of FOXF1 deletion data. Since the authors data demonstrate that FOXI1 gene deletion may alter the numbers and functions of other epithelial cells, including tuft and NE cells, the authors should discuss that progenitors of all these cell types and their functions may be altered after deletion of FOXI1-Cre.

Specific suggestions for improvement:

This leads to the major limitation of data presentation in what is otherwise a comprehensive study – the lack of in vivo lineage tracing data to support the in vitro observation of a common progenitor of rare cell types in the lung with expresses FOXI1. At minimum, data from a lineage

trace with early labeling of FOXI1+ cells (presumably a subset of basal cells?) during lung development leading to labeling of tuft, NE, and ionocytes later in life is needed to validate the lineage relationships suggested by the ALI culture. Better still would be labeling of FOXI1+ cells at 2-3 time points of airway cell differentiation during development or following an airway injury (naphthalene?) to demonstrate the numerical capacity of FOXI1+ cells to generate these rare cell types. Presuming that the in vitro data is supported by these in vivo lineage studies, together these data would change the paradigm of rare airway cell differentiation and suggest a new therapeutic target in the FOXI1+ rare airway progenitor given the key roles of NE, ionocyte, and tuft cells in diverse lung diseases.

Other specific comments:

- 1) Page 7 - The CFTR KO experiments would benefit from additional discussion of the lineage tracing results. Since FOXI1-CreT is likely expressed in progenitors from which a number of different airway cells are derived, can these non-ionocytes cells derived from FOXI1+ lineage contribute to the phenotype? Which submucosal gland regions, ducts, glands, or cells and in which cells (mucus, serous) were lineage labeled and expressed high levels of CFTR?
- 2) Page 7 - The finding that dehydration and ASL-induced ionocyte influenced ionocyte activity and number are important findings. Can the authors comment on the mechanism for sensing? Is it osmolality or changes in ion concentrations or other mechanical stimuli, for example, shear stress and/or ciliary activity?
- 3) Page 9 - Can the authors comment on the potential impact of disrupting tuft and NE cells on the ionocyte deficient phenotype?
- 4) Page 10 - Are genetic expression changes in other epithelial cell types, for example, basal, secretory, and ciliated cells likely to be regulated or contribute to the ionocyte changes in airway fluid homeostasis? Can changes in gene expression in other epithelial cell types alter ASL?
- 5) Page 12 - The pulse-seq experiments suggested a common progenitor for ionocytes, tuft, and NE cells; does in vivo lineage labeling support this or as the authors suggest there are findings related to in vitro conditions and potential plasticity of shared progenitors?
- 6) Page 14 - Regarding hyperosmolar stress and numbers of lineage trace ionocytes; were ion and lineage conditions maintained during the stress? Can the authors comment on the signals that are inducing increased ionocytes lineage labeled cells?

References are appropriate.

Clarity and content are generally lucid and logical.

Referee #2 (Remarks to the Author):

Summary of the key results:

In the present study, authors generated several transgenic ferret models including FOXI1-KO, FOXI1-CreERT2(lineage tracing ionocytes), and CFTR conditional KO (CFTRL/L) ferrets to characterize pulmonary ionocyte function in the ferret, as ferret develops a CF lung disease phenotype like humans. Using these models, authors demonstrate that ionocyte plays important roles in regulating ASL volume and pH; and mucociliary clearance (MCC). Single-cell profiling of lineage-traced ionocytes identified three ionocyte subtypes and a common rare progenitor for ionocytes, tuft cells and pulmonary neuroendocrine cells (PNECs). Collectively, the work provides new insights of the ionocytes heterogeneity and function; supports the important roles of ionocyte in CFTR related function and CF related phenotypes.

Originality and significance:

The originality and significance of the study were evident as highlighted in following points:

- Convincing evidence from in vivo and in vitro data support the important roles of FOXI1-expressing pulmonary ionocytes regulate ASL volume, pH and viscosity.
- Pulmonary ionocytes directly transport anions in the proximal airway in a CFTR-dependent fashion with direct relevance to CF airway disease.
- The ferret genetic approaches can be used to study the functional roles of individual cell type contributions to other genetic disease states.
- Single cell analysis revealed ionocyte heterogeneity; identified three subtypes and further cross validated via RNAscope.
- Osmolarity level influences basal cell specification of ionocyte.
- "rare cells" is not that rare; the proportion of FOXI1-CreERT2 lineage-labeled EGFP+ ionocytes indicated that single-cell processing significantly underestimated the frequency of ionocytes in proximal airway epithelia.

Data and Methodology:

Both experimental data and single cell data are in high quality. Methodologies and approaches are appropriate in general. The usage the precise conditional genetics and fate mapping in ferrets to dissect an individual cell type function is novel and state-of-the-art. The application of statistics in experimental design and single cell analysis are appropriate. A critical concern is that single cell data and customized code for the study are currently not available for reviewer assessment. Another concern is that the interspecies comparison of human, ferret, and mouse using publicly available scRNA-seq datasets. More details are needed about how well they integrated data and removed the batch effect since this will largely affect the results and conclusion (details see "Specific suggestions for improvement" section below).

Conclusions:

Major conclusions are strongly supported by experimental data in general, e.g., convincing evidence support that FOXI1-expressing pulmonary ionocytes are essential for regulating ASL volume, pH and effective MCC; osmolarity level influences basal cell specification of ionocyte; and pulmonary ionocytes preformed CFTR-dependent anions transport in cell-autonomous manner in the proximal airway with direct relevance to CF airway disease.

The role of ionocytes in cystic fibrosis and its extremely rare abundance has raised a lot of debate and discussions. The present study observed that single-cell data significantly underestimated the frequency of ionocytes in proximal airway epithelia, suggesting the rare ionocyte is not actually rare. Authors further estimate that ionocyte damage during 10X capture may raise ambient RNA for genes highly expressed in ionocytes. More evidence is needed to support this important finding (details see "Specific suggestions for improvement" section below).

"FOXI1+ common rare cell progenitor" is a novel and important observation, further characterization and validation using bioinformatics and experimental approaches are needed to support the conclusion (details see "Specific suggestions for improvement" section below).

Specific suggestions for improvement:

Authors preformed interspecies comparisons of rare cell types transcriptomes using publicly available scRNA-seq datasets and concluded that the human pulmonary ionocyte was significantly more similar to ferret than to mouse ionocytes.

- From Fig5B UMAP, one can clearly see the batch difference by species. More details are needed about how well they integrated data and removed the batch effect since this will largely affect the results and conclusion.
- Authors can show how well the integration worked using visualizations and standard metrics such

as kBET or LISI.

- Can authors comment on why ferret ionocytes are transcriptionally more similar to that of human ionocytes; while ferret Tuft and PNE are more similar to corresponding cell types in mouse (Fig 5I)

“FOXI1+ common rare cell progenitor” is a novel and important observation, little is known about the paths of the common progenitor takes to differentiate into these rare cell types, further characterization and validation using bioinformatics and experimental approaches are needed to support the conclusion.

- For example, authors could apply scVelo to reconstruct the lineage relationships between the common rare cell progenitor and ionocyte, tuft and PNE cells to infer their developmental trajectories.

- Lineage tracing of FOXI1+ rare cell progenitor and follow its differentiation trajectory through lung development.

- Can authors comment on the different observation and conclusions from the previous scRNA-seq, lineage and CRISPR knockout studies suggesting that novel tuft-like cells are the likely progenitor of PNE and ionocytes; and epithelial CFTR activity is not reduced by loss of ionocytes. (Goldfarbmuren et al).

Authors pointed out that single-cell data significantly only captured <3% of ionocytes in proximal airway epithelia, and the ionocyte damage during 10X capture increased ambient RNA for genes highly expressed in ionocytes. This is an important finding to address a controversial research issue. More evidence is needed to support this important finding.

- If the loss of ionocytes occur during single cell sample preparation, it is important to show that a higher frequency of ionocytes present in in ALI prior to single cell sequencing.

- Extended Data Fig. 8A-C showed a significant correlation of CFTR, ASCL3, FOXI1 expression in secretory cell with the number of ionocytes detected in each ALI culture. Please provide the correlation plots for other cell types since ambient RNA in the environment would potentially influencing the barcode of all different cell types.

Page 9. Authors identified signature genes of ferret pulmonary ionocytes and discussed their functional relevance to CFTR related ion transporters. Some are selectively enriched in ionocyte including CFTR, CLCNKA, BSND, ATP6V1G3, ASCL3 and SCNN1G; while some are not selective including KCNK1, KCNQ1, KCNK5, KCNMA1, SCNN1A, ATP1B1, and SLC12A2.

References: appropriate.

Clarity and context: manuscript is well written with clarity and logic flow.

Referee #3 (Remarks to the Author):

This manuscript investigated the differentiation of ionocytes and the function of CFTR in ionocytes. As the authors summarize in the paper, data in this paper revealed that FOXI1- expressing pulmonary ionocytes functionally regulate ASL volume and pH, two hallmark defects in the CF airway that lead to increased mucus viscosity and impair MCC.

I was excited about these genetic ferret models, which will provide more information to understand the functions of rare CFTR-rich pulmonary ionocytes in cystic fibrosis. However, I don't find many conceptually advances over the current CFTR and ionocyte fields, but there is a thorough extension of prior concepts to CFTR in ionocytes. Previous studies using human and ferret airway epithelial cells have already shown that the proper differentiation of ionocytes is regulated by FOXI1 and NOTCH. And the function of ionocytes is required for ion transport and maintaining airway surface physiology, including mucus viscosity (Montoro D, Nature, 2018; Plasschaert LW, Nature, 2018, Shah, VS,, 2022).

1. Although there are some differences between mouse ionocytes and ionocytes in human and ferret, the experimental depletion of ionocytes in the mouse trachea perturbs epithelial physiology. What remains mysterious is the role of these rare cells in the pathogenesis of human CF. It will be more important to address the activity and function of ionocytes in the pathological settings since the authors have these very nice ferret models.
2. Their sc-RNAseq results show that there are three subtypes of ionocytes. This result is interesting but also expected given that ionocytes are found in different parts of the airways. sc-RNAseq UMAP shows that the Type A and Type subsets contain a similar proportion of ionocytes. Therefore, we can expect that the number of BSND negative ionocytes (Type B and Type C) will not lower than the number of BSND positive ionocytes (Type A). But the staining results showed more BSND positive ionocytes in human and ferret airway epithelial cells. Moreover, it seems that the differential expression of BSND between Type A and Type B ionocytes will not clearly distinguish Type A and Type B ionocytes. What is the relationship and function of these different subtypes?
3. The proposed models are largely dependent on the gene expression at the mRNA level. It is unclear whether these transporters and channels are expressed in heterogeneous levels or differential expression in ionocytes at different airway locations.

Author Rebuttals to Initial Comments:

Responses to Reviewers' Comments

We greatly appreciate the reviewers' thoughtful comments and questions regarding our manuscript. We believe that addressing these concerns has greatly strengthened the manuscript, enabling us to draw stronger conclusions about origins and functions of rare pulmonary ionocytes. We have added extensive additional data that strengthen our original findings: Revisions to Figure 4, five new Extended Data Figure 11-15, and Supplementary Video 7 (total of 32 new data panels). Additionally, we also provide four additional figures within this response to address reviewer comments.

Highlights of the new data:

- *In vivo* developmental fate mapping now confirms that a *FOXI1*-lineage rare cell progenitor gives rise to PNECs, tuft cells and pulmonary ionocytes. This occurs during postnatal airway development in neonatal ferrets, but not in adult animals at homeostasis.
- Regional ionocyte mapping at different levels of the airway and submucosal glands in both wild-type and cystic fibrosis (CF) ferrets. These studies demonstrate ionocyte expansion in the intrapulmonary bronchial submucosal glands of cystic fibrosis (CF) ferrets as compared to wild-type ferrets.
- Expanded osmotic stress studies in ALI cultures demonstrate that opposing forces on ASL volume, induced by hyperosmotic or hypoosmotic conditions, signal basal cells to produce greater numbers of specific ionocyte subtypes when ASL is over hydration or dehydration.
- Tracheal short-circuit current on wildtype and *FOXI1*-KO animals (not requested, but we believe is useful *in vivo* support to the *in vitro* culture studies).
- Localization of ionocyte enriched channels. These studies demonstrate the existence of what we have termed an "apical cap", a structure that enriches for both apical (CFTR) and basolateral (NKA / Na-K-ATPase) channels involved in salt movement across the epithelium. Notably, single ionocytes can have more than one apical cap, perhaps enhancing their capacity to direct salt and fluid movement.
- Experiments addressing the reduced 10X capture rate of ionocytes by demonstrating ionocyte viability following FACS.
- New analyses addressing the bioinformatic questions pertaining to cross species analysis, batch effects, ambient RNA, and pseudotime analysis.
- scRNA-seq trajectory analysis of ferret rare cell types demonstrate that Type-C ionocytes, which we show can replicate, are transcriptionally most similar to the rare cell progenitor. This pseudotime analysis predicts that Type-C ionocytes are a committed precursor of Type-A and Type-B ionocytes.

Details of the revisions are provided below in the point-by-point responses to the reviewers' comments. The reviewers' queries/comments are marked in italics by (Q) and responses are marked by (R) in non-italics font. Revisions to the manuscript are marked in blue font to assist the reviewers in seeing the major edits. Figure citation are all marked in red, even if it was a new figure.

Reviewers' comments

Reviewer #1 (Remarks to the Author):

Q1. Conclusions: Are generally strongly supported by experimental data. Multiple in vivo and in vitro approaches provide orthogonal validity to the work. A critical issue regarding the major conclusions relates to the interpretation of the impact of FOXI1 deletion data. Since the authors data demonstrate that FOXI1 gene deletion may alter the numbers and functions of other epithelial cells, including tuft and NE cells, the authors should discuss that progenitors of all these cell types and their functions may be altered after deletion of FOXI1-Cre.

Specific suggestions for improvement: This leads to the major limitation of data presentation in what is

otherwise a comprehensive study – the lack of *in vivo* lineage tracing data to support the *in vitro* observation of a common progenitor of rare cell types in the lung with expresses FOXI1. At minimum, data from a lineage trace with early labeling of FOXI1+ cells (presumably a subset of basal cells?) during lung development leading to labeling of tuft, NE, and ionocytes later in life is needed to validate the lineage relationships suggested by the ALI culture.

R1. We appreciate the referee's concern about the functional implications if FOXI1-Cre is active in a rare cell progenitor and FOXI1-KO alters rare cell frequencies. The referee suggested an excellent experiment to perform *in vivo* lineage tracing during airway development. To this end, we performed *in vivo* fate mapping of FOXI1-Cre lineages in neonatal and adult ferrets. These new studies provide support for a common FOXI1-lineage rare cell progenitor and also demonstrate that FOXI1-Cre lineage tracing of tuft cells, PNECs, and pulmonary ionocytes is only detectable during an early postnatal window. FOXI1-Cre lineage tracing of adult ferrets only labels ionocytes — 95.7% of FOXI1-Cre traced EGFP+ cells in adult ferrets expressed the ionocyte specific marker ATP6V1G3; note ~4% of ionocytes in mature ALI do not express ATP6V1G3 by scRNA-seq. Thus, we conclude that functional studies in adult ferret or mature ALI that involved FOXI1-Cre mediated CFTR deletion should not impact tuft cell and PNECs functions. Furthermore, tuft cells and PNECs do not express CFTR. We have highlighted key findings from these studies below.

- 1) One month old FOXI1-Cre^{ERT2}::ROSA-TG ferrets were induced with tamoxifen on five sequential days and harvested one week after the last tamoxifen injection for analysis. Results demonstrated FOXI1-Cre traced EGFP+ cells included ionocytes (74.5%, ATP6V1G3+), PNEC (8.6%, SYP+) and tuft cells (6.5%, TRPM5+) (Extended Data Fig. 11a-e). Lineage tracing of adult FOXI1-Cre^{ERT2}::ROSA-TG ferrets demonstrated 95.7% of EGFP+ were also ATP6V1G3+ ionocytes and 4.3% of EGFP+ cells were ATP6V1G3- negative (Extended Data Fig. 11a-b). Of note, ~4% of FOXI1-Cre traced ionocytes in ALI cultures do not express ATP6V1G3 by scRNA-seq (zero expression). We observed no traced PNECs or tuft cells in adult ferrets. Thus, we conclude that the FOXI1-lineage rare cell progenitor is most active during ferret airway development, which continues postnatally in ferret. In adult ferrets, only mature ionocytes are labeled by FOXI1-Cre^{ERT2}.
- 2) Experiments performed but not included in the revision. Ferret airways develop postnatally and become fully ciliated by ~3-4 weeks of age. The postnatal time point for rare cell specification was unclear. We also performed lineage labeling in 1 week old FOXI1-Cre^{ERT2}::ROSA-TG ferrets, harvested at 2 weeks post-labeling, and found very few lineage labeled cells (15-60 cells in the trachea of two animals). Notably, the majority of lineage-labeled cells were PNECs (~60%). While this finding is interesting, we felt the number of quantifiable cells were too low to draw firm conclusions and thus have not included the data within the manuscript. While the tracing of 1 month old animals may underrepresent tuft and PNEC tracing by FOXI1-Cre, we believe it makes the point when compared to adult animals.
- 3) We identified synaptophysin (SYP) as a ferret PNEC marker by scRNA-seq and immunofluorescence staining (Extended Data Fig. 11d, g). In one month old FOXI1-Cre^{ERT2}::ROSA-TG traced animals, FOXI1-Cre lineage traced SYP+ PNECs interact with sensory nerve fibers (Extended Data Fig. 11d). Extended Data Fig. 11g,h quantifies the reduction in SYP+ PNECs and ATP6V1G3+ ionocytes in FOXI1-KO adult trachea, as compared to WT.
- 4) We identified TRPM5 as a ferret tuft cell marker by scRNA-seq and immunofluorescence staining (Extended Data Fig. 11e,f). In Extended Data Fig. 11e, high resolution images show FOXI1-Cre traced tuft cells colocalized with TRPM5 in 1 month old ferrets. However, there were no traced tuft cells in adult ferret trachea (Extended Data Fig. 11f).

Q2. The *CFTR* KO experiments would benefit from additional discussion of the lineage tracing results. Since *FOXI1-CreT* is likely expressed in progenitors from which a number of different airway cells are derived, can these non-ionocytes cells derived from *FOXI1+* lineage contribute to the phenotype? Which submucosal gland regions, ducts, glands, or cells and in which cells (mucus, serous) were lineage labeled and expressed high levels of *CFTR*?

R2. We thank the reviewer for raising the question of whether non-ionocyte cells derived from the *FOXI1*-lineage rare cell progenitors contribute to the phenotype we found following *CFTR* deletion. As discussed in R1, using trachea whole mount staining we find that lineage tracing of adult *FOXI1-Cre^{ERT2}::ROSA-TG* ferrets only labels ATP6V1G3⁺ ionocytes (Extended Data Fig. 11a,b,f). We couldn't find *FOXI1-Cre* traced tuft and PNECs in adult traced tracheas. scRNA-seq results showed *CFTR* expression was found to be high in pulmonary ionocytes and low in a subset of secretory cells and basal cells, but absent expression in tuft cells and PNECs. We also performed *CFTR* protein staining in adult WT and *FOXI1-Cre^{ERT2}::ROSA-TG* ferret trachea (Extended Data Fig. 12b,c,h). The results show that *CFTR* protein was located in an "apical cap" in ATP6V1G3⁺ pulmonary ionocyte (Extended Data Fig. 12b,c; Supplementary Video 7) and *FOXI1-Cre* labeled EGFP⁺ ionocytes in *FOXI1-Cre^{ERT2}::ROSA-TG* trachea (Extended Data Fig. 12h). Given that in mature ferret airway epithelia 95.7% of *FOXI1-Cre* labeled EGFP⁺ were ATP6V1G3⁺ ionocytes and the 4.3% of EGFP⁺ cells were ATP6V1G3⁻ (which correlated with scRNA-seq data on traced ionocytes), we believe it is highly unlikely that *FOXI1-Cre* mediates *CFTR* deletion in non-ionocytes cell types contributes to the *CFTR*-dependent phenotype we described.

Regarding the second referee's comment, we found *FOXI-Cre* lineage labeled cells primarily in KRT5⁺ submucosal gland primary ducts (ciliated ducts) and NKA⁺ collecting ducts, although they can also be found in NKA⁺ tubules to a lesser extent; all traced cells expressed the ionocyte marker ATP6V1G3 (Extended Data Fig. 12i,k; Extended Data Fig. 13c,d). We realized neither of these markers are specific for various glandular compartments, but given the low frequency of ionocytes, we relied heavily on whole mount staining, morphology/structure, and proximity to gland duct opening when we called out glandular ducts and tubules. *CFTR* is expressed at lower levels in both mucous and serous cells of submucosal glands, but our new data shows these cell types are not labeled by *FOXI-Cre* since all the cells are ATP6V1G3⁺.

Q3. The finding that dehydration and ASL-induced ionocyte influenced ionocyte activity and number are important findings. Can the authors comment on the mechanism for sensing? Is it osmolality or changes in ion concentrations or other mechanical stimuli, for example, shear stress and/or ciliary activity?

R3. The reviewer raises an important question, but one for which we do not have definitive answers. All mechanisms proposed are possible. To partially address the reviewer's question, we have added hypoosmotic stress experiments (Extended Data Fig. 13a,b and Fig. 4h,i) to supplement the hyperosmotic stress studies included in the original manuscript. This new data demonstrates that hyperosmotic and hypoosmotic conditions produce opposing effects on ASL volume, dehydration vs over hydration, respectively (Fig. 4h). We hypothesized that these opposing stresses on the maintenance of ASL volume might promote expansion of different ionocyte subtypes that control salt and fluid absorption and secretion. Indeed, both hyperosmotic and hypoosmotic induced ionocyte numbers (Fig 4h; Extended Data Fig. 13a,b). Hyperosmotic stress decreased expression of *CFTR* and the Type-B ionocyte marker *ID3*, while increasing the expression the Type-C ionocyte marker *CXCL17*, while the Type-A marker gene *SCNN1A* remained unchanged (Fig. 4i). By contrast, hypoosmotic stress increased expression of *CFTR*, Type-B ionocyte marker *ID3*, and Type-A ionocyte marker *SCNN1A* (ENaC), while expression of the Type-C marker *CXCL17* remained unchanged (Fig. 4i). Based on enriched channelomes between the subtypes, Type-A ionocytes appear best suited for fluid absorption since they are enriched in CLCNKA, BSND, SCNN1A (ENaC alpha subunit), SCNN1B (ENaC beta subunit). Thus, the induction of Type-A ionocytes under hypoosmotic stress (i.e., over hydrated ASL or increased ASL height) is consistent with the expansion of ionocytes with a greater capacity to facilitate

salt and fluid absorption (i.e., channel composition enriched for basolateral CLCNK outward rectifying chloride channel complex [CLCNKA / BSND], CFTR, and ENaC). Given our findings that hyperosmotic and hypoosmotic conditions both induce basal cell specification of ionocytes, we hypothesize that a “sensing” mechanism likely occurs early during polarization and establishment of the ASL. Thus, we favor a sensing mechanism that is impacted by the biophysical properties of fluid at the airway surface. We have expanded a bit in the discussion of these topics. We realize that defining this mechanism is important, however, getting leads on the pathways and signals involved will like require more extensive pulse-seq experiments over the time course of ionocyte specification under different type of environmental stress.

Q4. Can the authors comment on the potential impact of disrupting tuft and NE cells on the ionocyte deficient phenotype?

R4. The reduction in PNEC numbers and increased Tuft cell abundance was observed in *FOXI1*-KO ALI cultures and we have now better confirmed this finding for PNECs with quantification (**Extended Data Fig. 11g,h**). While we cannot rule out that subtle changes in PNEC and Tuft cell numbers produce a phenotypic alteration in *FOXI1*-KO ferrets (e.g., changes in airway immunity, since PNEC producing bioactive amines and neuropeptides PMID 33882306), since PNECs and Tuft cells do not express CFTR mRNA at the single cell level, we believe they are unlikely to contribute to altered airway fluid regulation and clearance. We have commented in the revised discussion that it remains unclear if the altered PNEC and tuft cell frequencies in *FOXI1*-KO ferrets impact the ionocyte-deficient phenotype.

Q5. Are genetic expression changes in other epithelial cell types, for example, basal, secretory, and ciliated cells likely to be regulated or contribute to the ionocyte changes in airway fluid homeostasis? Can changes in gene expression in other epithelial cell types alter ASL?

R5: We observed transcriptional changes in other epithelial cell types of *FOXI1*-KO airway cultures (**Extended Data Fig. 4h**). We highlighted differential channel expression within various epithelial cell types derived from WT and *FOXI1*-KO ALI cultures. We believe these changes are secondary consequences of altered ASL volume caused by ionocyte depletion. ASL height is primarily controlled by two ion apical channels: epithelium sodium channel (ENaC) and CFTR. As we showed in **Figure 3e**, both CFTR and ENaC functional required subunit (SCNN1G) were highly enriched in ionocyte. Alterations to channel expression in secretory cells and ciliated cells are likely compensatory and certainly could partially counteract for the loss of ionocytes in maintaining fluid balance for *FOXI1*-KO epithelia. For example, while CF and *FOXI1*-KO airway epithelium absorb fluid more slowly than WT epithelia, they non-the-less do absorb fluid and then become dehydrated. This suggests that there are likely multiple mechanisms for fluid transport. Ionocytes appear to function as rapidly modulators of fluid homeostasis and we believe their association with submucosal glands reflects their involvement in equilibrating gland secretions at the airway surface. Other cell types, for example ciliated cells and secretory cells may impact fluid movement on a finer scale more slowly through aquaporins (some of which were upregulated in *FOXI1*-KO epithelia).

Q6. The pulse-seq experiments suggested a common progenitor for ionocytes, tuft, and NE cells; does in vivo lineage labeling support this or as the authors suggest there are findings related to in vitro conditions and potential plasticity of shared progenitors?

R6: In vivo lineage labeling of one month old *FOXI1*-Cre^{ERT2}::ROSA-TG ferrets demonstrated lineage labeling of tuft cells, PNEC, and pulmonary ionocytes (**Extended Data Fig. 11**). Thus *in vivo* lineage labeling supports the *in vitro* pulse-seq experiment. By contrast, lineage tracing of adult *FOXI1*-Cre^{ERT2}::ROSA-TG ferrets only labels ionocytes (**Extended Data Fig. 11**). Furthermore, we have now performed in-depth trajectory analysis of ferret rare cells (**Extended Data Fig. 14**) using a partition-based graph abstraction algorithm (PAGA) [PMID: 30890159]. The low-dimensional PAGA embedding recapitulated the expected cellular topology, with putative progenitor cells linked with each of the mature cell type clusters (**Extended Data Fig. 14d**). Finally, to identify cell state transitions, we fit an

elastic principal graph (PMID: 33286070) to the data, which identified a branching trajectory, consistent with interpretation of the progenitor cluster as the precursor for all three of the mature rare types. By projecting the cells in each ionocyte subcluster onto the principal graph topology (Extended Data Fig. 14a,b), we observed that the Type-C ionocytes are most similar to the rare cell progenitor (Extended Data Fig. 14b), which would be consistent with this subtype giving rise to both Type-A and Type-B ionocytes. Supporting this notion is our data (Fig. 4f) demonstrating that Type-C ionocytes can replicate and have a proliferative transcriptomic signature (Fig. 4d). Type-C ionocytes were also the least abundant of the ionocyte subtypes, which is consistent with a progenitor cell state. For these reasons, we have classified Type-C ionocytes as an ionocyte precursor state in the revised manuscript.

Q7. Regarding hyperosmolar stress and numbers of lineage trace ionocytes; were ion and lineage conditions maintained during the stress? Can the authors comment on the signals that are inducing increased ionocytes lineage labeled cells?

R7: Ionic conditions were maintained throughout the hyperosmolar stress experiment. We are a bit unclear about the meaning of the question about whether *lineage conditions were maintained during the stress*. When *FOX11-Cre* fate mapping was used for quantification of ionocytes, the differentiated epithelium was traced at full differentiation (17-20 days as stated in the Figure 4h legend), so there was no lineage labeling of PNECs or tuft cells. Additionally, we have also used ATP6V1G3 staining to quantify ionocytes and see the same results.

The reviewer raises an important question about the signals that induce ionocyte expansion under osmotic stress. The fact that hyperosmotic conditions induce basal cell specification of ionocyte only in actively differentiating basal cells (but not on fully differentiated epithelium), does suggest that the “sensing” mechanism likely occurs early during polarization and establishment of ASL layer. We hypothesize that basal cells may have a sensing mechanism for the level of ASL hydration that impacts specification of ionocytes. With this hypothesis, we have added additional data demonstrating that hypoosmotic conditions also induces ionocyte formation during basal cell differentiation (Fig. 4h, Extended Data Fig. 13b). Since hyperosmotic and hypoosmotic conditions produce opposing effects on ASL (dehydration under hyperosmotic conditions vs over hydration under hypoosmotic conditions) (Fig. 4h), we favor a sensing mechanism that is impacted by the biophysical properties of fluid at the airway surface (see response to Q3). A decrease in the abundance of Type-B ionocyte when ASL is dehydrated, and an increase in Type-B ionocytes when ASL is over hydrated, is consistent with an ASL sensing mechanism that influences basal cell specification of ionocyte subtypes. Although not included in this paper, we have mapped the timing of ionocyte formation during basal cell differentiation and it occurs at around 5-7 days after lifting to ALI (full differentiation at 21 day). Thus, altered ASL properties induced by changes in osmolarity may signal basal cells to produce more ionocytes with specific phenotypes that may attempt to compensate for ASL over hydration or dehydration. We have expanded a bit in the discussion of these topics.

Reviewer #2 (Remarks to the Author):

Q1. A critical concern is that single cell data and customized code for the study are currently not available for reviewer assessment.

R1: We have uploaded all data related to the study to GEO, accession (GSE233654), available at: <https://www.ncbi.nlm.nih.gov/geo/query/acc.cgi?acc=GSE233654>. The token ‘cronugegbbhjuv’ should be used for reviewer access. The vast majority of the code used is standard, but we have uploaded the customized code used to GitHub (https://github.com/yifand64/ferret_ionocyte).

Q2. Authors preformed interspecies comparisons of rare cell types transcriptomes using publicly available scRNA-seq datasets and concluded that the human pulmonary ionocyte was significantly

more similar to ferret than to mouse ionocytes. From Fig5B UMAP, one can clearly see the batch difference by species. More details are needed about how well they integrated data and removed the batch effect since this will largely affect the results and conclusion. Authors can show how well the integration worked using visualizations and standard metrics such as kBET or LISI. Can authors comment on why ferret ionocytes are transcriptionally more similar to that of human ionocytes; while ferret Tuft and PNE are more similar to corresponding cell types in mouse (Fig 5I)

R2: To examine why human and ferret ionocytes appear more similar than either PNECs or tuft cells using a transcriptome wide aggregate measure (Pearson's R correlation, **Fig. 5I**), we evaluated gene-level information using scatter plots (**Figure R1**). We found that human ionocyte gene expression was very much recapitulated in ferrets, while particularly for tuft cells, cassettes of particular genes were markedly different. For example, the eicosanoid synthesis enzymes *HPGD* and *ALOX5AP* were much higher in ferret than human, perhaps indicating that lipid mediators are less used by human tuft cells.

Regarding the question of data integration, we would argue that examining whether or not data mixes by species is distinct from the usual consideration of a batch effect – which considers whether data are well-mixed by replicates that are typically intended to be biologically similar. In the latter case, it is typically the goal to minimize variation between replicates and any lack of mixing indicates undesirable variation – a batch effect. However, in the case of data from multiple species, there are likely to be clear and biologically interesting differences between species – both in transcriptional programs within a cell type and the relative proportions of different cell types. Thus, *we would not expect to see perfect mixing*, especially for data as high resolution as single-cell RNA seq.

As described in the manuscript, we integrated data using the 'anchor' based approach implemented in Seurat [PMID: 31178118]. To assess the extent of this type of mixing/batch effect when integrating data from multiple species, we implemented the use of kBET and LISI metrics as suggested by the reviewer. As noted by its authors, kBET is a particularly sensitive tool [PMID: 30573817]. To assess whether kBET was suitable for this task, we applied it to simulated data with only an extremely marginal batch effect, but kBET nonetheless suggested an average rejection rate of 1, indicating perfect confidence of the presence of a batch effect (**data not shown**). Given this degree of 'oversensitivity', we turned to the LISI algorithm.

We computed the LISI score for each cell by species, along with the species composition of each cell type cluster. We distinguished the putative progenitor cells by a further round of unsupervised clustering (**Figure R2A**). Given that there are three species being integrated (human, ferret, mouse), perfect interspecies mixing would result in a LISI score near 3, while poor mixing would yield a score closer to 1. We observed that tuft cells had the lowest LISI score, indicating relatively poor mixing by species – i.e., a 'batch effect' and/or clear species-divergence in transcriptomes (**Figure R2B-C**). We examined the prevalence of the three cell types, and found that in these data, tuft cells were much more numerous in mice relative to humans and ferrets (**Figure R2D-E**), which likely also contributed to the low LISI score. On the other hand, ionocytes – which were more evenly drawn from all three species – demonstrated superior mixing, and the putative rare cell progenitor was better mixed than PNECs, which were quite sparse in the human data. This strong relationship between the composition of each cell type cluster and its LISI distribution underscores that clusters can only be perfectly mixed when evenly drawn from all species in equal numbers. Taken together, we conclude that with the exception of tuft cells, which as we described above appear the most divergent of the three rare cell types, our approach to data integration resulted in sufficient mixing and enabled characterization of the novel rare cell progenitor, which perhaps most importantly, we have now validated using independent genetic evidence (**Extended Data Fig. 11**).

Reviewer Figure 1. Ferret ionocytes and PNECs recapitulate the human transcriptome while tuft cells diverge. Scatter plots show the expression in human (x axis) vs ferret (y axis) for 2000 most highly variable genes detected (points) in each of the three rare cell types (each respective plot). Labeled genes are substantially divergent – 2-fold higher in human than ferret (red), 4-fold higher in ferret than human (blue) – or highly co-expressed – both species $\log_2(\text{TPM}+1) > 1.5$ (green). Pearson's R correlation is also shown (bottom right), and dotted line is the line $y=x$.

Reviewer Figure 2. Characterization of mixing by species after scRNAseq integration of rare cells. A-B. UMAP embedding of 4,180 single airway epithelial transcriptomes (points) from the three 'rare' cell types (ionocyte, PNEC, tuft), colored by unsupervised clustering (A) and Local Inverse Simpsons Index (LISI) (B). C. Violin plots show the distribution of LISI scores (x axis) for cells in each cluster (y axis). D. Composition of each cell type cluster drawn from each of the three species (color legend). E. Species evenness or 'diversity' (higher values indicate a more even distribution) of each cell type estimated using the Inverse Simpsons Index (core of the LISI algorithm).

Q3. "FOX11+ common rare cell progenitor" is a novel and important observation, little is known about the paths of the common progenitor takes to differentiate into these rare cell types, further characterization and validation using bioinformatics and experimental approaches are needed to support the conclusion.

R3: We thank the reviewer for this important comment. We have now performed *in vivo* lineage tracing during neonatal ferret airway development that demonstrates a FOXI1-Cre lineage rare cell progenitor gives rise to tuft cells, PNEC, and pulmonary ionocytes. As predicted from *in vitro* results in ALI cultures, FOXI1-Cre lineage tracing of adult ferrets only labels ionocytes. We have highlighted key findings from these studies below:

- 1) One month old *FOXI1-Cre^{ERT2}::ROSA-TG* ferrets were induced with tamoxifen on five sequential days and harvested one week after the last tamoxifen injection for analysis. Results demonstrated *FOXI1-Cre* traced EGFP⁺ cells included ionocytes (74.5%, ATP6V1G3⁺), PNEC (8.6%, SYP⁺) and tuft cells (6.5%, TRPM5⁺) (Extended Data Fig. 11a-e). Lineage tracing of adult *FOXI1-Cre^{ERT2}::ROSA-TG* ferrets demonstrated 95.7% of EGFP⁺ were also ATP6V1G3⁺ ionocytes and 4.3% of EGFP⁺ cells were ATP6V1G3⁻ negative (Extended Data Fig. 11a-b). Of note, ~4% of *FOXI1-Cre* traced ionocytes in ALI cultures do not express ATP6V1G3 by scRNA-seq (zero expression). We observed no traced PNECs or tuft cells in adult ferrets. Thus, we conclude that the *FOXI1*-lineage rare cell progenitor is most active during ferret airway development, which continues postnatally in ferret. In adult ferrets, only mature ionocytes are labeled by *FOXI1-Cre^{ERT2}*.
- 2) Experiments performed but not included in the revision. Ferret airways develop postnatally and become fully ciliated by ~3-4 weeks of age. The postnatal time point for rare cell specification was unclear. We also performed lineage labeling in 1 week old *FOXI1-Cre^{ERT2}::ROSA-TG* ferrets, harvested at 2 weeks post-labeling, and found very few lineage labeled cells (15-60 cells in the trachea of two animals). Notably, the majority of lineage-labeled cells were PNECs (~60%). While this finding is interesting, we felt the number of quantifiable cells were too low to draw firm conclusions and thus have not included the data within the manuscript. While the tracing of 1 month old animals may underrepresent tuft and PNEC tracing by *FOXI1-Cre*, we believe it makes the point when compared to adult animals.
- 3) We identified synaptophysin (SYP) as a ferret PNEC marker by scRNA-seq and immunofluorescence staining (Extended Data Fig. 11d, g). In one month old *FOXI1-Cre^{ERT2}::ROSA-TG* traced animals, *FOXI1* lineage traced SYP⁺ PNECs interact with sensory nerve fibers (Extended Data Fig. 11d). Extended Data Fig. 11g,h quantifies the reduction in SYP⁺ PNECs and ATP6V1G3⁺ ionocytes in *FOXI1-KO* adult trachea, as compared to WT.
- 4) We identified TRPM5 as a ferret tuft cell marker by scRNA-seq and immunofluorescence staining (Extended Data Fig. 11e,f). In Extended Data Fig. 11e, high resolution images show *FOXI1-Cre* traced tuft cells colocalized with TRPM5 in 1 month old ferrets. However, there were no traced tuft cells in adult ferret trachea (Extended Data Fig. 11f).

Regarding further bioinformatic support for the existence of a distinct rare cell progenitor please refer to Extended Data Fig. 14 and Q4/R4 below.

Q4. For example, authors could apply *scVelo* to reconstruct the lineage relationships between the common rare cell progenitor and ionocyte, tuft and PNE cells to infer their developmental trajectories.

R4. We thank the referee for this comment. We would like to point out that the use of unspliced messages for trajectory analysis come with limitations when using an orphan genome such as ferret with suboptimal exon/intron annotation. Furthermore, in contrast to Smart-Seq2, 10X sequences are heavily biased toward the last exon and 3' UTR, yielding no intronic information. Nonetheless, in response to the review's comment, we have applied several computational approaches to infer relationships between the common progenitor and the mature rare cell types. First, following the reviewer's suggestion, we applied *scVelo*. However, perhaps unsurprisingly given the above caveats, diagnostic plots indicated that *scVelo* did not produce accurate models of splicing dynamics in our

dataset (Figure R3A), as these phase portraits provide a guide of whether scVelo is appropriate [PMID: 37002403]; Consistently, the inferred trajectories were not biologically plausible (Figure R3B). For example, ionocyte→PNEC→Tuft and secretory→basal.

We therefore turned to the partition-based graph abstraction algorithm (PAGA) [PMID: 30890159], a trajectory inference algorithm that does not depend on modeling transcript splicing and which has performed well in comparative benchmarks [PMID: 30936559]. We isolated the 1,497 rare cell types (PNECs, tuft cells, ionocytes, and rare cell progenitor) from our scRNAseq profiling of ferret airway epithelial cells and fit the PAGA algorithm (Extended Data Fig. 14a-c). The low-dimensional PAGA embedding recapitulated the expected cellular topology, with putative progenitor cells (as annotated by unsupervised clustering in response R2 above) linked with each of the mature cell type clusters (Extended Data Fig. 14d). Finally, to identify cell state transitions, we fit an elastic principal graph (PMID: 33286070) to the data, which identified a branching trajectory, consistent with interpretation of the progenitor cluster as the precursor for all three of the mature rare types.

Reviewer Figure 3. scVelo does not accurately model 10X data from ferret genome. A. Phase state plots showing fit of scVelo's dynamical model (solid line) to spliced (x axis) vs. unspliced (y axis) read counts. **B.** UMAP embedding of entire ferret airway epithelial cell dataset (77,099 cells, embedding identical to main text Fig. 3b) and (spurious) cell fate transitions inferred by scVelo.

Our new trajectory inference described above provides a framework to assess the relationship of ionocyte subtypes. By projecting the cells in each ionocyte subcluster onto the principal graph topology (Extended Data Fig. 14a,b), we observed that the Type-C ionocytes are most similar to the rare cell progenitor (Extended Data Fig. 14b), which would be consistent with this subtype giving rise to both Type-A and Type-B ionocytes (Extended Data Fig. 14c). Supporting this notion is our data (Fig. 4f) demonstrating that Type-C ionocytes can replicate and have a proliferative transcriptomic signature (Fig. 4d). Type-C ionocytes were also the least abundant of the ionocyte subtypes, which is consistent with a progenitor cell state. For these reasons, we have classified Type-C ionocytes as an ionocyte precursor state in the revised manuscript.

Q5. Can authors comment on the different observation and conclusions from the previous scRNA-seq, lineage and CRISPR knockout studies suggesting that novel tuft-like cells are the likely progenitor of PNE and ionocytes; and epithelial CFTR activity is not reduced by loss of ionocytes. (Goldfarbmuren et al).

R5: [This has been redacted]). We believe our *FOXI1*-Cre rare cell progenitor lineage findings are not inconsistent with that study. Goldfarbmuren et al. found that "48% of *FOXI1*⁺ cells exhibited an ionocyte pattern (*CFTR*⁺/*POU2F3*⁻), whereas 38% of *FOXI1*⁺ cells exhibited a tuft-like pattern (*CFTR*⁻/*POU2F3*⁺)". Thus, what Goldfarbmuren calls a novel tuft-like variant could be the same cell state we are calling a "rare cell progenitor". Nonetheless, we now have definitive evidence using *FOXI1*-Cre lineage tracing that a *FOXI1*-lineage progenitor gives rise to all three rare cell types during postnatal airway development in the ferret (Extended Data Fig. 11).

Regarding the second comment about ionocyte function, as the reviewer points out our studies are inconsistent with the Goldfarbmuren CFTR functional studies following CRISPR/Cas9 disruption of *FOXI1*. What is missing from that paper is molecular confirmation of the efficiency of FOXI1 disruption. They had no selection system to isolate cells that contained near complete biallelic indels. A small fraction of *FOXI1* wildtype or heterozygous basal cells could compensate to produce ionocytes, in a manner similar to our hypertonic and hypotonic studies that demonstrate reactive ionocyte expansion during disruption of ASL balance. Rather they used FOXI1 protein staining to determine the extent of knockout and we would argue this approach does not have the sensitivity or accuracy to determine the extent of *FOXI1* disruption. Thus, it is unclear from this study if incomplete *FOXI1* disruption contributed to the lack of change in CFTR current. Human basal cells also lose differentiation capability as they are passaged and CFTR currents are lost first prior to a decline in ciliated and secretory cells [PMID: 27320041], which makes it more difficult to manipulate genes and study CFTR function. This loss in CFTR current with increased passage of human basal cells appears to be related to a loss in ionocytes differentiation. To further support our findings, we have added new data evaluating short circuit currents of WT and FOXI1-KO tracheal tissue and the results reproduced the studies on differentiated ALI cultures. *FOXI1*-KO trachea demonstrated significantly decreased Forskolin & IBMX induced CFTR currents that were inhibitable by the CFTR channel blocker GlyH101 (**Extended Data Fig. 12a**).

[This has been redacted]

Q6a. Authors pointed out that single-cell data significantly only captured <3% of ionocytes in proximal airway epithelia, and the ionocyte damage during 10X capture increased ambient RNA for genes highly expressed in ionocytes. This is an important finding to address a controversial research issue. More evidence is needed to support this important finding. If the loss of ionocytes occur during single cell sample preparation, it is important to show that a higher frequency of ionocytes present in ALI prior to single cell sequencing.

R6a. To experimentally address the higher frequency of ionocytes present in ALI prior to single-cell sequencing, we performed cell viability counts prior to and following FACS of *FOXI1*-CreER::*ROSA*-TG lineage traced ferret ionocytes. We used two methods to assess viability prior to 10X sequencing: 1) a live/dead FACS assay showed 61.75% +/- 6.29% tdTomato⁺ cells and 73% +/- 2.51% EGFP⁺ cells were viable (See below **Figure R4a,b**) and 2) a trypan blue assays show 83% +/- 2.02% cells are alive and 18% +/- 1.8% cells are dead after FACS prior to 10X chip G loading (See below **Figure R4c,d**). These data demonstrate acceptable cell viability prior to 10X chip G loading and thus the FACS isolation cannot account for the loss in numbers of ionocytes detected in scRNAseq data. Of note, best practice single cell sample preparation from 10X protocols include cell viability greater than 70%. Our sample preparation met this 10X protocol cell viability requirement.

In total we loaded 14,966 EGFP⁺ cells spiked with 77,145 tdTomato⁺ cells into the 10X chip, but cell recovery was only 2,370 EGFP⁺ cells, as compared to 37,299 tdTomato⁺ cells. The EGFP recovery rate was 15.8% as compared to 48% for tdTomato⁺ other cell types. Using standard 10X protocols, approximately 40%-65% of cells loaded into the 10X genomics chip will be partitioned and recovered. Considering that hydroxytamoxifen labeling of actively differentiating ALI cultures labels ~40% ionocytes and 60% PNEC and tuft cells (**Extended Data Fig. 9e**), we estimate that 5,986 ionocytes were loaded onto the 10X chip. This produced a recovery rate of 7.3% for 434 ionocyte as compared to the 48% for other cell tdTomato⁺ types. Thus, ionocytes do appear to be more susceptible to damage during 10X capture compared to other airway epithelial cell types. Of note, as originally written, the 2.9% 10X recovery rate for ionocytes was discussed prior to our new finding that PNECs and tuft cells were traced using a pulse-seq approach (i.e., hydroxytamoxifen treatment during basal cell differentiation). We have revised sections of the text to account for the discovery that *FOXI1*-Cre labels other rare cells in the calculation of the 7.3% ionocyte recovery rate during 10X sequencing.

Reviewer Figure 4. Cell viability test prior to 10X Chip G loading. **A.** FACS of *FOXI1*^{CreERT2}::*ROSA*-TG cells from differentiated ALI cultures treated with hydroxy-tamoxifen. Hoechst 33258 (250µg/ml) was used to label dead cells. FACS plot represents viable tdTomato⁺ and EGFP⁺ cells. **B.** Live/dead FACS assay demonstrates 61.8% +/- 6.3% tdTomato⁺ cells and 73% +/- 2.5% EGFP⁺ cells were viable. **C.** Example assay for trypan blue stained *FOXI1*^{CreERT2}::*ROSA*-TG traced airway epithelial cells following FACS. Green dots in the image represent live cells and red dots in the image represent dead cells. **D.** Quantification of live and dead cells from the trypan blue assay. Graph shows 83% +/- 2.0% cells were alive and 18% +/- 1.8% cells were dead.

Q6b. *Extended Data Fig. 8A-C* showed a significant correlation of *CFTR*, *ASCL3*, *FOXI1* expression in secretory cell with the number of ionocytes detected in each ALI culture. Please provide the correlation plots for other cell types since ambient RNA in the environment would potentially influencing the barcode of all different cell types.

R6b. We initially focused on secretory cells because a previous study had suggested that this subtype 'dominates' *CFTR* expression [PMID: 33321047]. In response to the reviewer's request, we have expanded our analysis of the relationship between proportion of ionocytes in each sample and the detection of ionocyte-associated transcripts to include all cell types (*Extended Data Fig. 15*). This revealed a general trend of positive correlations for the canonical ionocyte marker transcripts *ASCL3*, *FOXI1*, *CFTR*, *BSND*, *CLCNKA*, and *ATP6V1C2* in secretory, but also in ciliated cells (*FOXI1*, *ATP6V1C2*, *CLCNKA*, *BSND*), and to a lesser extent in basal cells (*BSND*) (*Extended Data Fig. 15a*). To make this analysis more systematic, we compared the same observed correlations to the top 30 most ionocyte-specific transcripts (*Extended Data Fig. 15b, right panel*), using the top 30 basal-specific transcripts as a control set of non-ionocyte-associated transcripts (*Extended Data Fig. 15b, left panel*). We reasoned that if ionocyte damage or loss is contributing to spurious detection of ionocyte-associated transcripts in non-ionocytes, then we would observe a correlation with ionocyte proportion for the most ionocyte-specific transcripts (ionocyte markers), *but not* for the non-ionocyte-associated transcripts (basal markers). This trend was indeed clearly observable, particularly for secretory cells, which showed a significant positive correlation for their expression of 21 of 30 ionocyte markers, while none showed a negative correlation. In contrast, secretory cell expression of only 1 of 30 basal markers was positively correlated, and two were negatively correlated. We also observed substantial putative spurious expression of ionocyte transcripts in ciliated cells (13/30) and basal cells (5/30). Taken together, this analysis provides evidence that damage during 10X capture may lead to detection of ambient RNA for genes highly expressed in ionocytes.

The finding of stronger correlations for secretory cells suggests this cell type may bind mRNA at higher levels than other cell types. Furthermore, the apparent transcript specific nature of ambient RNA contamination is puzzling. For example, ionocyte enriched transcripts *ASCL3*, *ATP1B1*, *ATP6V0C*, and *ATP6V1C2* are highly expressed, yet only two of these transcripts (*ASCL3* and *ATP6V1C2*) were positively correlated with ionocyte frequency in secretory cells. Furthermore, *ATP6V1C2* demonstrated a strong correlation across multiple cell types (secretory, ciliated, PNEC, and Tuft), whereas the strong correlation for *ASCL3* was limited to secretory cells (*Extended Data Fig. 15b, right panel*). The reason for these findings is unclear, but could involve transcript-specific RNA binding proteins that impart some level of cell type specific surface adherence. Despite a lack of understanding as to the mechanism

behind this limitation of 10X, we feel presenting all this data is important to the field and we appreciate the reviewer urging us to investigate this further.

Q7. Page 9. Authors identified signature genes of ferret pulmonary ionocytes and discussed their functional relevance to CFTR related ion transporters. Some are selectively enriched in ionocyte including CFTR, CLCNKA, BSND, ATP6V1G3, ASCL3 and SCNN1G; while some are not selective including KCNK1, KCNQ1, KCNK5, KCNMA1, SCNN1A, ATP1B1, and SLC12A2.

R7: We thank the reviewer for this comment. The pulmonary ionocyte ion channel atlas defined in this manuscript demonstrates many channels shared with analogous cells, such as kidney intercalated cells, fish gill and frog skin, thus adding value to those beyond the biomedical field.

Reviewer #3 (Remarks to the Author):

This manuscript investigated the differentiation of ionocytes and the function of CFTR in ionocytes. As the authors summarize in the paper, data in this paper revealed that FOXI1- expressing pulmonary ionocytes functionally regulate ASL volume and pH, two hallmark defects in the CF airway that lead to increased mucus viscosity and impair MCC.

Q1. I was excited about these genetic ferret models, which will provide more information to understand the functions of rare CFTR-rich pulmonary ionocytes in cystic fibrosis. However, I don't find many conceptually advances over the current CFTR and ionocyte fields, but there is a thorough extension of prior concepts to CFTR in ionocytes. Previous studies using human and ferret airway epithelial cells have already shown that the proper differentiation of ionocytes is regulated by FOXI1 and NOTCH. And the function of ionocytes is required for ion transport and maintaining airway surface physiology, including mucus viscosity (Montoro D, Nature, 2018; Plasschaert LW, Nature, 2018, Shah, VS, 2022).

R1. We thank the reviewer for the critical comment on the limited conceptual advances from our study. We recognize that the impact of our studies is perhaps equally split between technology advancements (i.e., conditional genetics in ferrets) and scientific advancements in the understanding of pulmonary ionocyte functions. The 2018 papers identified ionocytes and suggested their involvement in airway surface salt and fluid regulation with imprecise or indirect tools available at the time (mouse model, no cell type specificity in human ALI cultures, μ OCT imaging as a proxy for mucus properties). Thus, the functional roles of ionocytes in airway fluid regulation and CF pathophysiology very much remained open questions of pressing importance that we aim to address with new model systems, a point underscored by the discussion in the Shah review cited by the reviewer [PMID: 34437820].

This manuscript takes on these open questions by developing the tools to more accurately define the development and identity of ionocytes (lineage tracing in adult and early postnatal animals, defining distinct ionocyte subsets, and clarifying the localization and expression patterns of ionocytes populations throughout the airway tree), in addition to defining the *in vivo* and *in vitro* physiology and function of ionocytes in CF-relevant contexts, using direct comparison with CF models. We have added a significant amount of new data to this revision that strengthens these conceptual advances. We enumerate the novel findings below and discuss how they advance the conceptual understanding of the role of ionocytes in airway fluid regulation and CF pathophysiology.

- We show for the first time that pulmonary ionocytes are responsible for both fluid absorption and secretion by airway epithelia. We believe this is a major conceptual advance in understanding CF pathogenesis as it demonstrates the CFTR channel in ionocytes limits movement of salt, and thus water, across the airway epithelium. Prior to this work, the field had viewed CF as a defect primarily in CFTR-mediated fluid secretion by the airway epithelium.

- We demonstrate that pulmonary ionocytes transport bicarbonate to regulate airway surface liquid pH. While this was speculated in the Shah 2022 review cited, it was not shown in the 2018 *Nature* papers.
- The Montoro *Nature* 2018 paper stated “neither depth nor pH (Extended Data Fig. 10c,d) of the ASL was significantly altered in Foxi1(KO) epithelial cultures”. Furthermore, the chloride current was increased in Foxi1(KO) epithelial cultures. These findings were difficult to reconcile with known phenotypes of CF epithelia (reduced ASL pH and volume with increased ASL viscosity). The conclusion of increased Foxi1(KO) ASL viscosity in this study were also indirect using uOCT reflectance intensity and observed increased ciliary beat frequency, which was hard to reconcile with other ASL properties. Our studies utilize a species that better models CF lung disease to clarify these ionocyte functions through a direct comparison of CF and Foxi1(KO) cultures and animals, and confirm that the absence of ionocytes phenocopies CF airways. While this may not be seen as a major conceptual advance, it may explain why mice are not ideal models of CF lung disease.
- With new data in the revision, we demonstrate using *in vivo* FOXI1-Cre^{ERT2} lineage tracing (Extended Data Fig. 11b-e) the existence of a common FOXI1-lineage rare cell progenitor for tuft, PNEC, and ionocytes, supporting the conclusion of the cross-species transcriptomic analysis and *in vitro* lineage tracing of these rare cells in the original submission.
- Previously, pulmonary ionocytes were not thought to have different subtypes, both 2018 papers concluded that they are homogenous. We demonstrate the existence of three ionocyte subtypes. We also demonstrate that one subtype (Type-C) has the ability to replicate, and our new trajectory inference analysis suggests Type-C ionocytes are a precursor state that derives Type-A and Type-B ionocytes (Extended Data Fig. 14). The former is not necessarily a conceptual advance when considering the cumulative ionocyte literature in other organs and species, however, to our knowledge the later has not been shown in any type of ionocyte and thus we feel is a conceptual advance.
- While the single cell imaging of anion movement through pulmonary ionocytes may at first glance appear to be simply a technical achievement (Fig. 2c-g), conceptually it shows that pulmonary ionocytes directly transport anions. While simple in concept, it is extremely important point to the field as there is much debate about how a rare cell type (~1%) can contribute to such a large percentage of salt movement across the epithelium, leading to speculation that pulmonary ionocytes may be sensory cells that serve to activate anion movement across other cell types. Collectively our channelome analysis and single cell imaging of anion movement through ionocytes, support the concept of cell-autonomous ionocyte functions in the movement of salt and fluid.
- With new data in the revision, we now show that hyperosmotic and hypoosmotic stress both enhance ionocyte specification in the setting of dehydrated or over hydrated ASL, respectively (Extended Data Fig. 13a, b, Fig. 4h,i). Notably, hyperosmotic stress reduced Type-B and increased Type-C ionocytes (based on marker gene expression), while hypoosmotic stress increased Type-B and Type-A ionocytes. Together with the finding that CF airways have greater number of ionocytes (Extended Data Fig. 12j, k; Extended Data Fig. 13c,d), this demonstrates the concept that epithelial sensing mechanisms react to changes in osmolarity and/or ASL volume to control ionocyte numbers and phenotype. With the suggestion from this reviewer (Q2), we now also demonstrate glandular ionocyte expansion in the intralobar airways of CF ferrets.

Q2. *Although there are some differences between mouse ionocytes and ionocytes in human and ferret the experimental depletion of ionocytes in the mouse trachea perturbs epithelial physiology. What remains mysterious is the role of these rare cells in the pathogenesis of human CF. It will be more important to address the activity and function of ionocytes in the pathological settings since the authors have these very nice ferret models.*

R2: We are grateful for the referee's suggestion and have now added new data evaluating pulmonary ionocytes in WT and CF ferret models. We now show expansion of ATP6V1G3⁺ pulmonary ionocytes in submucosal glands of CF ferrets (Extended Data Fig. 12j, k; Extended Data Fig. 13c,d). In ALI cultures derived from CF ferret tracheal airway basal cells, we show a lack of CFTR-mediated currents (Extended Data Fig. 12l) and a significant increase in pulmonary ionocytes as assessed by the expression of transcription factors *FOXI1* and *ASCL3* (Extended Data Fig. 12m,n).

Regarding the referee's comments "*What remains mysterious is the role of these rare cells in the pathogenesis of human CF*", we have an alternative view. After decades of debate, there is now a fairly uniform consensus that disease pathogenesis in CF airways is the result of dehydrated ASL and poor airway clearance. Furthermore, reduced bicarbonate secretion is thought to hinder mucus unfolding. Combined, these two features lead to increased mucus viscosity in the airways and an environment conducive to bacterial growth. Collectively, our *in vitro* studies demonstrate that the lack of pulmonary ionocytes (*FOXI1*-KO epithelia) phenocopies defects in CF airway epithelia (reduced ASL volume and pH due to the lack of chloride and bicarbonate secretion, which leads to increased mucus viscosity). Our *in vivo* MCC studies comparing *FOXI1*-KO, *CFTR*-G551D, and *FOXI1*-CreER::*CFTR*^{L/L} ferrets definitively demonstrate that ionocyte-derived CFTR is required for proper clearance. What still remains to debate is the function of CFTR in different types of secretory cells (i.e., is CFTR functioning in the mucus granules to facilitate release and unfolding of mucins and/or does CFTR function within secretory cells to move salt and water across the airway?).

Q3. *Their sc-RNAseq results show that there are three subtypes of ionocytes. This result is interesting but also expected given that ionocytes are found in different parts of the airways. sc-RNAseq UMAP shows that the Type A and Type B subsets contain a similar proportion of ionocytes. Therefore, we can expect that the number of BSND negative ionocytes (Type B and Type C) will not be lower than the number of BSND positive ionocytes (Type A). But the staining results showed more BSND positive ionocytes in human and ferret airway epithelial cells. Moreover, it seems that the differential expression of BSND between Type A and Type B ionocytes will not clearly distinguish Type A and Type B ionocytes. What is the relationship and function of these different subtypes?*

R3: *BSND* is significantly enriched in Type A ionocytes as compared to Type B ($p=4.1e-7$) and Type C ($p=5.5e-11$), but nonetheless *BSND* is still expressed in Type B and C ionocytes (Figure 4a-c). We agree with the reviewer that *BSND* protein localization will not distinguish ionocyte subtypes and for this reason we did not use it for this purpose (in fact we have never found a *BSND* antibody that reacts with ferret *BSND*); but nonetheless *BSND* transcripts are enriched in Type A ionocytes and may implicate a larger role of this subtype is salt and fluid absorption, since Type A ionocytes were also enriched in *CLCNKA* – basolateral chloride channel regulated by *BSND*.

Regarding the relationship of different ionocyte subtypes, our new trajectory inference analysis now defines the relationship of the novel rare cell progenitor to the mature rare cell types (Extended Data Fig. 14c). In addition, it provides a framework to assess the relationship of ionocyte subtypes. By projecting the cells in each ionocyte subcluster onto the principal graph topology (Extended Data Fig. 14a,b), we observed that the Type-C ionocytes are most similar to the common rare cell progenitor, which would be consistent with this subtype giving rise to both Type-A and Type-B ionocytes. Supporting this notion is our data (Fig. 4f) demonstrating that Type-C ionocytes can replicate and have a proliferative transcriptomic signature (Fig. 4d). Type-C ionocytes were also the least abundant of the ionocyte subtypes, which is consistent with a progenitor cell state. For these reasons, we have classified Type-C ionocytes as an ionocyte precursor state in the revised manuscript.

Regarding the function of different ionocyte subtypes, the reviewer raises an excellent question. To address the reviewer's question, we have added hypoosmotic stress experiments (Extended Data Fig. 13b and Fig. 4h,i) to supplement the hyperosmotic stress studies included in the original manuscript. This new data demonstrates that hyperosmotic and hypoosmotic conditions produce opposing effects on ASL volume, dehydration vs over hydration, respectively (Fig. 4h). We hypothesized that these

opposing stresses on the maintenance of ASL volume might promote expansion of different ionocyte subtypes that control salt and fluid absorption and secretion. Indeed, both hyperosmotic and hypoosmotic induced ionocyte numbers (Fig 4h; Extended Data Fig. 13a, b). Hyperosmotic stress decreased expression of *CFTR* and the Type-B ionocyte marker *ID3*, increased the expression of the Type-C ionocyte marker *CXCL17*, while the Type-A marker gene *SCNN1A* remained unchanged (Fig. 4i). By contrast, hypoosmotic stress increased expression of *CFTR*, Type-B ionocyte marker *ID3*, and Type-A ionocyte markers *SCNN1A* (ENaC), while expression of the Type-C marker *CXCL17* remained unchanged (Fig. 4i). Based on enriched channelomes between the subtypes, Type-A ionocytes appear best suited for fluid absorption since they are enriched in *CLCNKA*, *BSND*, *SCNN1A* (ENaC alpha subunit), *SCNN1B* (ENaC beta subunit). Thus, the induction of Type-A ionocytes under hypoosmotic stress (i.e., over hydrated ASL or increased ASL height) is consistent with the expansion of ionocytes with a greater capacity to facilitate salt and fluid absorption (i.e., channel composition enriched for basolateral *CLCNK* outward rectifying chloride channel complex [*CLCNKA* / *BSND*], *CFTR*, and ENaC). While the functional implications of changes in the distribution of ionocyte subtypes under hyperosmotic stress are less clear from a channel standpoint, the opposite change in Type-B ionocyte marker and *CFTR* expression under hyperosmotic (decreased) or hypoosmotic (increased) stress suggests this subtype may have unique functions that limit or promote ASL absorption under hyperosmotic or hypoosmotic conditions, respectively.

Q4. The proposed models are largely dependent on the gene expression at the mRNA level. It is unclear whether these transporters and channels are expressed in heterogeneous levels or differential expression in ionocytes at different airway locations.

R4: We appreciate the referrer's constructive criticism and suggestion to dig deeper into ionocyte enriched channels with functional *in vivo* relevance. To this end, we characterized *CFTR*, *NKA* (Na-K-ATPase), and *ATP6V1G3* (H^+ transporter) expression patterns in pulmonary ionocyte in ferret trachea, primary bronchi, and intralobar bronchi (Extended Data Fig. 12b-k; Supplementary Video 7). These data demonstrated that *NKA*, *CFTR*, and *ATP6V1G3* are pulmonary ionocyte markers, similar to fish gill ionocytes. Using whole mount tissue staining, we found that *CFTR* protein localizes to what we have termed an ionocyte "apical cap". *ATP6V1G3* protein was localized to the ionocyte apical cap and cytoplasm (Extended Data Fig. 12b,c), whereas *NKA* protein was enriched to a basolateral rim around the apical cap (Extended Data Fig. 12d; Supplementary Video 7). Notably, single ionocytes can have more than one apical cap (Extended Data Fig. 12c), perhaps enhancing their capacity to direct salt and fluid movement. We have also now more clearly defined how channel composition in ionocytes differs regionally at distinct locations in the airway. For example, proximal tracheal ionocyte expressed higher levels of the *NKA* transporter as compared to primary bronchus (Extended Data Fig. 12d). Using whole mount trachea tissue staining from both WT and *FOXI1-Cre^{ERT2}::ROSA-TG* ferrets, we also found greater ionocyte numbers in trachea and primary bronchus (Extended Data Fig. 12e, f and h), while in the secondary bronchus they were primarily located to submucosal gland primary ducts and collecting ducts (Extended Data Fig. 12i; Extended Data Fig. 13c,d).

Reviewer Reports on the First Revision:

Referees' comments:

Referee #1 (Remarks to the Author):

This is a revised manuscript reporting studies in transgenic ferrets that demonstrate the important role of ionocytes in the regulation of airway fluid homeostasis and muco-ciliary clearance. Comprehensive, novel *in vivo* and *in vitro* data demonstrate the role of ionocytes as an epithelial progenitor cell and that expression of CFTR in this rare cell type regulates airway fluid /electrolyte transport critical for muco-ciliary clearance. The authors show that levels of osmolality regulate the numbers of ionocytes, their numbers and regulation of fluid homeostasis being dependent on CFTR function in the ionocyte. The work convincingly resolves several important questions in lung/CF biology regarding the importance of this relatively rare cell type and its high levels of expression of CFTR to the pathogenesis of CF. Strong *in vivo* and *in vitro* data support the authors' conclusions and its importance to the field. The authors thoughtfully addressed and discussed the issues raised by all of the reviewers and provide extensive new data that further strengthened the work. The new data demonstrate, through careful lineage analysis, the role of the ionocytes as progenitors of other rare epithelial cell subtypes and ionocyte subsets during morphogenesis-but not in adults and that production of ionocytes from basal cells is influenced by osmolality and fluid, electrolyte balance on the airway surface. The authors carefully explain differences in estimation of their cell numbers and clarify their localization along the airways. They provide evidence for an "apical cap" where transporters are colocalized on the ionocytes. An extensive "Atlas" of airway epithelial cells and similarities and differences relative to mouse and human data are included. The ferret model recapitulates many aspects of human lung structure and function and provides an important model for study and for development of new treatments for CF in the future. Extensive *in vivo*, *in vitro* and *in silico* analyses provide strong orthogonal validity to their conclusions. References are complete and appropriate. Their discussion incorporates the new data and fairly addresses conflicts in the literature while addressing unresolved issues for the field. Jeff Whitsett

Referee #2 (Remarks to the Author):

The manuscript has significantly improved. The authors have addressed my concerns.

Referee #3 (Remarks to the Author):

The authors addressed my concerns. I have no other questions.